# CountsDiff: A Diffusion Model on the Natural Numbers for Generation and Imputation of Count-Based Data

Renzo G. Soatto [1 2 3]  Anders Hoel [1 2]  Greycen Ren [1]  Shorna Alam [1]  Stephen Bates [1 2]  Nikolaos P. Daskalakis [4]
Caroline Uhler [1 2 3]  Maria Skoularidou [1 3]

## Abstract

Diffusion models have excelled at generative tasks for both continuous and token-based domains, but their application to discrete ordinal data remains underdeveloped. We present *CountsDiff*, a diffusion framework designed to model distributions on the natural numbers. CountsDiff extends the Blackout diffusion framework by simplifying its formulation through a direct parameterization in terms of a survival probability schedule and an explicit loss weighting. This introduces flexibility through design parameters with direct analogues in existing diffusion modeling frameworks. Beyond this reparameterization, CountsDiff introduces features from modern diffusion models, previously absent in counts-based domains, including continuous-time training, classifier-free guidance, and churn/remasking reverse dynamics that allow non-monotone reverse trajectories. We propose an initial instantiation of CountsDiff and validate it on natural image datasets (CIFAR-10, CelebA), exploring the effects of the introduced design parameters in a complex, well-studied, and interpretable data domain. We then highlight biological count assays as a natural use case, evaluating CountsDiff on single-cell RNA-seq imputation in fetal and heart cell atlases. Remarkably, we find that even this simple instantiation matches or surpasses the performance of a state-of-the-art discrete generative model and leading scRNA-seq imputation methods, while leaving substantial headroom for further gains through optimized design choices in future work.

[1]Massachusetts Institute of Technology (MIT), Cambridge, MA, USA [2]Laboratory for Information and Decision Systems (LIDS), MIT, Cambridge, MA, USA [3]Eric and Wendy Schmidt Center, Broad Institute, Cambridge, MA, USA [4]Boston University, Boston, MA, USA. Correspondence to: Maria Skoularidou <mskoular@broadinstitute.org>.

*Proceedings of the 43rd International Conference on Machine Learning*, Seoul, South Korea. PMLR 306, 2026. Copyright 2026 by the author(s).

## 1. Introduction

Diffusion modeling (Sohl-Dickstein et al., 2015; Ho et al., 2020) is the state-of-the-art generative modeling framework, producing diverse and high-quality samples across various domains, including but not limited to images (Saharia et al., 2022; BlackForestLabs, 2025), audio (Lemercier et al., 2025), videos (Ho et al., 2022), and proteins (Abramson et al., 2024; Watson et al., 2023). These models define a forward noising process that iteratively corrupts data samples, transforming the data distribution into an easily sampled "noise" distribution, and then learn to reverse the noising process. Diffusion models have been well studied and developed in both continuous (Ho et al., 2020; Song et al., 2020b; 2021) and discrete, categorical (Austin et al., 2021; Hoogeboom et al., 2021; Campbell et al., 2022) data types, including for popular use cases such as images and tokenized text. However, data from biological assays such as whole-genome sequencing, RNA sequencing (including single-cell RNA sequencing; scRNA-seq), ATAC-seq (including single-cell ATAC-seq), and metagenomic read counts are direct measurements of abundance in the form of natural numbers. Like the reals, the natural numbers are an unbounded ordered set. However, the natural numbers are clearly discrete. Given this, both approaches must be adapted to count-based data. One can either (1) relax the natural numbers to the reals and train a continuous diffusion model (Kotelnikov et al., 2023; Jolicoeur-Martineau et al., 2024) or (2) treat each number up to some maximum as an independent class and train a discrete diffusion model. Both of these solutions present potential pitfalls. Training a continuous diffusion model optimizes over the much larger space of real-valued distributions, only to quantize at inference time, which, as we illustrate using a simple toy dataset, can be ineffective. On the other hand, the categorical adaptation ignores the natural ordering of numerical data, and can quickly become computationally expensive as the maximum value increases since a distinct category for each possible value is required.

To address these challenges, we introduce *CountsDiff*, a modern diffusion model that operates on the set of natural numbers $\mathbb{N}_0 := \{0, 1, 2, \dots\}$. CountsDiff builds on the theoretical underpinnings of Blackout Diffusion (Santos

et al., 2023), but, for greater clarity and generality, reparameterizes the forward process in terms of a survival probability $p(t)$, stabilizes the outputs via random rounding, and introduces analogs to the complete toolkit of contemporary diffusion models, including continuous-time training and sampling (Campbell et al., 2022), weighted objectives (Kingma & Gao, 2023), guidance (Dhariwal & Nichol, 2021; Ho & Salimans, 2022; Nisonoff et al., 2025), and churn/remasking (Song et al., 2021; Karras et al., 2022; Wang et al., 2025a). Furthermore, we evaluate CountsDiff across three settings. First, we use synthetic count data to illustrate the limitations of existing diffusion frameworks on $\mathbb{N}_0$ compared to CountsDiff. Next, we test CountsDiff on natural images (CIFAR-10 (Krizhevsky et al., 2009) and CelebA (Liu et al., 2015)) to stress its ability to scale to high-dimensional distributions and examine the relative effects of the design parameters we introduce: noise schedules, loss weighting, and attrition. Finally, we turn to single-cell RNA-seq imputation, benchmarking CountsDiff on fetal and heart cell atlases (Cao et al., 2020; Litviňuková et al., 2020), to demonstrate a natural application for our count-based model. An implementation of CountsDiff and all baselines necessary to reproduce experiments is available at https://github.com/rsoatto/countsdiff.

## 2. Background and notation

### 2.1. Generative latent variable models

Given data $\mathcal{D} = \{x^{(i)}\}_{i=1}^N$ sampled from an unknown distribution $P_{\text{data}}$, the goal of generative modeling is to learn a distribution $P_{\text{gen}}$ to approximate $P_{\text{data}}$, often by minimizing negative log-likelihood (NLL, see Appendix A.1). A common approach to this is latent variable modeling, where one samples from a simple distribution $P_{\text{noise}}$ (e.g. unit Gaussian) and learns a transformation to the data domain.

Diffusion models follow this paradigm by defining a sequence of forward corruption kernels $\{q(x_t \mid x_{t-1})\}_{t=1}^T$ that gradually transform samples from $x_0 \in P_{\text{data}}$ into $x_T \in P_{\text{noise}}$. By composition, this defines a forward process $q(x_{1:T} \mid x_0)$ that maps the data to "noise" e.g., samples from an isotropic Gaussian. Diffusion modeling seeks to approximate the corresponding reverse kernels $q(x_{t-1} \mid x_t)$, which ideally invert the corruption at each step. For instance, as we detail in Appendix A.2, when the support $\mathcal{X} = \mathbb{R}$, *Gaussian diffusion models* (Ho et al., 2020) use Gaussian transitions as forward kernels.

### 2.2. Discrete diffusion models

For categorical data, one can define a diffusion process over a finite support with $|\mathcal{X}| = N$ via forward transition kernel matrices $\{Q_t\}_{t=1}^T$, where $[Q_t]_{i,j} = q(x_t = j \mid x_{t-1} = i)$.

By composition,

$$q(x_t \mid x_0 = i) = e^{(i)} \bar{Q}_t, \quad \bar{Q}_t = Q_1 Q_2 \ldots Q_t,$$

where $e^{(i)}$ is the $i$-th standard basis vector.

This general formulation admits a variety of forward processes that converge to simple $P_{\text{noise}}$'s. A common choice is to choose a simple categorical distribution $\text{Cat}(\mathcal{X}, F)$ with $F \in \Delta^{N-1}$ and define

$$Q_t = (1 - \beta_t)\, \mathbf{I} + \beta_t\, \mathbf{1} F^\top,$$

with schedule $\{\beta_t\}$. This yields marginals with $\bar{Q}_t = \bar{\alpha}_t \mathbf{I} + (1 - \bar{\alpha}_t)\, \mathbf{1} F^\top$, where $\bar{\alpha}_t = \prod_{s=1}^t (1 - \beta_s)$. Two important special cases are uniform discrete diffusion (Hoogeboom et al., 2021) when $F = \frac{1}{N}\mathbf{1}$ and masked diffusion (Austin et al., 2021; Sahoo et al., 2024; Shi et al., 2024) when $F = e^{(\text{mask})}$. In both cases, the model is trained to approximate the reverse kernels $q(x_{t-1} \mid x_t)$ (Austin et al., 2021). We will refer to these models operating on *finite* categorical spaces as "categorical diffusion models" to contrast with CountsDiff, which is also formally discrete diffusion.

These processes extend naturally to continuous time (Shi et al., 2024; Sahoo et al., 2024). As $T \to \infty$, cumulative transition matrices $\bar{Q}(t)$ evolve according to the Kolmogorov forward equations (Norris, 1997):

$$\tfrac{d}{dt}\bar{Q}(t) = \bar{Q}(t)\, R(t), \quad q(x_t \mid x_0 = i) = e^{(i)}\bar{Q}(t). \quad (1)$$

$R(t)$ specifies infinitesimal transition rates via

$$q(x_{t+dt} = j \mid x_t = i) = \delta_{i,j} + R_{i,j}(t)\, dt + o(dt),$$

where $\delta$ is the Kronecker delta and $o(dt)$ represents terms that tend to zero faster than $dt$. This continuous-time formulation via transition rates enables the extension of diffusion principles beyond finite categorical data, see (Benton et al., 2024; Holderrieth et al., 2025).

#### 2.2.1. DISCRETE DIFFUSION ON $\mathbb{N}_0$

To model data on $\mathbb{N}_0$, a natural choice is the family of birth–death processes (Karlin & McGregor, 1957; Feller et al., 1971), in which each state can only increase by one with birth rate: $R_{i,i+1}(t) = \lambda_i(t)$, decrease by one with death rate $R_{i,i-1}(t) = \mu_i(t)$ or stay the same with rate $R_{i,i}(t) = -(\lambda_i(t) + \mu_i(t))$. Explicitly,

$$R_{i,j}(t) = \lambda_i(t)(\delta_{i+1,j} - \delta_{i,j}) + \mu_i(t)(\delta_{i-1,j} - \delta_{i,j}).$$

Santos et al. (2023) restricts this to a pure-death process with $\lambda_i(t) = 0$ and $\mu_i(t) = i$, which is an absorbing-state forward process (e.g. masked categorical diffusion), i.e., where $P_{\text{noise}}$ is a Dirac delta. Solving the Kolmogorov forward (equation 1) yields binomial marginals:

$$q(x_t \mid x_0) = \binom{x_0}{x_t} p(t)^{x_t} (1 - p(t))^{x_0 - x_t}, \quad p(t) = e^{-t}.$$

The corresponding reverse process is a pure-birth process with maximum state $x_0$ with rates

$$\boldsymbol{R}_{i,j}^{(\text{rev})}(t) = (x_0 - i)\frac{\text{e}^{-t}}{1-\text{e}^{-t}}(\delta_{i+1,j} - \delta_{i,j}). \qquad (2)$$

Learning this reverse process amounts to predicting the number of remaining elements $y_t = x_0 - x_t$ given $(x_t, t)$. This can be optimized by minimizing $\sum_{x_0 \in \mathcal{D}} \mathbb{E}_t \left[ (\text{e}^{-t_{k-1}} - \text{e}^{-t_k})(\hat{y}_t - y_t \log \hat{y}_t) \right]$, which is equivalent to minimizing the NLL (Santos et al., 2023).

## 3. Methods

Herein, we formally introduce the CountsDiff framework, defining a forward process parameterized by a $p$-schedule, a weighted loss, a family of reverse processes parameterized by an attrition schedule, predictor-free guidance, and a stochastic rounding algorithm that prevents a failure mode of Blackout Diffusion.

### 3.1. CountsDiff forward process

For our forward process, we consider the following *inhomogeneous* pure-death process

$$\boldsymbol{R}_{i,j}^{(\text{fw})}(t) = i\mu(t)(\delta_{i-1,j} - \delta_{i,j}).^1 \qquad (3)$$

We define a CountsDiff forward process with $p$-*schedule* $p(t)$ as a pure death process with transitions given by equation 3, with $\mu(t)$ such that the process has marginals

$$q(x_t \mid x_0) = \binom{x_0}{x_t} p(t)^{x_t} (1 - p(t))^{x_0 - x_t}, \qquad (4)$$

and conditionals

$$q(x_t \mid x_s) = \binom{x_s}{x_t} \left(\frac{p(t)}{p(s)}\right)^{x_t} \left(1 - \left(\frac{p(t)}{p(s)}\right)\right)^{x_s - x_t}. \qquad (5)$$

We have the following existence proposition, proven in Appendix B.1:

**Proposition 3.1.** *Given* $p : [0,1] \to [0,1]$ *differentiable, monotonically decreasing, and with endpoints* $p(0) = 1$, $p(1) = 0$, *there exists a CountsDiff forward process with* $p$-*schedule* $p(t)$.

This forward process is visualized in Figure 1. Blackout diffusion's forward process is a special case of CountsDiff's; see Appendix B.2. Intuitively, this $p(t)$ controls the rate at which information is destroyed in the noising process. As we show in Appendix B.3, the signal-to-noise ratio of a Gaussian diffusion proccess as a function of its noise schedule $\bar{\alpha}_t$ matches the signal-to-noise ratio of a CountsDiff forward process as a function of its $p$-schedule exactly.

---

[1]Blackout diffusion is the special case with $\mu(t) \equiv 1$ and with time rescaled to $[0, \infty)$.

This correspondence enables direct application of any noise schedule from Gaussian diffusion to CountsDiff; for this work, we propose $p(t) = \cos(\frac{\pi t}{2})^2$ from Nichol & Dhariwal (2021), which also has some theoretical advantages over the schedule in Blackout diffusion (see Appendix B.7).

### 3.2. CountsDiff objective

Our general objective takes the form

$$\mathbb{E}_{t \sim \phi} \left[ w(t)(\hat{y}_t - y_t \log \hat{y}_t) \right], \qquad \hat{y}_t = (\text{NN}_\theta(x_t, t))^+ \quad (6)$$

where $\text{NN}_\theta$ is a neural network, $w(t) > 0$ is a weighting term, and $(a)^+ := \text{softplus}(a) = \log(1 + \text{e}^a)$.

When $w(t) = \frac{-p'(t)}{(1-p(t))\,\phi(t)}$, where $\phi(t)$ is the distribution from which $t$ is sampled, the loss recovers the NLL (see Appendix B.4). Since this objective is minimized pointwise by taking $\hat{y}_t = y$, the minimizer remains unchanged for any $w(t) > 0$; the choice of weight only influences our training dynamics. We refer to Kingma & Gao (2023) for a more in-depth discussion on the weighting of diffusion model losses and their correspondence with importance sampling.

For our cosine $p$-schedule $p(t) = \cos(\frac{\pi t}{2})^2$ we propose a weighting term $w(t) = \frac{\pi}{2}\sin(\pi t)$, which can be derived using two orthogonal heuristics: matching the sigmoid weighting commonly used in Gaussian diffusion, or by matching the form $w(t) = -p'(t)$ in Blackout diffusion. In fact, the choice of $-p'(t)$ also results in equation 6 corresponding to an exact NLL (see prop B.1 in Appendix B.5). For a detailed discussion, see Appendix B.6.

### 3.3. CountsDiff reverse process with attrition

The reverse process (proof in Appendix B.8) corresponding to equation 3,

$$\boldsymbol{R}_{i,j}^{(\text{rev})} = (x_0 - i)\frac{p'(t)}{1-p(t)}(\delta_{i,j} - \delta_{i+1,j}), \qquad (7)$$

generates a monotonic trajectory via a pure-birth process. This monotonicity, analogous the irreversibility of unmasking in masked diffusion models, is undesirable since if a model "overshoots", the error cannot be corrected and can accumulate. Discrete diffusion models overcome this through remasking (Wang et al., 2025a). Similarly, we generalize the reverse process to allow *attrition*, a nonzero death rate compensated with births, yielding a *non-monotonic process* with the same binomial marginals:

**Proposition 3.2** (Reverse step with attrition). *Given* $x_t$, *a* $p$-*schedule* $p(t)$, *and an attrition rate* $\sigma_{t,s} \in [0, \sigma_{t,s}^{\max}]$, *where* $\sigma_{t,s}^{\max} := \min(1, \frac{1-p(s)}{p(t)})$, *let* $\beta_{t,s} = \frac{p(s)-(1-\sigma_{t,s})p(t)}{1-p(t)}$. *Then the following sampling procedure preserves the marginal distribution of* $\text{x}_s$ *defined in equation 4:*

$$x_s = n_t + b_t$$
$$n_t \sim \text{Bin}(x_t, 1 - \sigma_{t,s}), \quad b_t \sim \text{Bin}(x_0 - x_t, \beta_{t,s}) \qquad (8)$$

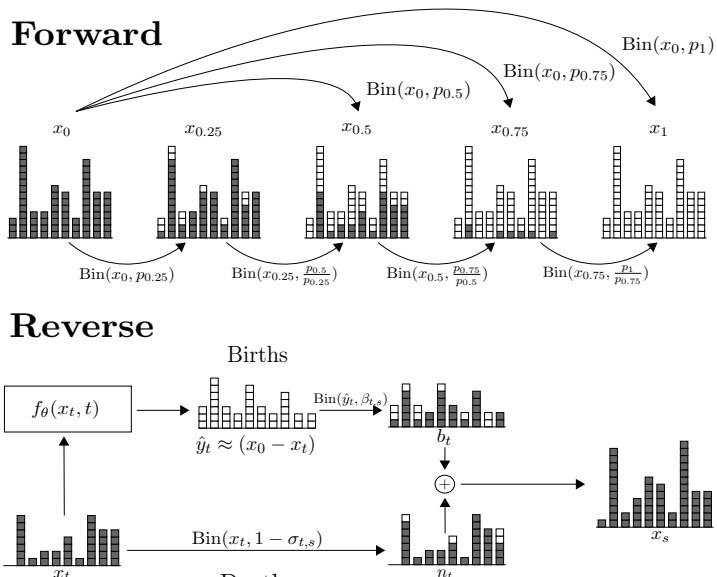

*Figure 1.* Visualization of CountsDiff's forward corruption process (top) and reverse sampling process (bottom). The top diagram depicts the progression of a $p$-schedule, a pure death process. The bottom shows a single step of the generalized, birth-death sampling process.

See Appendix B.9 for a proof of this proposition. Varying this $\sigma_{t,s}$, which is analogous to the churn parameter in Gaussian diffusion (Song et al., 2021; Karras et al., 2022) and remasking in discrete diffusion (Wang et al., 2025a), yields a family of birth-death processes. All elements of this family are valid given a trained model because the training procedure (Algorithm 1 in Appendix C) depends only on the marginals. Then, given an attrition schedule $\sigma$, we can generate samples by iteratively performing equation 8, where $x_0 - x_t$ is replaced by the prediction $\hat{y}$ from a neural network trained to optimize equation 6. This sampling procedure is visualized in Figure 1 and described in Algorithm 2 in Appendix C.

Both $\sigma_{t,s}^{\max}$ and $\beta_{t,s}$ as a function of $\sigma_{t,s}$ have the same form as their analogs in ReMDM (Wang et al., 2025a), providing an interpretation of a birth as an *unmasking* event and a death as a *remasking* event. Thus, we borrow a remasking strategy from ReMDM as a starting point: ReMDM-rescale, which sets $\sigma_{t,s} = \eta_{\mathrm{rescale}} \sigma_{t,s}^{\max}$, where $\eta_{\mathrm{rescale}}$ is a tunable sampling hyperparameter.

### 3.4. Guidance

Classifier-free guidance is a widely used technique in continuous and categorical diffusion models (Dhariwal & Nichol, 2021; Ho & Salimans, 2022), and was recently extended to discrete state spaces in continuous time by Nisonoff et al. (2025); Schiff et al. (2025); Li et al.. We adapt the method of Nisonoff et al. (2025) to enable guidance for diffusion models on the natural numbers. Formally, given a conditional reverse rate $R_{i,j}^{(\mathrm{rev})}(t \mid c)$ for class $c$ and its unconditional

counterpart $R_{i,j}^{(\mathrm{rev})}(t)$, the guided rate $R_{i,j}^{(\mathrm{rev})}(t; \gamma \mid c)$ with strength $\gamma \geq 0$ is defined as

$$R_{i,j}^{(\mathrm{rev})}(t; \gamma \mid c) = R_{i,j}^{(\mathrm{rev})}(t \mid c)^\gamma \, R_{i,j}^{(\mathrm{rev})}(t)^{1-\gamma}.$$

For CountsDiff, this simplifies to

$$\hat{y}^{(\gamma)} = (\hat{y} \mid c)^\gamma \, \hat{y}^{1-\gamma},$$

where guided samples are obtained by substituting $\hat{y}^{(\gamma)}$ directly into the binomial reverse process. As in other diffusion frameworks, we implement predictor-free guidance by training a single neural network that outputs both conditional and unconditional predictions, achieved by randomly zeroing out class embeddings with probability $p_{\mathrm{uncond}}$.

### 3.5. Rounding

At inference time, rounding is necessary to convert the real-valued output of neural networks into predictions $\hat{y} \in \mathbb{N}_0$, as required by the reverse process. Naively rounding to the nearest integer causes mode collapse at 0 when $\hat{y} < 0.5$ (which occurs frequently for near-zero counts). Santos et al. (2023) and Chen & Zhou (2023) consider a scheme based on a Poisson approximation; however, this expression is an unfaithful approximation of the binomial distribution in this small $y$ setting. Instead, we adopt a randomized rounding scheme that preserves the expectation of $\hat{y}$ while keeping exact binomial draws, preventing 0-collapse (appendix E.1.3) in a principled manner.

$$\hat{y}_{\mathrm{clipped}} = \lfloor \hat{y} \rfloor + \xi, \qquad \xi \sim \mathrm{Bernoulli}(\hat{y} - \lfloor \hat{y} \rfloor).$$

### 3.6. Adapting CountsDiff to data imputation

We adapt CountsDiff to imputation using the RePaint algorithm (Lugmayr et al., 2022, Algorithm 1), originally developed for image inpainting. RePaint requires no retraining: after each reverse step during sampling, observed entries are reset to their noised ground-truth values, and only masked entries are resampled. This procedure has been successfully repurposed in other domains (*eg.* Forest Diffusion (Jolicoeur-Martineau et al., 2024)), and we adopt it here for biological count data.

## 4. Experiments

We validate CountsDiff in three experimental settings. We begin with a small toy dataset of counts, comparing it with Gaussian and masked (categorical) diffusion. We follow with experiments on digital images (CIFAR-10 and CelebA (Krizhevsky et al., 2009; Liu et al., 2015)) to show that CountsDiff is capable of modeling complex, high-dimensional distributions, and to probe the relative effect of different $p$-schedules, guidance, and reverse process dynamics in a visually interpretable setting with well-defined benchmarks. Finally, we propose scRNA-seq imputation as a natural real-world use-case for CountsDiff and benchmark it against existing imputation methods.

### 4.1. Simulated counts

To validate CountsDiff's strength in generating counts, we train three simple models on synthetic 10-dimensional sparse count vectors: a Gaussian diffusion model operating in log-space, a (categorical) masked diffusion model (MDLM (Sahoo et al., 2024), which is equivalent to ReMDM (Wang et al., 2025a) with no remasking), and CountsDiff without attrition. Data are generated from negative binomial distributions with multiplicative size factors, yielding $\approx 50\%$ zeros and maximum counts near 50. Each model uses a small multi-layer perceptron (MLP) backbone with matching parameter counts. We evaluate sample quality by computing Maximum Mean Discrepancy (MMD) (Gretton et al., 2012) and sliced Wasserstein-1 distance (SWD), a scalable approximation of the Wasserstein-2 distance, to the ground truth data. SWD is computed using 100 random projections following Bonneel et al. (2015). We also computed the MMD and Wasserstein-1 distance between the true and generated marginal distributions in each dimension, plot distributions of a subset of pairs of dimensions using Kernel Density Estimator (KDE) plots, and compute the variances of each dimension. Full distributional parameters and architecture details are in Appendix D.1.

### 4.2. Natural images

We train CountsDiff on CIFAR-10 and CelebA using U-Net architectures adapted from prior diffusion baselines (Song et al., 2020b), following the hyperparameters of Santos et al. (2023) wherever possible. We report three main experiments:

1. **Guidance.** We implement predictor-free guidance with $p_{\text{uncond}} = 0.1$ and evaluate conditional sampling across guidance scales. We measure Fréchet Inception Score (FID) (Heusel et al., 2017) and Inception Score (IS) (Salimans et al., 2016) and inspect CIFAR-10 samples qualitatively.

2. **Reverse-process dynamics.** We introduce nonzero attrition during sampling and assess its effect on FID/IS, as well as any visual effects on images. To validate robustness, we illustrate this on both CIFAR-10 and CelebA.

3. **Quantitative results.** We compare the FI discrete schedule implied by (Santos et al., 2023), its continuous analog, and our proposed cosine schedule, and report the best-performing combinations of $p$-schedule, guidance, and attrition on 50k CIFAR-10 samples.

### 4.3. Single cell RNA-Seq imputation

We evaluate CountsDiff on three imputation tasks on heart cell (Litviňuková et al., 2020) and fetal cell (Cao et al., 2020) atlases: 50% missing complete at random (MCAR) for fetal cells, a 25% low-biased missing not at random (MNAR) regime (with low counts masked out at a higher rate) for fetal cells, and 50% MCAR for heart cells. See Appendix A.3 for further discussion of missingness and imputation. We preprocess our data to select a subset of genes that are commonly but differentially expressed across cells (see Appendix D.3.1 for preprocessing details). A test set containing masked-out target sites is then passed into CountsDiff and baselines for imputation.

**Baselines:** We compare CountsDiff to a series of baselines. For context, we include naive baselines, which impute without any learning: imputing zeroes, the mean expression of the missing gene in the training set (Mean imputation), or the mean expression of the missing gene for all samples with matching conditions (Conditional Mean). We also compare to specialized methods that are explicitly designed for scRNAseq data or for imputation representing the state-of-the-art applications of a range of modeling frameworks to scRNAseq imputation: a graph-smoothing approach (MAGIC, Van Dijk et al. (2018)), a generative adversarial networks (GAN) based approach (GAIN, Yoon et al. (2018)), a variational autoencoder (VAE) based approach (HI-VAE, Nazabal et al. (2020)) with both Gaussian and Poisson likelihoods, a Gaussian diffusion-based approach (Zhang & Liu (2024), and two masked-autoencoder (MAE) based approaches (xTrimoGene, Gong et al. (2023), scGPT Cui et al. (2024)). Finally, for a more direct comparison of diffusion frameworks, we compare CountsDiff to other diffusion methods, adapted to imputation using the RePaint algorithm: Forest-Diffusion (Jolicoeur-Martineau

et al., 2024), which is based on Gaussian diffusion, Re-masking Masked Diffusion Models (ReMDM) (Wang et al., 2025b), a state-of-the-art, masking-based discrete diffusion model, and Blackout diffusion (Santos et al., 2023), a precursor and special case of CountsDiff. See Appendix D.3.2 for more details.

**Metrics:** CountsDiff and the baselines are evaluated with both pointwise metrics: Root mean-squared error (RMSE), bias, and Spearman's rank correlation, and distributional metrics: Energy Distance (ED), Maximum Mean Discrepancy (MMD) (Gretton et al., 2012), Sliced Wasserstein Distance (SWD) (Bonneel et al., 2015), and scFID (Rizvi et al., 2025), an adaptation of FID for single cell data. Pointwise metrics are computed individually for each sample and averaged over the evaluation set, while distributional metrics are computed on the entire evaluation set. For each metric, we obtain standard error via ten bootstrap resamples (50k data points for the fetus cell atlas and 20k for the heart cell atlas). Implementation details and further discussion of metrics are detailed in Appendix D.3.3.

# 5. Results

## 5.1. Toy example

Both CountsDiff and masked diffusion are easily able to learn the marginals of the toy distribution, with comparably low marginal MMD and Wasserstein-1 distances across all dimensions (Figure 2). Both also perform well on the joint MMD, indicating they capture the dominant modes well.

However, masked diffusion performs poorly on SWD, indicating it may be overfitting to outliers in the training set and is therefore more prone to generating excessive outliers/low-quality "hallucinations". We confirm this by noting that the variance in each dimension of masked diffusion's samples is far greater than the other two models and the real data (see Figure 2 and Appendix E.1). The higher joint-SWD relative to marginal Wasserstein-distance also indicates masked diffusion is less effective in learning the correlations between dimensions; we observe this in the joint KDE plots between the first two dimensions (Figure 8).

Gaussian diffusion was entirely unable to learn these sparse, discrete, ordinal data and suffered from extreme mode collapse. Increasing model capacity and training time and varying learning rate did not affect this result.

## 5.2. Natural images

**Guided Image Generation** performs as expected, enabling class-conditioned sampling for CIFAR-10 (Figure 3). Moderate levels of guidance also improve FID and IS (Table 1).

**Increasing attrition rate has a smoothing effect**: rough or noisy parts of a generated image can be interpreted as

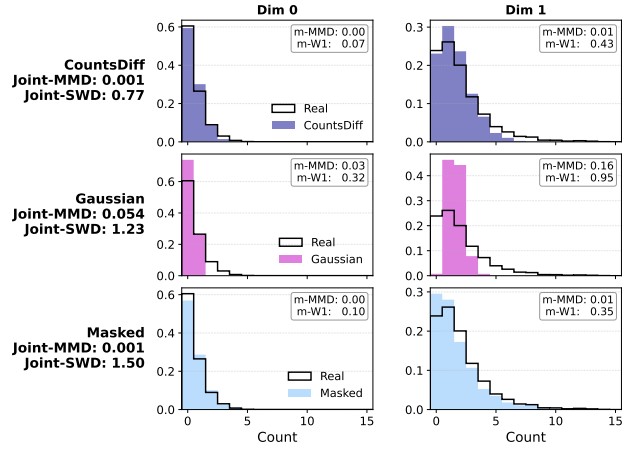

| Variance ($\sigma^2$) – *Closer to True is Better* | | | | | |
|---|---|---|---|---|---|
| **Model** | **Dim 0** | **Dim 1** | **Dim 2** | **Dim 3** | **Dim 4** |
| True *(Target)* | 0.78 | 4.71 | 0.10 | 0.12 | 0.28 |
| **CountsDiff** | **0.55** | **1.99** | **0.07** | **0.08** | **0.21** |
| Gaussian | 0.19 | 0.46 | 0.01 | 0.02 | 0.05 |
| Masked | 3.06 | 9.22 | 1.89 | 2.49 | 7.27 |

*Figure 2.* Histogram of model-generated samples versus ground truth and distributional distance metrics (top); variance statistics (bottom) for a subset of dimensions. Existing diffusion models exhibit failure cases even in a simple toy dataset: Gaussian diffusion suffers from mode collapse, while masked diffusion overfits outliers (inflated variance). Full results for all ten dimensions can be found in Appendix E.1.

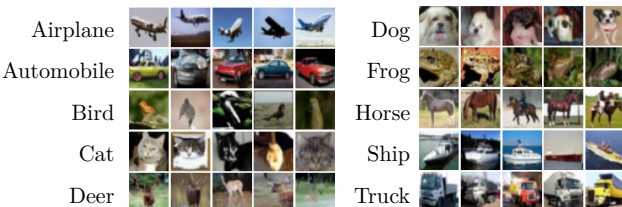

*Figure 3.* 5 images guided by each Cifar10 class sampled from CountsDiff with guidance scale 2.0 and $\eta_{rescale} = 0.005$.

"overshooting" the correct value; allowing for these values to decrement manifests as smoothing. Taken to the extreme, we see dramatic oversmoothing, which results in a complete removal of texture and perspective as $\eta_{rescale} \to 1$. See Figure 4 for CIFAR-10 and Figure 11 for CelebA.

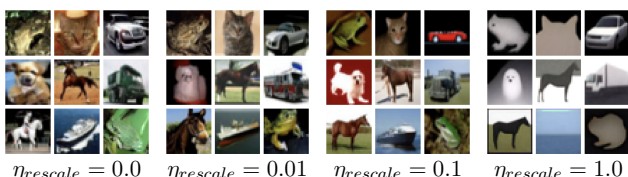

$\eta_{rescale} = 0.0$  $\eta_{rescale} = 0.01$  $\eta_{rescale} = 0.1$  $\eta_{rescale} = 1.0$

*Figure 4.* Nine images drawn from CountsDiff trained on CIFAR-10 with $\eta_{rescale}$ attrition schedule for varying levels of $\eta_{rescale}$

### 5.2.1. QUANTITATIVE RESULTS

We find both moderate guidance and small, nonzero attrition to improve the FID and IS of samples generated by Counts-Diff. Generalizing the Fisher Information (FI) $p$-schedule from Santos et al. (2023) to continuous time improves FID. Across hyperparameters, FI noise schedule generally results in slightly better FID, while the cosine schedule results in slightly better IS, indicating the cosine schedule generates slightly higher fidelity samples at the expense of slightly poorer sample diversity. Notably, even this limited exploration of our extended design space with choices drawn directly from existing diffusion frameworks resulted in significant improvements in sample quality and diversity, underscoring the potential for the elucidated CountsDiff design space to enable rapid, systematic advances in count-based diffusion.

Training curves were also more stable for our cosine noise schedule (see Appendix E.2.1), consistent with the original motivation of this schedule in Gaussian diffusion. For more extensive ablations on $\gamma$ and attrition see Appendix E.2.2, E.2.3. For results on CelebA see Appendix E.3.

*Table 1.* FID and IS of 50k images sampled with from CountsDiff trained on CIFAR-10 for a selected set of sampling hyperparameters. FI Discrete with unconditional generation and no attrition is equivalent to Blackout (Santos et al., 2023)

| $p$-**schedule** | $\gamma$ | $\eta_{\text{rescale}}$ | **FID** ↓ | **IS** ↑ |
|---|---|---|---|---|
| FI Discrete | uncond | none | 5.73 | $9.12 \pm 0.05$ |
| FI Continuous | uncond | none | 5.44 | $9.09 \pm 0.16$ |
| FI Continuous | 1.0 | 0.01 | **5.20** | $9.64 \pm 0.17$ |
| FI Continuous | 2.0 | 0.02 | 9.71 | $9.77 \pm 0.11$ |
| Cosine | uncond | none | 5.76 | $9.29 \pm 0.18$ |
| Cosine | 1.0 | 0.01 | 5.26 | $9.85 \pm 0.08$ |
| Cosine | 2.0 | 0.02 | 11.55 | $\mathbf{9.93} \pm 0.13$ |

### 5.3. scRNA-seq imputation

Our imputation results for the MCAR and MNAR scenarios on the fetus cell atlas (Cao et al., 2020) are shown below in Table 2 and Table 3. Further results on the heart cell atlas (Litviňuková et al., 2020) can be found in Table 10 in Appendix E.4; more metrics can be found in Appendix E.6

CountsDiff (CountsDiff 1-sample) is consistently a top performer across all distributional metrics in MCAR tasks, both in the fetus data (Table 2) and the heart data (Table 10), comparable only to ReMDM (ReMDM 1-sample), a state-of-the-art discrete generative model. The higher RMSE and RMSE standard error, significantly higher SWD in 1-imputation indicates that ReMDM likely exhibits similar over-sampling of outliers to what we observe in the simulated data (Figure 2), though to a less catastrophic extent than standard masked diffusion. This tendency is particularly undesirable in scientific settings, where outliers are precisely the observations

used to infer signal; over-sampling is therefore likely to generate spurious findings and false conclusions. Furthermore, CountsDiff has about half as many parameters (one-fourth in the heart cell atlas task) as ReMDM, due to ReMDM's output layer size depending on the max count.

In the pointwise metrics, CountsDiff is outperformed by some existing metrics, but *this outcome is expected*. Pointwise metrics reward models that predict conditional expectations rather than full distributions; for example, RMSE is minimized by the true conditional mean, while CountsDiff and other generative models sample from an approximate conditional distribution. See Appendix D.3.3 for a more complete discussion. As such, we expect non-generative approaches such as xTrimoGene and MAGIC to perform well; even naive baselines like imputing the conditional mean, which requires no learning whatsoever, are strong. Despite strong point estimate performance, these non-generative methods perform poorly in distributional metrics and less suitable for tasks that require preservation of distributional structure across cells, such as learning gene programs within cell types or gene regulation. Nonetheless, we recognize that pointwise performance is sometimes valuable, and show that taking the mean of just 5 samples from CountsDiff (Counts-Diff 5-sample) improves these metrics. We expect taking more samples too further improve the pointwise metrics, though this makes inference more costly.

In the MNAR task, ScGPT and xTrimoGene are quite strong in all metrics apart from Spearman correlation. However, this apparent advantage can be explained largely by fortuitous miscalibration of these models. While all methods, regardless of architecture, exhibit a shift in bias between the MCAR and low-biased MNAR tasks, the negative bias of scGPT and xTrimoGene, due to their tendency of over-sampling zeros, coincidentally aligns with the low-biased MNAR missingness pattern, resulting in inflated metrics across the board. We note that this is not an indication of robustness to MNAR-tasks: a high-biased MNAR task would amplify scGPT and xTrimoGene's bias. Furthermore, both models perform poorly in Spearman correlation, where they are outperformed by ReMDM, Blackout diffusion, Counts-Diff, and the simple conditional mean baseline, indicating ineffectiveness at differentiating genes. Among the generative models, CountsDiff has the strongest performance in all metrics but Spearman correlation and scFID.

Guidance and moderate levels of attrition improve sample quality, matching trends in imaging data. The cosine-schedule results in slightly better performance and yields more significant improvements from guidance/attrition than the Blackout noise schedule; see appendix E.5.

With further optimization in $p$-scheduling, loss weighting, and attrition scheduling beyond the scope of this work, there is substantial room for empirical improvement.

*Table 2.* Benchmarking on scRNA-seq imputation, human fetus cell atlas with 50% MCAR. Mean (standard error). Methods are grouped into three categories: naive baseline (top), imputation/scRNAseq-specific (middle), and general generative (bottom). Best performance in each category for each metric is bolded, and second best is italicized.

| | Pointwise | | | Distributional | | | |
|---|---|---|---|---|---|---|---|
| Method | Spearman↑ | RMSE↓ | Bias | ED↓ | log(scFID)↓ | log(MMD)↓ | SWD↓ |
| Zero imputation | N/A | 1.91(0.01) | *-1.31(0.00)* | 1.44(0.00) | -2.35(0.00) | -4.08(0.00) | 0.17(0.00) |
| Mean imputation | *0.17(0.00)* | *1.31(0.04)* | **0.00(0.00)** | *0.17(0.00)* | *-5.01(0.01)* | -7.03(0.00) | *0.12(0.01)* |
| Conditional Mean | **0.20(0.00)** | **1.12(0.01)** | **0.00(0.00)** | **0.15(0.00)** | **-6.23(0.01)** | **-7.49(0.01)** | **0.08(0.00)** |
| MAGIC | **0.21(0.00)** | 1.88(0.01) | -1.31(0.00) | 1.44(0.00) | -2.35(0.00) | -4.07(0.00) | 0.17(0.00) |
| scIDPMs, 1-sample | 0.08(0.00) | 2.25(0.02) | 0.86(0.00) | 0.68(0.00) | -3.04(0.01) | -3.39(0.00) | 0.20(0.00) |
| scIDPMs, 5-sample | 0.10(0.00) | 1.89(0.01) | 0.86(0.00) | 0.88(0.00) | -2.92(0.01) | -3.65(0.00) | 0.18(0.00) |
| GAIN | 0.04(0.00) | 1.87(0.02) | -1.27(0.00) | 1.44(0.00) | -2.34(0.00) | -4.07(0.00) | 0.17(0.00) |
| HI-VAE (Poisson) | 0.15(0.00) | 1.27(0.02) | *0.02(0.00)* | **0.11(0.00)** | *-6.26(0.01)* | -6.68(0.00) | 0.11(0.00) |
| HI-VAE (Gaussian) | 0.14(0.00) | 1.27(0.03) | **-0.01(0.00)** | *0.12(0.00)* | -5.78(0.01) | **-6.93(0.01)** | 0.10(0.01) |
| scGPT (scratch) | 0.17(0.00) | **1.05(0.02)** | -0.20(0.00) | 0.25(0.00) | -5.95(0.01) | -6.20(0.00) | *0.09(0.01)* |
| scGPT (pretrained) | 0.13(0.00) | 1.19(0.02) | -0.44(0.00) | 0.51(0.00) | -4.63(0.00) | -5.33(0.00) | 0.11(0.00) |
| xTrimoGene | *0.18(0.00)* | *1.08(0.02)* | -0.11(0.00) | 0.37(0.00) | **-6.71(0.02)** | *-6.75(0.01)* | **0.08(0.00)** |
| Forest-Diffusion | 0.03(0.00) | 27.18(0.06) | 8.41(0.01) | 2.18(0.00) | -1.54(0.00) | -0.80(0.00) | 4.94(0.01) |
| ReMDM, 1-sample | *0.11(0.00)* | 1.66(0.06) | **0.00(0.00)** | **0.01(0.00)** | *-8.98(0.05)* | *-11.69(0.03)* | 0.12(0.01) |
| ReMDM, 5-sample | **0.12(0.00)** | *1.20(0.03)* | **0.00(0.00)** | 0.28(0.00) | -8.44(0.04) | -8.76(0.01) | *0.08(0.01)* |
| Blackout | *0.11(0.00)* | 1.39(0.02) | *0.03(0.00)* | *0.02(0.00)* | -7.37(0.01) | -10.78(0.02) | **0.07(0.01)** |
| **CountsDiff (Ours),** 1-sample | 0.09(0.00) | 1.42(0.03) | **0.00(0.00)** | **0.01(0.00)** | **-9.18(0.05)** | **-11.94(0.02)** | *0.08(0.01)* |
| **CountsDiff (Ours),** 5-sample | **0.12(0.00)** | **1.17(0.03)** | **0.00(0.00)** | 0.30(0.00) | -7.60(0.03) | -8.64(0.01) | *0.08(0.01)* |

# 6. Related work

## 6.1. Generative Models

Our work is most closely related to Blackout Diffusion (Santos et al., 2023), which can be interpreted as a special case of CountsDiff with no guidance, fixed $p$-schedule and loss weighting, and no sampling with attrition. Santos et al. (2023) prove the NLL objective and validity of the reverse process only in this special case.

JUMP (Chen & Zhou, 2023) models positive, real-valued data by projecting it into counts ($z_0 \sim \text{poisson}(\lambda x_0)$), then noises through a binomial thinning of $z_0$, resulting in a similar noising process, parametrized by $\alpha_t$. Their loss objective, derived from the ELBO, resembles equation 6 but with a different predictive target and constant loss weighting. For natively count-based data, Chen & Zhou (2023) also propose Binomial-JUMP, which can be interpreted as another special case of CountsDiff with constant weights, no guidance, and no attrition, and using the Poisson sampling scheme mentioned in section 3.5. JUMP's primary advantage lies in its ability to handle continuous non-negative data, which is outside the scope of the present work. The underlying noising and denoising resembles Blackout Diffusion and has a similarly limited design space. JUMP and CountsDiff are complementary approaches, and CountsDiff's improvements on modeling counts (extended design space, continuous-time formulation, and exact loss) can be readily extended to non-negative reals using JUMP's Poisson data randomization trick (Appendix A.5).

We would also like to point the reader towards relevant works in Gaussian and categorical diffusion that can help inform design choices of the CountsDiff design space. In particular, (Karras et al., 2022) and (Kingma & Gao, 2023) explore noise schedules and loss weighting in Gaussian diffusion; Wang et al. (2025a) introduces remasking for masked discrete diffusion, which is analogous to attrition; and Sahoo et al. (2025) introduces a framework to bridge Gaussian diffusion with discrete diffusion in order to more easily transfer design choices.

## 6.2. scRNA-seq imputation

Due to the high sparsity and missingness of scRNA-seq data (as discussed in Appendix A.3), imputation of scRNA-seq data is an important but challenging problem. Methods based on various modeling paradigms, which we benchmark against, have been proposed to address this issue; see the full list in section 4.3.

ScDiffusion (Luo et al., 2024) and Squidiff (He et al., 2026) are also diffusion models trained on scRNAseq. However, both are latent diffusion models that output whole transcriptomes, meaning they cannot be as easily adapted to imputation tasks. Recently, some single-cell foundation models such as xTrimoGene (Gong et al., 2023) and scGPT (Cui et al., 2024), which aim to yield informative representations of cells from their expression profiles, have used masked expression prediction (which is effectively imputation) as the training objective. As a result, these models can also be adapted to impute scRNAseq values.

*Table 3.* Benchmarking on scRNA-seq imputation, human fetus cell atlas with 25% low-biased missingness (MNAR). Mean (standard error). Methods are grouped into three categories: naive baseline (top), imputation/scRNAseq-specific (middle), and general generative (bottom). Best performance in each category for each metric is bolded, and second best is italicized.

| | Pointwise | | | Distributional | | | |
|---|---|---|---|---|---|---|---|
| **Method** | Spearman↑ | RMSE↓ | Bias | ED↓ | log(scFID)↓ | log(MMD)↓ | SWD↓ |
| Zero imputation | N/A | 1.00(0.00) | -0.99(0.00) | *1.41(0.00)* | *-5.29(0.01)* | -6.64(0.00) | 0.03(0.00) |
| Mean imputation | *0.26(0.00)* | *0.51(0.00)* | *0.32(0.00)* | **0.12(0.00)** | -4.81(0.01) | *-7.99(0.01)* | *0.02(0.00)* |
| Conditional Mean | **0.36(0.00)** | **0.47(0.00)** | **0.28(0.00)** | **0.12(0.00)** | **-5.82(0.01)** | **-8.69(0.01)** | **0.01(0.00)** |
| MAGIC | 0.09(0.02) | 1.00(0.00) | -0.99(0.00) | 1.41(0.00) | -5.28(0.00) | -6.63(0.00) | 0.03(0.00) |
| scIDPMs, 1-sample | 0.03(0.00) | 2.22(0.01) | 1.20(0.00) | 0.98(0.00) | -3.61(0.00) | -4.20(0.00) | 0.14(0.00) |
| scIDPMs, 5-sample | 0.03(0.00) | 1.77(0.00) | 1.20(0.00) | 1.17(0.00) | -3.52(0.00) | -4.41(0.00) | 0.11(0.00) |
| GAIN | 0.02(0.00) | 0.91(0.00) | -0.91(0.00) | 1.40(0.00) | -5.42(0.01) | -6.64(0.00) | 0.03(0.00) |
| HI-VAE (Poisson) | 0.03(0.00) | 0.59(0.00) | 0.39(0.00) | 0.22(0.00) | -5.24(0.00) | -8.23(0.00) | 0.02(0.00) |
| HI-VAE (Gaussian) | 0.02(0.00) | 0.73(0.00) | 0.45(0.00) | 0.37(0.00) | -4.83(0.00) | -7.55(0.00) | 0.03(0.00) |
| scGPT (scratch) | **0.33(0.00)** | *0.28(0.00)* | *0.12(0.00)* | *0.07(0.00)* | *-7.79(0.01)* | **-11.92(0.01)** | *0.01(0.00)* |
| scGPT (pretrained) | *0.31(0.00)* | **0.24(0.00)** | **-0.09(0.00)** | **0.04(0.00)** | **-8.71(0.02)** | *-11.19(0.01)* | **0.00(0.00)** |
| xTrimoGene | 0.29(0.00) | 0.38(0.00) | 0.25(0.00) | 0.12(0.00) | -6.93(0.01) | -11.06(0.00) | *0.01(0.00)* |
| Forest-Diffusion | 0.03(0.00) | 30.53(0.27) | 9.80(0.03) | 2.36(0.00) | **-7.26(0.02)** | -4.13(0.00) | 0.88(0.01) |
| ReMDM, 1-sample | **0.47(0.00)** | 1.01(0.05) | *0.31(0.00)* | 0.31(0.00) | *-6.72(0.01)* | -7.56(0.01) | 0.05(0.01) |
| ReMDM, 5-sample | 0.38(0.00) | *0.66(0.01)* | *0.31(0.00)* | *0.28(0.00)* | -6.61(0.00) | *-8.64(0.00)* | **0.02(0.00)** |
| Blackout | *0.45(0.00)* | 0.88(0.01) | **0.30(0.00)** | 0.31(0.00) | -6.32(0.01) | -7.73(0.00) | *0.03(0.00)* |
| **CountsDiff (Ours),** 1-sample | 0.42(0.00) | 0.87(0.00) | **0.30(0.00)** | 0.30(0.00) | -6.41(0.01) | -7.62(0.00) | *0.03(0.00)* |
| **CountsDiff (Ours),** 5-sample | 0.35(0.00) | **0.58(0.00)** | **0.30(0.00)** | **0.26(0.00)** | -6.24(0.01) | **-8.78(0.00)** | **0.02(0.00)** |

# 7. Discussion

In this paper, we introduced CountsDiff, a diffusion framework designed to handle discrete ordinal data using birth/death processes as the noising and denoising mechanisms. Our main contribution is an elucidated and deconvolved design space, where each design parameter, noise schedule, loss weighting, reverse-process modifications, and guidance, has a direct and interpretable analogue in modern continuous and categorical diffusion models. This framing both extends and clarifies the design space of Blackout Diffusion (Santos et al., 2023). Concretely, we unlocked continuous-time training, reparameterized the model with a more intuitive $p$-schedule, introduced a principled loss weighting, derived attrition as the counterpart to churn/remasking, and incorporate classifier-free guidance.

We proposed principled starting points for each of these new design parameters, demonstrating how our unified design space enables seamless transfer across diffusion families. Through experiments on a range of applications, from image generation to scRNA-seq imputation, we demonstrated that this initial instantiation of CountsDiff matches the performance of a state-of-the-art discrete diffusion model while avoiding a key failure case in count-based regimes. Further, CountsDiff also outperformed specialized scRNA-seq imputation methods across multiple metrics.

Our work yields promising results for scRNA-seq imputation, and we expect further hyperparameter optimization and task-specific adaptations to unlock the potential of the CountsDiff framework for this and other large-scale biology applications, such as ATAC-Seq imputation, perturbation effect prediction, and single-cell foundation modeling.

# 8. Future Work and Limitations

Below we outline limitations and directions of future work for CountsDiff in both the generative modeling and the biological application spaces:

**Generative Modeling**: CountsDiff requires a moderate number of sampling steps, matching Denoising Diffusion Implicit Models (DDIM)-style (Song et al., 2020a) Gaussian diffusion in sampling speed, but is significantly slower than recent few- and one-step samplers such as Consistency Models (Song et al., 2023), Mean Flows (Geng et al., 2025), and Categorical Flow Maps (Roos et al., 2026)). Extending such ideas to CountsDiff is an important avenue to explore. We also only consider one transferred candidate per introduced design choice (noise schedule, loss weighting, attrition). Further optimization, particularly attrition, which is the least studied of the three, is likely to yield meaningful modeling gains.

**scRNAseq/Biological applications**: For our scRNA-seq imputation task specifically, our evaluation is limited to a subset of highly variable genes, in line with related methods in scRNA-seq. We also do not explore pre-training on multiple scRNA-seq datasets or task/domain-specific adaptations to the training objective or model architecture, such as those developed for Gaussian diffusion-based models in scIDPMs (Zhang & Liu, 2024) and Squidiff (He et al., 2024).

## Impact Statement

In addition to advancing the field of machine learning, the goal of the work presented in this paper is to improve generative modeling for count-valued biological data. By enabling practical diffusion modeling on the natural numbers, this work aims to support more faithful generation and imputation of biological count data in order to facilitate improved understanding of underlying biological systems.

However, the use of generative AI in biological settings carries important considerations: generated or imputed values misinterpreted as direct measurements may lead to incorrect scientific conclusions if model limitations are not carefully accounted for. These risks are not unique to our approach, instead applying broadly to generative modeling in biology.

## Acknowledgements

This work was originally proposed by Valentin De Bortoli and would not have been possible without his mathematical insight and guidance throughout.

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

# A. Extended background

## A.1. Kullback-Leibler Divergence equivalence with Negative Log Likelihood

Because the data distribution is unknown and unknowable, requiring that $P_{\text{gen}} \approx P_{\text{data}}$ is ill-defined. This problem is commonly addressed by approximating the objective using Monte-Carlo sampling. For example, if the error is quantified by the Kullback-Leibler divergence (Kullback & Leibler, 1951):

$$D_{\text{KL}}(P_{\text{data}}|P_{\text{gen}}) = \mathbb{E}\left[\log \frac{P_{\text{data}}(x)}{P_{\text{gen}}(x)}\right],$$

we can approximate it via

$$\mathbb{E}\left[\log \frac{P_{\text{data}}(x)}{P_{\text{gen}}(x)}\right] \approx \frac{1}{|D|} \sum_{x \in D} \log(P_{\text{data}}(x)) - \log(P_{\text{gen}}(x)),$$

which is minimized with respect to $P_{\text{gen}}$ when the Negative Log Likelihood (NLL) of the data with respect to $P_{\text{gen}}$, $NLL = -\sum_{x \in D} \log(P_{\text{gen}}(x))$, is minimized, since the first term is constant with respect to $P_{\text{gen}}$.

## A.2. Gaussian diffusion models

Gaussian diffusion models are diffusion models where the forward kernels are Gaussian transitions

$$q(\mathbf{x}_t|\mathbf{x}_{t-1}) = \mathcal{N}(\sqrt{1 - \beta_t}\mathbf{x}_{t-1}, \; \beta_t\mathbf{I}),$$

where the sequence $\{\beta_t\}_{t=1}^T$ is a monotonic *variance schedule* with $\beta_0 = 0$ and $\beta_T = 1$. The Gaussian diffusion forward process can be rewritten in the following closed form:

$$\mathbf{x}_t = \sqrt{\bar{\alpha}_t}\mathbf{x}_0 + \sqrt{1 - \bar{\alpha}_t}\epsilon, \quad \epsilon \sim \mathcal{N}(0, \mathbf{I}), \tag{9}$$

with $\bar{\alpha}_t = \Pi_{s=1}^t(1 - \beta_t)$, resulting in $P_{\text{noise}}(z|\mathbf{x}) = q(\mathbf{x}_T|\mathbf{x}_0) = \mathcal{N}(0, \mathbf{I})$. Optimizing $p_\theta(\mathbf{x}_{t-1}|\mathbf{x}_t)$ to minimize the NLL of the observed $\mathbf{x}_{t-1}$ can be reduced to predicting the added noise $\epsilon$, the original signal $\mathbf{x}_0$, or some hybrid of the two. Though these objectives are equivalent up to reweighting of the loss objective, their empirical performance can vary (Kingma & Gao, 2023). This process and the corresponding objectives can also naturally be extended to the continuous time domain by taking the limit as $T \to \infty$ and $\Delta t \to 0$. The forward and reverse processes become stochastic differential equations (SDEs) (Song et al., 2020b), but the marginals can still be written in closed form, similar to equation 9:

$$\mathbf{x}_t = \sqrt{\alpha(t)}\mathbf{x}_0 + \sqrt{1 - \alpha(t)}\epsilon, \quad \epsilon \sim \mathcal{N}(0, \mathbf{I}), \tag{10}$$

where $\alpha(t)$ is commonly referred to as a *noise schedule*, and is a continuous monotonic function of $t$. Although the resulting training objectives are nearly identical, the continuous extension allows for more flexibility at the generation stage: one can sample using numerical stochastic differential equations / ordinary differential equations solvers (Hartman, 2002) (ODE), or if they choose to discretize the reverse SDE, they are no longer bound to a specific number of time steps. Due to the similarity in the training objectives and the Gaussianity of the marginals $q(x_t|x_0)$ in these continuous extensions, we will consider them a subclass of Gaussian diffusion models, namely *continuous-time* (as opposed to *discrete-time*) Gaussian diffusion models.

## A.3. Data missingness

The standard theory for missing data depends on the notion of "missing at random" (MAR, (Rubin, 1976; Seaman et al., 2013; Schafer, 1997)). There are three types of missingness mechanisms: (i) *missing completely at random* (MCAR), when the process determining missingness is assumed to be independent of (the values of) the variables; (ii) *missing at random* (MAR), when the missingness mechanism depends only on the observed variables, and (iii) *missing not at random* (MNAR), when the missingness depends on both observed and unobserved (missing) variables (Little & Rubin, 2020).

More formally, suppose there is some ground truth vector of counts $\mathbf{x}^{\text{true}}$, and some binary mask o, and we observe $\mathbf{x}^{\text{obs}} = \mathbf{x}^{\text{true}} \cdot o$.

The MCAR assumption is simply that each position $o_i$ of o is distributed according to

$$o_i \overset{i.i.d}{\sim} \text{Bernoulli}(1 - d)$$

for some dropout probability $d$.

The MAR assumption is that $o \perp\!\!\!\perp x^{\text{true}}$, and MNAR covers all remaining cases. In particular, we are interested in the setting where for each $o_i$, we have

$$o_i \sim \text{Bernoulli}(1 - d_i),$$

where $d_i$ is larger for smaller values of $x_i$. This form of MNAR is relevant for scRNA-seq imputation tasks, as missingness can be induced by low read counts (Qiu, 2020).

## A.4. Imputation

Missingness can be addressed with either *single* or *multiple imputation*. Multiple imputation (MI) (Rubin, 1987) is a widely studied method (Sterne et al., 2009; Hayati Rezvan et al., 2015) that uses the distribution of observed data to estimate a set of likely values for missing data. By contrast, single imputation generates only a single point.

Several methods have been developed to handle data missingness. Early methods include complete case analysis, in which samples with missing data are simply removed from the dataset, and mean imputation, where missing values are filled with the per-variable mean across all (or a subset satisfying a specific condition) of the observed data points. Early machine learning methods include random forest imputation (Hastie et al., 2009; Stekhoven & Bühlmann, 2012; Shah et al., 2014), which recursively splits the data via a predictor from known samples to estimate unobserved values.

More recently, generative models, including variational autoencoders (VAEs) (Kingma & Welling, 2014), generative adversarial networks (GANs) (Goodfellow et al., 2014), and diffusion models (Sohl-Dickstein et al., 2015; Ho et al., 2020), have been explored for data imputation. These methods rely on the principle that generative models are capable of learning the underlying data-generating distributions. In Burda et al. (2015); Nazabal et al. (2020); Roskams-Hieter et al. (2023), the authors extend VAEs, initially replacing missing values with zeros and training a neural network to predict these values. Similarly, Li et al. (2019); Yoon et al. (2018); Xu et al. (2019) use GANs, setting up a game between a generator $G$ that generates *both* observed and imputed values and a discriminator $D$ that decides whether a particular data point was imputed or not. Diffusion models can be designed for imputation (Tashiro et al., 2021; Alcaraz & Strodthoff, 2022) or be adapted for imputation through algorithms such as RePaint (Lugmayr et al., 2022), which passes the noised ground truth at each time step of denoising to fix the imputation target sites (Jolicoeur-Martineau et al., 2024), or methods such as DiffPuter (Zhang et al., 2025), which uses the Expectation-Maximization algorithm to guide a diffusion model to fill in missingness.

### A.4.1. MULTIPLE IMPUTATION

As described in Huque et al. (2018), there are two general approaches for imputing incomplete data: (a) joint modeling (JM) (Hughes et al., 2014) and (b) fully conditional specification (FCS) or multiple imputation using chained equations (MICE) (Van Buuren et al., 2006; Van Buuren & Groothuis-Oudshoorn, 2011). In JM a multivariate distribution of the missing data is sampled using Markov chain Monte Carlo (MCMC) (Gilks, 1995). In cases where this multivariate distribution is suitable for the data, this method is appealing. FCS employs a set of conditional densities, one for each partially observed variable, and performs imputation in a variable-by-variable manner. This is done by starting from an initial imputation and then imputing by iterating a few times (usually 10-20) over the conditional densities.

## A.5. Extending CountsDiff to continuous data

While our work focuses on modeling count data (which allows us to preserve the exact NLL), the Poisson-based data randomization trick from Chen & Zhou (2023) can be combined with CountsDiff via the following procedure:

1. Nonnegative inputs $x_0$ are mapped to latent counts via $z_0 \sim \text{Poisson}(\lambda x_0)$, $\lambda \geq 1$.

2. CountsDiff is applied directly to model the distribution of $z_0$

3. Generated samples are divided by $\lambda$ at inference time. Chen & Zhou (2023) show that the original distribution of $x_0$ is recovered as $\lambda \to \infty$.

This simple procedure would extend the benefits of CountsDiff (guidance, schedule design, loss weighting, and attrition) to JUMP and therefore provide a principled way to model continuous, non-negative domains. The continuous time formulation also in principle unlocks fast ODE/SDE solvers (Ren et al., 2025) for JUMP. We note that this would not be a *strict* generalization of JUMP, as the model would operate directly in the latent counts space, as opposed to combining the Poisson randomization with binomial thickening/thinning at each forward and reverse step, and the training target would be $z_0 - z_t$ as opposed to $x_0$ in JUMP.

# B. Proofs and derivations

### B.1. Proof of Proposition 3.1

We restate the proposition here for clarity:

Given $p : [0, 1] \to [0, 1]$ differentiable, monotonically decreasing, and with endpoints $p(0) = 1$, $p(1) = 0$, there exists a CountsDiff forward process with $p$-schedule $p(t)$.

*Proof.* Fix $x_0 \in \mathbb{N}_0$ and consider $x_0$ independent, two-state (0/1) time-inhomogeneous Markov processes

$$\mathrm{y}_t^{(m)} \in \{0, 1\}, \qquad m = 1, \ldots, x_0,$$

each with transition rates $\boldsymbol{R}^{(\mathrm{fw})}(t) = \begin{bmatrix} 0 & 0 \\ \mu(t) & -\mu(t) \end{bmatrix}$. Let $\mathrm{x}_t = \sum_{m=1}^{x_0} \mathrm{y}_t^{(m)}$ be the number of ones at time $t$; then $\mathrm{x}_t$ is governed by the pure-death process in equation 3.

For a single particle, consider the survival probability $\mathbb{P}(\mathrm{y}_t = 1 \mid \mathrm{y}_0 = 1)$. The Kolmogorov forward equation (1) then yields

$$\frac{d}{dt}\mathbb{P}(\mathrm{y}_t = 1 \mid \mathrm{y}_0 = 1) = -\mu(t)\mathbb{P}(\mathrm{y}_t = 1 \mid \mathrm{y}_0 = 1), \qquad P(\mathrm{y}_0 = 1 \mid \mathrm{y}_0 = 1) = 1,$$

which is a separable ODE with solution $\mathbb{P}(\mathrm{y}_t = 1 \mid \mathrm{y}_0 = 1) = \exp\left(-\int_0^t \mu(u)\, du\right)$. Let $\mu(t) := -\frac{p'(t)}{p(t)}$ for $t \in (0, 1)$, and $\mu(0) = \mu(1) = 0$. This choice is always valid since $p$ is differentiable and non-increasing, and is positive at $t \in (0, 1)$. Furthermore, the choice of value of $\mu$ at $t \in \{0, 1\}$ does not influence the dynamics of the process, as the Lebesgue integral $\exp\left(-\int_0^t \mu(u)\, du\right)$ is invariant to function values on sets of measure zero. Thus we can by inserting our ansatz derive

$$\mathbb{P}(\mathrm{y}_t = 1 \mid \mathrm{y}_0 = 1) = \exp\left(\int_0^t \frac{p'(u)}{p(u)}\, du\right) = \exp\left(\log p(t) - \log p(0)\right) = \frac{p(t)}{p(0)} = p(t).$$

Thus $\mathrm{y}_t^{(m)} \sim \mathrm{Bernoulli}(p(t))$ i.i.d. across $m$.

Note that as $t \to 1$ implies $p(t) \to 0$ and hence $\mu(t) \to +\infty$. However, this divergence is not a problem as $\mu(t)$ only interacts with the process through $\exp\left(-\int_0^t \mu(u)\, du\right)$, where the divergence implies a rapid convergence to zero.

Since the sum of i.i.d Bernoulli's is Binomial, we have

$$\mathbb{P}(\mathrm{x}_t = x_t \mid \mathrm{x}_0 = x_0) = \binom{x_0}{x_t} p(t)^{x_t} (1 - p(t))^{x_0 - x_t},$$

which is equation 4.

For $0 \le s < t \le 1$, we have by Bayes theorem

$$\mathbb{P}(\mathrm{y}_t = 1 \mid \mathrm{y}_s = 1) = \frac{\mathbb{P}(\mathrm{y}_s = 1 \mid \mathrm{y}_t = 1)\mathbb{P}(\mathrm{y}_t = 1 \mid \mathrm{y}_0 = 1)}{\mathbb{P}(\mathrm{y}_s = 1 \mid \mathrm{y}_0 = 1)} = \frac{1 \cdot p(t)}{p(s)} = \frac{p(t)}{p(s)},$$

where $\mathbb{P}(\mathrm{y}_s = 1 \mid \mathrm{y}_t = 1) = 1$ because a pure-death process alive at $t$ has to have been alive at $s$. Conditioning on $X_s = x_s$, the $x_s$ survivors evolve independently, so

$$X_t \mid X_s = x_s \sim \mathrm{Bin}\left(x_s, \frac{p(t)}{p(s)}\right),$$

which gives the binomial conditionals in equation 5. $\qquad\square$

## B.2. Reparameterization of Blackout diffusion time schedule as a $p$-schedule

Santos et al. (2023) work with a pure death-noising process with constant individual death rate $\mu \equiv 1$. As such, in order to adjust their corruption process for a constant decay in Fisher Information (FI), they define the following time schedule:

$$t_k = -\log\left[\sigma\left(\text{Logit}(1 - e^{-t_T}) + \frac{k-1}{T-1}[\text{Logit}(e^{-t_T}) - \text{Logit}(1 - e^{-t_T})]\right)\right], \quad k = 1, \ldots T. \tag{11}$$

However, note that the $\log$ term undoes the $\exp$ in their $p$-schedule, so this schedule is effectively a workaround to allow for a non-exponential $p$-schedule

$$p_k = \sigma\left(\text{Logit}(1 - e^{-t_T}) + \frac{k-1}{T-1}[\text{Logit}(e^{-t_T}) - \text{Logit}(1 - e^{-t_T})]\right), \quad k = 1, \ldots T.$$

However, using the time-inhomogeneous pure-death process in equation 3, we can bypass the time schedule trick, so that configurability lies directly in $p$ space, which we find to be a more intuitive schedule that better matches existing diffusion literature. For greater consistency with continuous-time diffusion frameworks, we have also rescaled the time steps to be on the closed unit interval. As a concrete example, the $p$-schedule from blackout diffusion becomes

$$p(t) = \sigma\left(\text{Logit}(1 - p_{min}) + t \cdot [\text{Logit}(p_{min}) - \text{Logit}(1 - p_{min})]\right), \quad t \in [0, 1],$$

where $p_{min}$ defines the values at the endpoints and is set to $e^{-15}$. Note that despite the extension of $p(t)$ to a continuous function, sampling $T$ uniform values from 0 to 1 exactly recovers the original formulation.

## B.3. Equivalence of $p$-schedule and Gaussian diffusion noise schedule

Our $p$-schedule is inspired by the cosine noise schedule in (Nichol & Dhariwal, 2021), where their noise schedule takes the following form:

$$\bar{\alpha}(t) = \cos\left(\frac{t/T + s}{1 + s}\frac{\pi}{2}\right)^2.$$

Taking the most canonical form, with $s = 0$ and $T = 1$, we have

$$\bar{\alpha}(t) = \cos\left(\frac{t\pi}{2}\right)^2.$$

To find the CountsDiff analog, we match the signal-to-noise ratio (SNR) of the cosine noise schedule in Gaussian diffusion with the SNR of the pure death process and solve for $p(t)$. In Gaussian diffusion, this takes the form

$$\text{SNR}(t) = \frac{\bar{\alpha}(t)}{1 - \bar{\alpha}(t)} = \frac{\cos^2(\pi t/2)}{\sin^2(\pi t/2)}.$$

In our pure-death process with $p$-schedule $p(t)$, the SNR can be expressed as $\frac{p(t)}{1-p(t)}$. Using the same signal to noise definition $\text{SNR}(t) = \frac{\mathbb{E}[x_t]^2}{\text{Var}(x_t)}$ as for the gaussian case. Thus for the independent Bernoullis underlying our pure-death process, we have

$$\text{SNR}(t) = \frac{p(t)^2}{p(t)(1 - p(t))} = \frac{p(t)}{1 - p(t)}.$$

Clearly then, $\bar{\alpha}_t$ is analogous to $p(t)$, so choosing $p(t) = \cos\left(\frac{t\pi}{2}\right)^2$ is a sensible choice. A similar exercise can be done for any $\bar{\alpha}$ schedule in Gaussian Diffusion, yielding a CountsDiff equivalent.

## B.4. Deriving Training objective; Proof of proposition B.1

Santos et al. (2023) derives a continuous time loss function by taking the Kullback-Leibler divergence between Bernoulli distributions corresponding to instantaneous transitions of the ground truth reverse process $R_{i,i+1}^{(\text{rev})}\Delta t$, and the reverse

process induced by the model predictions $\kappa_\theta(t)\Delta t$, at time $t$, where $\Delta t$ is an infinitesimal time differential. This corresponds to the negative log-likelihood and in our notation takes the form

$$\text{NLL}(t) = \boldsymbol{R}^{(\text{rev})}_{x_t, x_t+1}\Delta t \log \frac{\boldsymbol{R}^{(\text{rev})}_{x_t, x_t+1}\Delta t}{\kappa_\theta(t)\Delta t} + (1 - \boldsymbol{R}^{(\text{rev})}_{x_t, x_t+1}\Delta t)\log \frac{1 - \boldsymbol{R}^{(\text{rev})}_{x_t, x_t+1}\Delta t}{1 - \kappa_\theta(t)\Delta t}.$$

We simplify by splitting the logarithm of the fraction in the second term, and Taylor expanding, yielding

$$-(1 - \boldsymbol{R}^{(\text{rev})}_{x_t, x_t+1}\Delta t)\log(1 - \kappa_\theta(t)\Delta t) = \kappa_\theta(t)\Delta t + \mathcal{O}(\Delta t^2).$$

Collecting terms that do not depend on our model parameters $\theta$ into the "constant" $C(t)$, and omitting the higher order terms of $\Delta t$ gives us the representation

$$\text{NLL}(t) = \Delta t\left(\kappa_\theta(t) - \boldsymbol{R}^{(\text{rev})}_{x_t, x_t+1}\log \kappa_\theta(t)\right) + C(t).$$

For the full negative log-likelihood, we take the integral over all times and get

$$\int_0^1 \left(\kappa_\theta(t) - \boldsymbol{R}^{(\text{rev})}_{x_t, x_t+1}\log \kappa_\theta(t)\right)dt + C,$$

where C is the integral of all the $\theta$-independent terms, which is omitted in the sequel. Now, we can multiply and divide by any probability density function $\phi : [0,1] \to [0,1]$ over $t$

$$\int_0^1 \frac{1}{\phi(t)}\left(\kappa_\theta(t) - \boldsymbol{R}^{(\text{rev})}_{x_t, x_t+1}\log \kappa_\theta(t)\right)\phi(t)dt,$$

which allows us to approximate this integral with the usual one-sample Monte Carlo estimate with $t \sim \phi(t)$, resulting in the objective

$$\frac{1}{\phi(t)}\left(\kappa_\theta(t) - \boldsymbol{R}^{(\text{rev})}_{x_t, x_t+1}\log \kappa_\theta(t)\right), \quad t \sim \phi(t).$$

Notice from equation 7, that, given $t$ and $x_t$, only unknown element of the reverse rate is $(x_0 - x_t) =: y_t$. Consequently, we train a neural network $\text{NN}_\theta(x_t, t)$ to output $\hat{y}(t, x_t) \approx y_t$.

Then, we have $\kappa_\theta(t) = \hat{y}(t, x_t)\frac{p'(t)}{1-p(t)}$, which together with equation 7 yields the objective

$$-\frac{p'(s)}{\phi(t)(1 - p(s))}\left(\hat{y}(x_t, t) - (x_0 - x_t)\log\left(-\hat{y}(x_t, x_t)\frac{p'(s)}{1 - p(s)}\right)\right).$$

Finally, dropping the $\hat{y}$-independent term, we get the objective

$$w(t)\left(\hat{y}(x_t, t) - (x_0 - x_t)\log(\hat{y}(x_t, t))\right), \quad w(t) = -\frac{p'(s)}{\phi(t)(1 - p(s))},$$

## B.5. NLL for chosen weighting function

**Proposition B.1.** *Let $p : [0,1] \to (0,1)$ be a continuously differentiable, strictly decreasing $p$-schedule of a CountsDiff forward process $C_p$. Define the training weight*
$$w(t) = -p'(t),$$

*and consider the Countsdiff objective, restated below*

$$\mathcal{L}(\theta) = \mathbb{E}_{t\sim\text{Unif}(0,1)}\left[w(t)\left(\hat{y}_t - y_t\log\hat{y}_t\right)\right], \quad y_t = x_0 - x_t.$$

*Then the continuous-time negative log-likelihood of the CountsDiff process $C_q$ with schedule*

$$q(t) = 1 - \exp\left(p(1) - p(t)\right),$$

*coincides with $\mathcal{L}(\theta)$ up to a $\theta$-independent scaling and additive constant.*

*Moreover, there exists $F : [0, 1] \to [0, 1]$ (the CDF of a density $\phi$) such that*

$$p(u) = q\big(F^{-1}(u)\big), \quad \text{for all } u \in [0, 1].$$

*Consequently, for a uniform grid $0 = u_0 < \cdots < u_K = 1$, running the forward or reverse sampler of $C_p$ at times $\{u_k\}$ is equivalent to running the sampler of $C_q$ at the warped grid $\{t_k\}$ with $t_k = F^{-1}(u_k)$.*

*Proof.* To prove the statement, we note that

$$\frac{q'(t)}{1 - q(t)} = \frac{\exp(p(1) - p(t))p'(t)}{\exp(p(1) - p(t))} = p'(t)$$

Therefore, we have

$$\mathbb{E}_{t \sim \mathrm{Unif}(0,1)} \big[-p'(t)\left(\hat{y}_t - y_t \log \hat{y}_t\right)\big] = \mathbb{E}_{t \sim \mathrm{Unif}(0,1)} \left[-\frac{q'(t)}{1 - q(t)}\left(\hat{y}_t - y_t \log \hat{y}_t\right)\right],$$

and for any valid importance sampling density $\phi$, this is equal to

$$\mathbb{E}_{t \sim \phi} \left[-\frac{q'(t)}{\phi(t)(1 - q(t))}\left(\hat{y}_t - y_t \log \hat{y}_t\right)\right],$$

which is the exact negative log-likelihood for $C_q$. Then, to show the final part of the proposition it suffices to take $\phi(t) \propto \frac{d}{dt}(p^{-1}(q(t)))$

$\square$

### B.6. Motivating weighting function

Our first method of motivating is from the weighting term in blackout diffusion, $w(t_k) = (t_k - t_{k-1})e^{-t_k}$. As we take the limit $t_{k-1} \to t_k$, we approach our continuous case, and we get the weight $w(t) = |p'(t)dt|$, where $p(t) = e^{-t}$.

In our case, we have a uniform time schedule, $dt$ is constant, so we simply reweight by $w(t) = |p'(t)| = \frac{\pi}{2} \sin(\pi t)$. This is an intuitive choice, since time derivative of $p(t)$ can be thought of (informally) as the rate of information decrease, and a steeper decrease corresponds to a more difficult task to undo. Thus, it is sensible to weight by the magnitude $p'(t)$.

An alternate way to motivate this weighting using the sigmoid weighting function commonly used in Gaussian diffusion (Kingma et al., 2021) . When the training task is $\epsilon$-prediction, the sigmoid weight, defined as a function of the log-SNR $\ell(t)$, is $w(\ell) = \sigma(b - \ell)$, where $\sigma(x) = \frac{1}{1+e^{-x}}$ is the sigmoid function and $b$ is a chosen constant. When the training task is $x_0$-prediction (which is equivalent to $\epsilon$-prediction with additional reweighting), the sigmoid weighting $\hat{w}(\ell) = e^b \sigma(\ell - b)$. Since the log SNR takes the form $\ell(t) = \ln(\frac{p(t)}{1-p(t)})$, plugging in $\ell$ and $b = 0$ into the two sigmoid weightings above, we get that

$$w(\ell(t)) = \sigma\left(-\ln\left(\frac{p(t)}{1 - p(t)}\right)\right) = \frac{1}{1 + \frac{p(t)}{1-p(t)}} = 1 - p(t),$$

and

$$\hat{w}(\ell(t)) = \sigma\left(\ln\left(\frac{p(t)}{1 - p(t)}\right)\right) = \frac{1}{1 + \frac{1-p(t)}{p(t)}} = p(t).$$

Heuristically, $(x_0 - x_t)$ is an interpolation of predicting the initial state $x_0$ and predicting the step-wise additive noise $\epsilon$. In

fact, taking a log-space interpolation between $w$ and $\hat{w}$:

$$w(\ell)^{1/2}\hat{w}(\ell)^{1/2} = \sqrt{p(t)(1-p(t))}$$
$$= \sqrt{\cos^2(\pi t/2)\sin^2(\pi t/2)}$$
$$= \cos(\pi t/2)\sin(\pi t/2)$$
$$= \frac{1}{2}\sin(\pi t),$$

matching our $w(t)$ up to a constant factor of $\pi$.

### B.7. Comparing proposed $p$-schedule and weighting with Blackout Diffusion

As was the case with early linear noise schedules in Gaussian Diffusion, the exponential $p$-schedule described in Blackout Diffusion has potentially undesirable properties $0$ and $1$, where $p(t)$ is almost completely flat. The cosine schedule, on

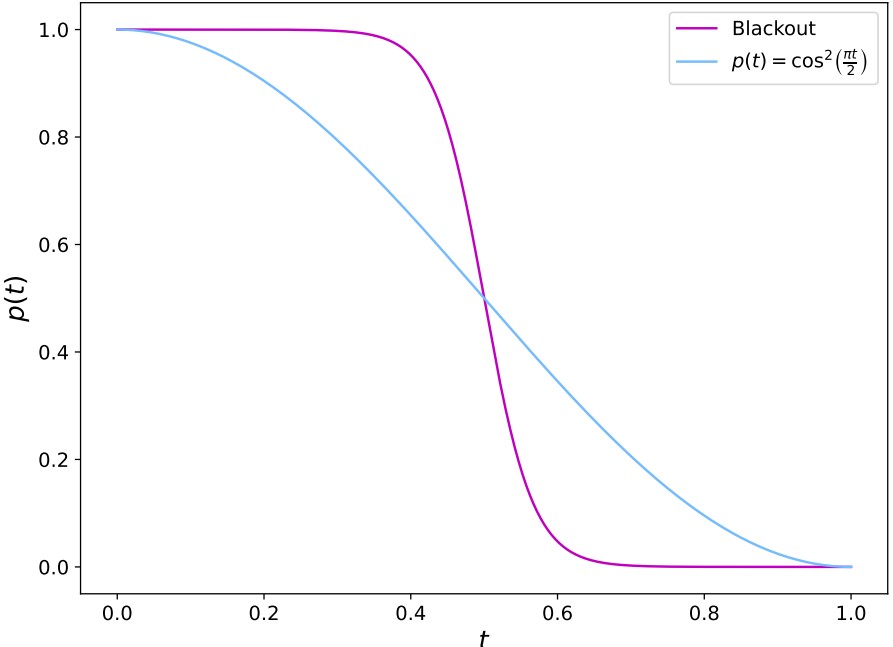

*Figure 5.* Converted $p$-schedule from Blackout Diffusion (see B.2) versus cosine $p$-schedule

the other hand, decreases more gradually (see Figure 5). As a result, the corresponding weighting function for Blackout diffusion puts substantially more emphasis on time steps near $0.5$, and close to no emphasis on those near the endpoints, effectively reducing the batch size, while our proposed weighting attributes a non-negligible weight to nearly all time points (Figure 6). These properties of the $p$-schedules and corresponding weighting schedules explain the improvement in training stability shown in Figure 10a, and may be a factor in the more stable inception scores of samples generated by CountsDiff when trained with the cosine $p$-schedule.

### B.8. Reverse process derivation

We here construct the form of the rates $\boldsymbol{R}_{i,j}^{(\text{rev})}(t)$ for the reverse process. Given the form of the binomial marginals $q(x_t \mid x_0)$ in equation 4, we can construct the reverse rate matrix by equating the forward and reverse rates between states $i$ and $i+1$

$$\boldsymbol{R}_{i,i+1}^{(\text{rev})}(s)q(x_s = i|x_0) = \boldsymbol{R}_{i+1,i}^{(\text{fw})}(t)q(x_s = i+1 \mid x_0).$$

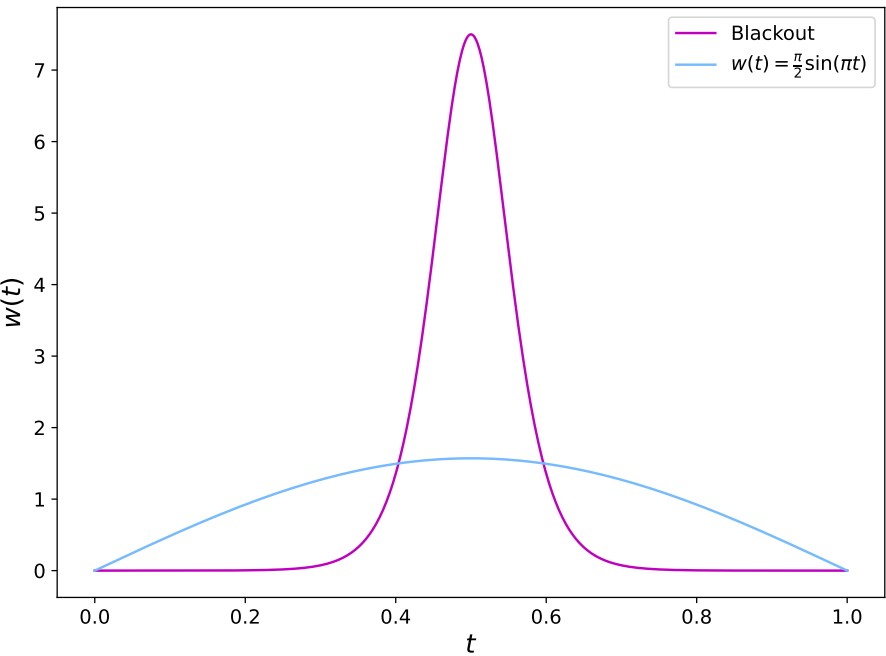

*Figure 6.* Weights from Blackout Diffusion versus proposed $p$-schedule

We then have the rate of an instantaneous transition from $i$ to $i+1$ as

$$
\begin{aligned}
\boldsymbol{R}^{(\mathrm{rev})}_{i,i+1}(s) &= \boldsymbol{R}^{(\mathrm{fw})}_{i+1,i}(t)\frac{q(x_s = i+1 \mid x_0)}{q(x_s = i \mid x_0)} \\
&= (i+1)\mu(s)\frac{q(x_s = i+1 \mid x_0)}{q(x_s = i \mid x_0)} \\
&= (i+1)\mu(s)\frac{\binom{x_0}{i+1}p(s)^{i+1}(1-p(s))^{x_0-(i+1)}}{\binom{x_0}{i}p(s)^i(1-p(s))^{x_0-i}} \\
&= (i+1)\mu(s)\frac{x_0!}{(x_0-(i+1))!(i+1)!}\frac{(x_0-i)!i!}{x_0!}\frac{p(s)}{1-p(s)} \\
&= (i+1)\mu(s)\frac{(x_0-i)}{i+1}\frac{p(s)}{1-p(s)} \\
&= (x_0-i)\mu(s)\frac{p(s)}{1-p(s)} \\
&= -(x_0-i)\frac{p'(s)}{1-p(s)},
\end{aligned}
$$

where in the final step we have inserted the explicit solution of $\mu(s) = -\frac{d}{ds}p(s)/p(s)$ expressed as a function of $p(t)$. This is an application of Bayes' theorem, but a more theoretical operator algebra-based treatment yields an analogous result in Appendix A of Santos et al. (2023). Since the reverse process is a pure birth process, the only allowed instantaneous transfers are between $i$ and $i+1$, and $i$ staying in state. Thus $\boldsymbol{R}^{(\mathrm{rev})}_{i,i}(s) = -\boldsymbol{R}^{(\mathrm{rev})}_{i,i+1}(s)$, $\boldsymbol{R}^{(\mathrm{rev})}_{i,j}(s) = 0$ otherwise. This yields the full formulation in equation 2.

## B.9. Proof of Proposition 3.2

We restate the proposition for clarity:

Given $x_t$, a $p$-schedule $p(t)$, and an attrition rate $\sigma_{t,s} \in [0, \sigma_{t,s}^{\max}]$, where $\sigma_{t,s}^{\max} := \min\left(1, \frac{1-p(s)}{p(t)}\right)$, let

$$\beta_{t,s} = \frac{p(s) - (1 - \sigma_{t,s})p(t)}{1 - p(t)}$$

Then the following sampling procedure preserves the marginal distribution of $x_s$ according to equation 4:

$$x_s = n_t + b_t, \qquad n_t \sim \text{Bin}(x_t, 1 - \sigma_{t,s}), \quad b_t \sim \text{Bin}(x_0 - x_t, \beta_{t,s}),$$

*Proof.* In order to prove the proposition, we need to show that if

$$q(x_t \mid x_0) = \binom{x_0}{x_t} p(t)^{x_t} (1 - p(t))^{x_0 - x_t},$$

then we have for the $x_s$ as sampled in the proposition statement that

$$q(x_s \mid x_0) = \binom{x_0}{x_s} p(s)^{x_s} (1 - p(s))^{x_0 - x_s}$$

As with B.1, we will model $x_t$ and $x_s$ as the sum of $x_0$ independent, two-state Markov processes. Then, the sampling procedure proposed in equation 8 is equivalent to

$$\mathbb{P}(y_s^{(m)} = 1 | y_t^{(m)} = 1) = 1 - \sigma_{t,s}, \qquad \mathbb{P}(y_s^{(m)} = 1 | y_t^{(m)} = 0) = \beta_{t,s}.$$

Then, at time $t$, since $\mathbb{P}(y_t^{(m)} = 1 | y_0 = 1) = p_t$, we have

$$\begin{aligned}
\mathbb{P}(y_s^{(m)} = 1 | y_0 = 1) &= (1 - \sigma_{t,s})p(t) + \beta_{t,s}(1 - p(t)) \\
&= (1 - \sigma_{t,s})p(t) + p(s) - (1 - \sigma_{t,s})p(t) \\
&= p(s),
\end{aligned}$$

where for the second equality we have inserted the form of $\beta_{t,s} = \frac{p(s) - (1 - \sigma_{t,s})p(t)}{1 - p(t)}$ from the proposition statement. Thus we can conclude that $x_s$ has the marginal binomial distribution we set out to prove.

To determine the range of validity for $\sigma_{t,s}$, we test the edge cases

$$\beta_{t,s} \leq 1 \implies \sigma_{t,s} \leq \frac{1 - p(s)}{p(t)}$$

$$\sigma_{t,s} \geq 0$$

One can easily check that the lower bound on $\beta_{t,s}$ does not impose an additional lower bound on $\sigma_{t,s}$.

Thus with our assumed attrition rate $\sigma_{t,s} \in [0, \sigma_{t,s}^{\max}]$, where $\sigma_{t,s}^{\max} := \min(1, \frac{1-p(s)}{p(t)})$, validity for $\beta_{t,s}, \sigma_{t,s}$ is guaranteed. $\qquad\square$

## C. Algorithms

The training algorithm, Algorithm 1, largely aligns with that of Blackout Diffusion. The sampling algorithm, including our contributions, is outlined in Algorithm 2.

---

**Algorithm 1** Training CountsDiff

---

**while** not converged **do**
    Draw $x_0 \sim \mathcal{D}$ from the training set
    Sample $t \sim \mathrm{Unif}([0,1])$
    Sample $x_t \sim \mathrm{Bin}(x_0, p(t))$ element-wise
    Predict: $\hat{y}_t \leftarrow \mathrm{NN}_\theta(x_t, t)$
    Compute: $y_t \leftarrow x_0 - x_t$
    Compute: $l_\theta \leftarrow \mathcal{L}(\theta; y_t, \hat{y}_t)$ (6)
    Take a gradient step on $\nabla_\theta l$
**end while**

---

**Algorithm 2** Generating from CountsDiff

---

**Input:** number of timesteps $T$; class $c$, guidance strength $\gamma$, attrition schedule $\sigma$
**Output:** $\hat{x}_0$   $x_1 = \vec{0}$
**for** $t_k \in \mathrm{linspace}([1,0], T)$ **do**
    $t \leftarrow t_k$
    $s \leftarrow t_{k-1}$
    $p(t) \leftarrow \cos^2(\pi t/2)$
    $p(s) \leftarrow \cos^2(\pi s/2)$
    $\beta_{t,s} \leftarrow \frac{p(s) - (1-\sigma_{t,s})\, p(t)}{1 - p(t)}$
    $\hat{y}_{uncond} \leftarrow NN_\theta(x_t, p(t))^+$
    $\hat{y}_{cond} \leftarrow NN_\theta(x_t, p(t); c)^+$
    $\hat{y} \leftarrow (\hat{y}_{cond})^\gamma (\hat{y}_{uncond})^{(1-\gamma)}$
    $\hat{y}_{clipped} \leftarrow \mathrm{random\_round}(\hat{y})$ (see 3.5);
    Sample $b_t \sim \mathrm{Bin}(\hat{y}_{clipped}, \beta_{t,s})$
    Sample $n_t \sim \mathrm{Bin}(x_t, 1 - \sigma_{t,s})$   $x_s \leftarrow b_t + n_t$
**end for**

---

## D. Experimental settings

### D.1. Simulated counts settings

Each of the $d = 10$ dimensions is sampled from a negative binomial distribution with parameters $\mu \sim$ log-uniform$(0.05, 0.5)$ and $\theta \sim$ log-uniform$(0.2, 5.0)$, which represent the mean and dispersion of the negative binomial, selected log-uniformly from $[0.05, 0.5]$ and $[0.2, 5.0]$ respectively. Each sample is then multiplied by a size factor $s \sim$ log-normal$(0, 0.6)$ that breaks the independence between dimensions. The parameters were chosen such that the data was sparse ($> 50\%$ zeros) and the max count was sufficiently large ($\approx 50$). The Gaussian diffusion algorithm is DDPM with a cosine noise schedule (Dhariwal & Nichol, 2021), trained on log-normalized ($y = \log(1 + x)$) counts. Log-normalization is a common pre-processing technique used for biological counts data before downstream analysis. Since absolute errors in log-space correspond to relative errors in count space, this normalization makes the MSE loss in Gaussian diffusion more sensible for the task at hand, where relative errors are more interesting (predicting 99 instead of 100 should not be penalized as much as predicting 1 instead of 2).The discrete diffusion algorithm is taken from Sahoo et al. (2025) with the linear mutual information interpolating schedule recommended in Austin et al. (2021). CountsDiff uses zero death-rate sampling, no guidance, and the continuous, time-inhomogeneous parametrization of the Blackout Diffusion noise schedule.

All three methods were trained with a 1-layer multilayer perceptron. Gaussian Diffusion and CountsDiff have 48-dimensional hidden layers, and masked diffusion has a 4-dimensional hidden layer to approximately match the total model weights of the other two models, since the output dimension of masked diffusion is the number of classes, which is $\max(X) + 1 \approx 50$.

All models were trained until convergence, at approximately 4000 gradient steps, $T = 200$ steps were used at sample time. 4000 samples were generated from each model and the joint MMD and SWD and marginal MMD and Wasserstein distance to 4000 samples generated from the ground truth distribution is computed. MMD was computed with the Radial Basis Function (RBF) (Buhmann, 2000) kernel with parameter $\gamma = 1$.

## D.2. Natural Images

Because Blackout Diffusion did not release model weights, we elected to re-implement their method using the Unet2D package from Diffusers Huggingface library. The model and training hyperparameters were matched as closely as possible to the default CIFAR-10 hyperparameters in (Song et al., 2020b), which Blackout Diffusion uses as its base model. We used most of the same hyperparameters for CelebA, except we were forced to reduce the batch size to 100, consistent with (Santos et al., 2023) due to memory constraints. We also slightly adjusted the U-Net architecture so that attention is performed at the $16 \times 16$ resolution.

Consistent with Blackout Diffusion, CIFAR-10 models were trained until evaluation metrics stopped improving, or 1 million steps, whichever came first. Similar to Blackout Diffusion, unconditional models stopped improving after approximately 300k gradient steps, while conditional models continued to improve until nearly 1 million steps.

We train CelebA for 1.3 million gradient steps, as is implemented in Blackout diffusion, though we expect conditional models to benefit from additional training.

See code for exact training configs.

## D.3. scRNA-seq Imputation

### D.3.1. SCRNA-SEQ DATA PREPROCESSING

Given the sparsity and dimensionality of scRNA-seq data, we first filter out genes that are rarely expressed across cells. Then, we select the top 1000 (fetus) and 500 (heart) genes, sorted by coefficient of variance. We follow Algorithm 1 from (Zhang & Liu, 2024) to identify missingness sites for each sample. We used an 80/10/10 train/validation/test split. CountsDiff and all baselines that require training were trained only on the training set. Any methods that required hyperparameter tuning were tuned on the validation set, and all evaluations were done on the test set.

### D.3.2. SCRNA-SEQ IMPUTATION BASELINES

To capture a diversity of the existing scRNA-seq imputation methodologies, we compare CountsDiff to MAGIC (Van Dijk et al., 2018), GAIN (Yoon et al., 2018), HI-VAE (Nazabal et al., 2020), scIDPMs (Zhang & Liu, 2024), ForestDiffusion (Jolicoeur-Martineau et al., 2024), and ReMDM (Wang et al., 2025a), along with naive baselines of zero-imputation, mean imputation, and conditional mean imputation (conditioned on sample covariates). All code implementations for scRNA-seq baselines were trained on the same data splits as CountsDiff. All hyperparameter tuning is on the same validation set as CountsDiff. Default training parameters for models were used where possible, though in some cases slight modifications were made to prevent catastrophic failure.

GAIN is an adaptation of GANs for imputation that trains an imputation generator and a discriminator to discriminate imputed versus real points. We adapt training loss to permit fully masked samples. We train for 10,000 iterations, consistent with the default in the original manuscript. At test time, we provide the entire masked-out test set (and corresponding missingness mask) in one shot, enabling GAIN to run normalization over the entire dataset during imputation. Hint matrices were constructed as in the original implementation, with a probability $h = 0.9$ of providing a hint at each locus in the mask. We remove the final sigmoid activation, since otherwise the heavy sparsity of the data results in saturation of gradients and GAIN performing zero-imputation. Instead, we clamp values at 0 at inference time to ensure no negative counts are outputted. We also provide a PyTorch re-implementation of GAIN for compatibility with the rest of the libary, which we use for the final reported results.

HI-VAE is a VAE-based method that factorizes the likelihood for imputation, using poisson likelihoods for counts and gaussian likelihood for continuous data. Since HI-VAE only accepts $\mathbb{N}$ for count-based imputation, we alter the model to run on zero counts by incrementing data by one prior to training/imputation and decrementing afterwards. While this enables HI-VAE to model zeros in the data, it slightly alters the count distribution compared to an alternative model with explicit modeling of zeros. We find that both training and validation losses of HI-VAE converge well within 100 epochs, and as such

we train for only 100 epochs (compared to 2000 in the original paper). We note that both of our datasets are 1-2 orders of magnitude larger than those in the original paper, so a comparable number of training iterations area performed. We provide a PyTorch re-implementation of HI-VAE as well.

MAGIC is a graph-based data-diffusion method for data denoising, but is commonly applied to scRNAseq imputation. It is highly hyperparameter dependent, so we ran a hyperparameter sweep on a validation set unseen by CountsDiff to pick the optimal K-nearest Neighbor (KNN) parameter, optimizing over scFID values. At evaluation time, we provide the MAGIC with the entire train and test set to construct the neighbor graph and run our evaluation on the imputed sites for the test set cells only.

scIDPMs is a gaussian diffusion-based model designd for imputation with an imputation-specific training task: models are conditioned on the observed samples and trained to conditionally generate the missing values. We train scIDPMs for 150 hours, which reaches the 65 and 120th epoch for fetus and heart data accordingly out of the default 500 epochs due to time and GPU constraints; this is already an order of magnitude longer than any of the other baselines. Since scIDPMs is a generative model, we consider both single imputation and multiple (5) imputation. While the original scIDPM implementation used multiple imputation, taking the median value item-wise over 100 imputations, this is also prohibitively expensive. We modify this behavior for a fair comparison between generative models in single imputation (CountsDiff, ForestDiffusion, and ReMDM).

Forest Diffusion is an adaptation of gaussian diffusion models for the generation of tabular data, trained using gradient-boosted trees, which is found to improve performance relative to comparable gradient-based approaches. However, lack of GPU acceleration means it takes several days to train and perform imputations, so we report only single imputation.

Remasking Diffusion Models (ReMDM) is a state-of-the-art masked diffusion model, trained with cross-entropy loss with implementing remasking to avoid the failure-to-remask limitation of masked diffusion models. We train ReMDM for 100k steps, the same as CountsDiff, and just as with CountsDiff, we hyperparameter sweep on number of sampling steps, remasking rate, and guidance scale.

### D.3.3. DISCUSSION OF SCRNA-SEQ IMPUTATION METRICS

We note that evaluating scRNA-Seq imputation methods is an open problem beyond the scope of this work, and individual metrics often do not tell the full story. To address this, we have included a breadth of metrics, both pointwise and distributional, that quantify different aspects of sample quality. Spearman correlation is a common metric for evaluating imputation that measures the preservation of the relative ordering of the genes sorted by expression. This metric is particularly relevant in downstream tasks where identifying the most highly differentially expressed genes, and not the degree of differential expression, is relevant. RMSE is an error metric that is sensitive to outliers and therefore tends to be higher for models that are more likely to predict unrealistic outliers. Moreover, RMSE is minimized by the true conditional mean of a distribution, and thus the conditional sample mean used in our experiments is expected to perform well on this metric. Bias is the mean difference between imputed values and real values, and is important to consider in scenarios where data is missing not at random. For example, a model that is negatively biased might show deceptively high performance when low counts are more probable to be missing than high counts.

While it is infeasible to encapsulate the high-dimensional nature of scRNA-seq data in a single number, we evaluate the models on a suite of distributional metrics. Energy Distance (ED), Maximum Mean Discrepancy (MMD) (Gretton et al., 2012), Sliced Wasserstein Distance (SWD) (Bonneel et al., 2015) are generic distributional metrics chosen to measure whether the empirical distribution of an imputation method's samples resembles the true empirical distribution. To complement these general measures, we utilize scFID (Rizvi et al., 2025), a domain-specific adaptation of FID (Heusel et al., 2017) for single-cell data. Note that scFID is dependent on underlying pre-trained embeddings used, for our choice of implementation see D.3.4.

### D.3.4. IMPLEMENTATION OF SCRNA-SEQ METRICS

scRNA-seq transcriptome embeddings were obtained using a pre-trained *Homo sapiens* SCVI model (Lopez et al., 2018) (version **2024-02-12**) from the CZI CELLxGENE Discover platform. SCVI was chosen as the embedding model because it takes raw count data as input. This version was trained on the CELLxGENE Human Census data (release 2023-12-15) and implemented using the scvi-tools package (Gayoso et al., 2022). To score imputations, the scFID score is calculated between the full ground truth expression profiles and the full expression profiles, with the targets replaced by the model predictions.

ED is computed on the flattened 1 dimensional pool of all imputed target values using energy distance function in scipy. MMD is computed on the full cell by gene matrix with an RBF kernel across 5 bandwidth scales, $(0.24\times, 0.5\times, 1\times, 2\times, 4\times)$ the heuristic bandwidth proposed by Gretton et al. (2012). SWD is also computed on the full cell by gene matrix by projecting both the real and imputed matrices onto 1000 random directions, sorts the projections, and computes the RMSE of sorted differences. This is an approximation of the Wasserstein-2 distance in this joint gene space.

# E. Additional experiments

## E.1. Simulated Data

### E.1.1. Remaining dimensions for simulated experiments

We consistently observe that CountsDiff and masked diffusion are comparable in joint MMD and marginal metrics, but CountsDiff consistently outperforms masked diffusion on joint-SWD. See Figure 7. CountsDiff also consistently maintains variance closer to (albeit lower than) the real data, whereas Gaussian diffusion collapses, and masked diffusion has much higher variance, indicating the presence of excessive outliers. See Table 4.

*Table 4.* Variance of generated samples per dimension. CountsDiff (Ours) consistently maintains variance closer to the Real Data, whereas Gaussian Diffusion collapses (near-zero variance) and masked diffusion suffers from extreme outliers (high variance).

| Dimension | Real Data (Ground Truth) | CountsDiff (Ours) | Gaussian | Masked |
|---|---|---|---|---|
| Dim 0 | 0.78 | **0.55** | 0.19 | 3.06 |
| Dim 1 | 4.71 | **1.99** | 0.46 | 9.22 |
| Dim 2 | 0.10 | **0.07** | 0.01 | 1.89 |
| Dim 3 | 0.12 | **0.08** | 0.02 | 2.49 |
| Dim 4 | 0.28 | **0.21** | 0.05 | 7.27 |
| Dim 5 | 1.97 | **1.10** | 0.24 | 4.78 |
| Dim 6 | 0.84 | **0.44** | 0.24 | 5.19 |
| Dim 7 | 16.09 | 4.50 | 1.04 | **19.83** |
| Dim 8 | 0.62 | **0.46** | 0.17 | 11.63 |
| Dim 9 | 1.77 | **0.96** | 0.22 | 4.09 |

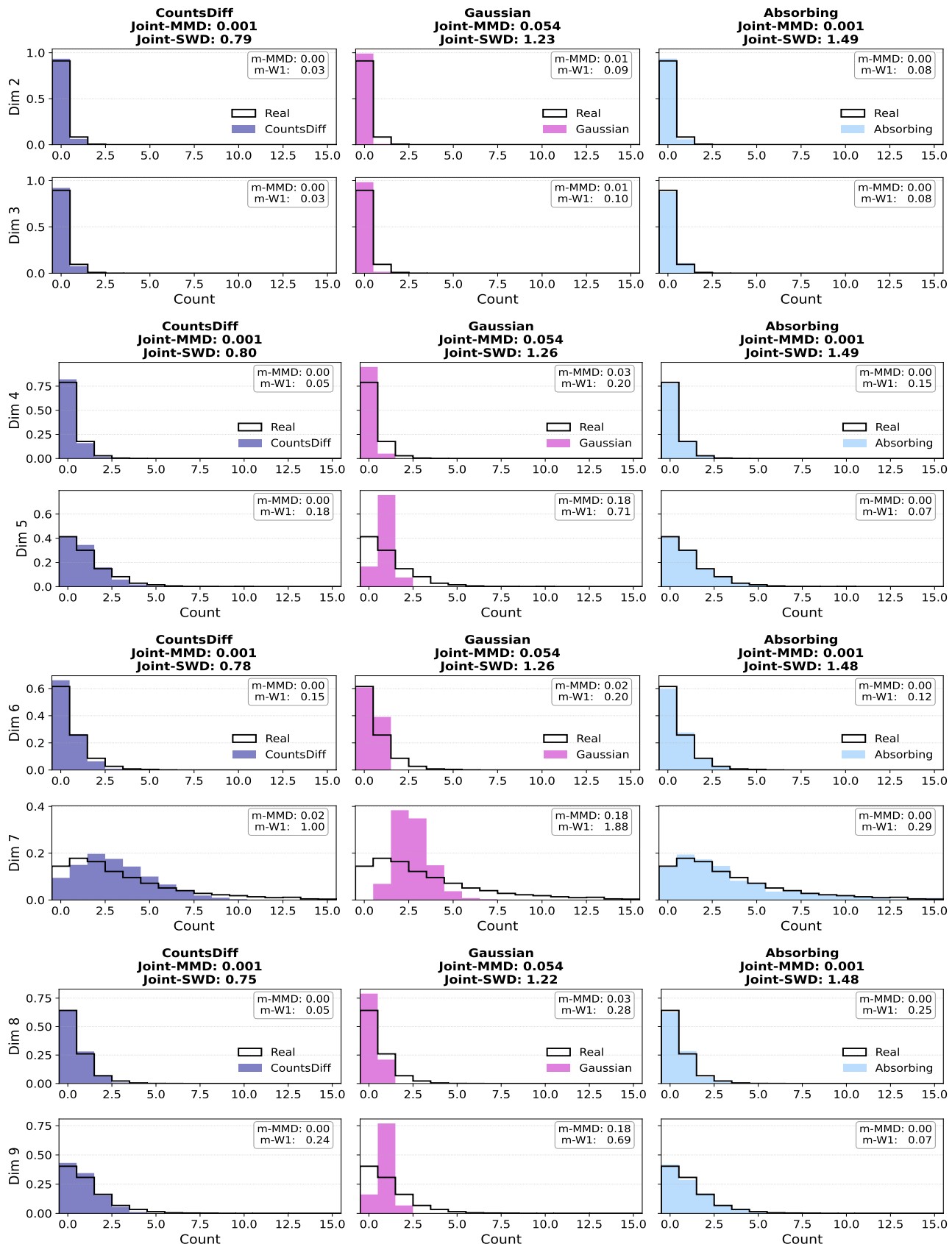

*Figure 7.* Histograms of marginals of dimensions 2-9

### E.1.2. Plots of joint distributions of toy data

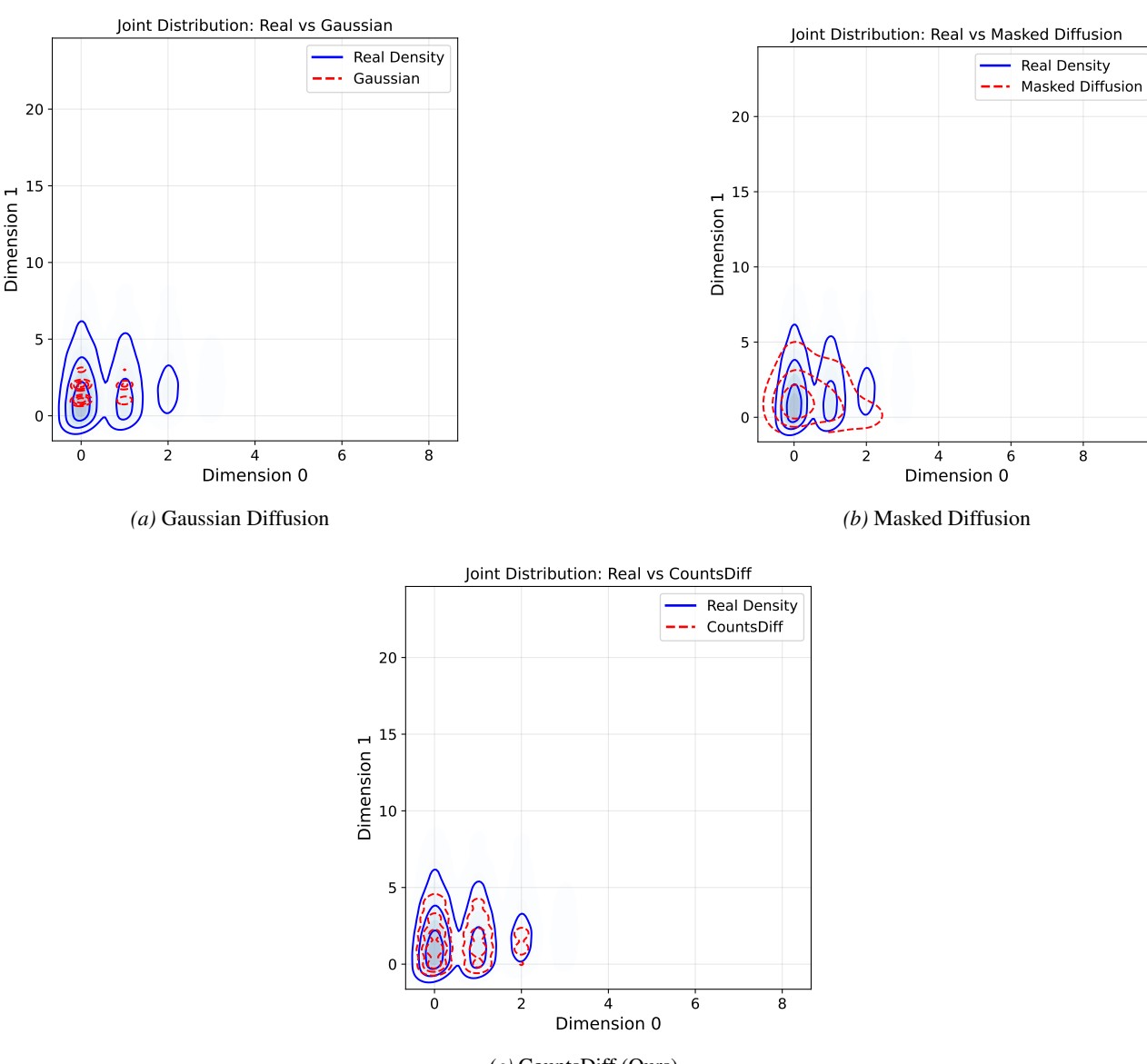

*(a)* Gaussian Diffusion

*(b)* Masked Diffusion

*(c)* CountsDiff (Ours)

*Figure 8.* Joint Kernel Density Estimate (KDE) plots between dimensions 0 and 1 of real data (blue contours) versus model-generated samples (red dashed contours). Gaussian Diffusion suffers from mode collapse, resulting in a much tighter, less diverse distribution. Masked Diffusion exhibits a broader, more diffuse distribution with 'leaked' probability mass and slightly less correlation between the dimensions, indicating outliers and overfitting to the marginals. CountsDiff (Ours) more closely aligns with the true data distribution.

### E.1.3. Randomized Rounding

In the low-counts setting (Figure 9a), we see that the no-rounding method suffers from mode collapse at 0, which fails to preserve the proper marginals. Although both the Poisson approximation and our stochastic random-rounding scheme are empirically effective, in the low-count setting of scRNA-seq data, the Poisson distribution is an unprincipled approximation of a Binomial distribution.

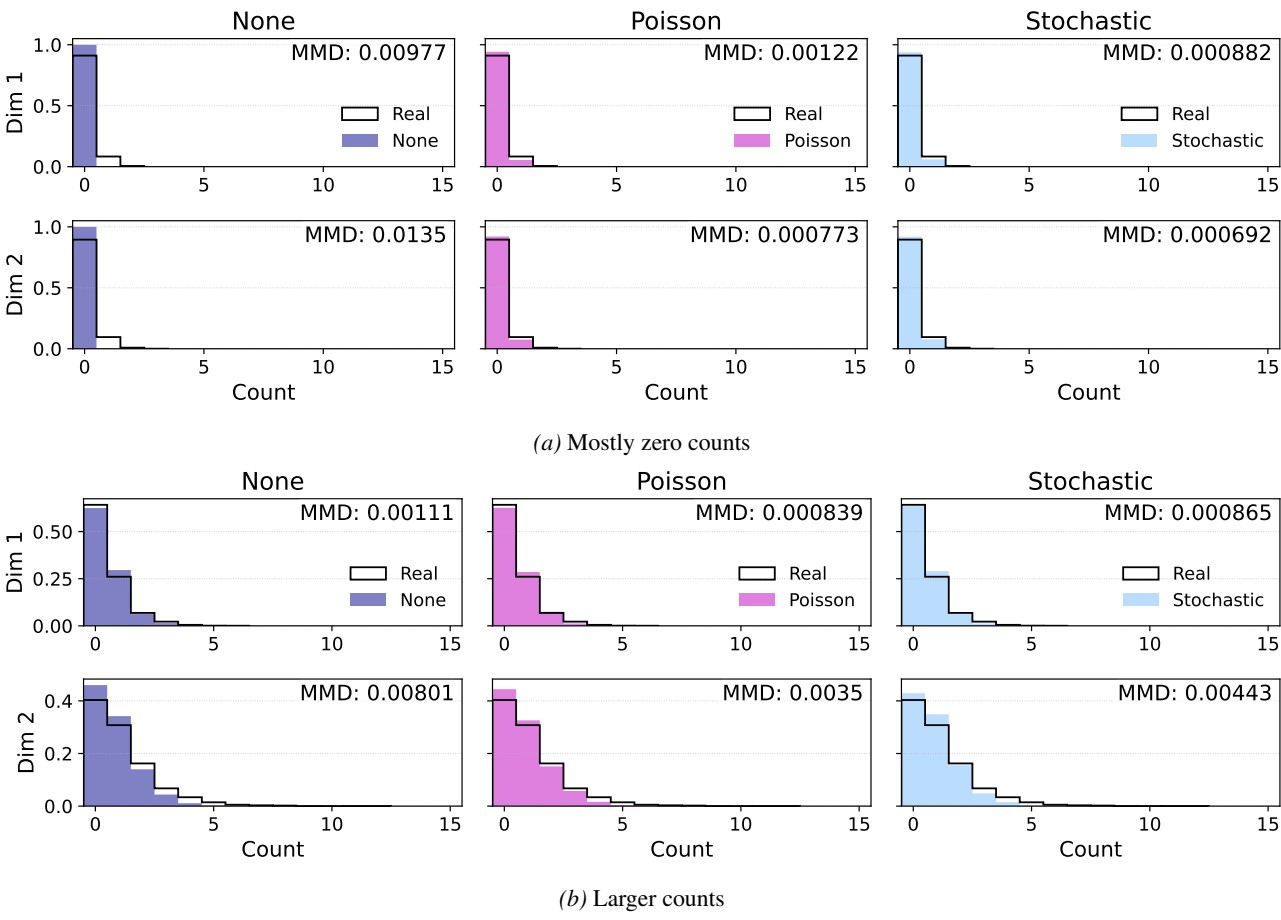

*(a)* Mostly zero counts

*(b)* Larger counts

*Figure 9.* Comparison of Binomial sampling standard rounding, Poisson with no rounding, and Binomial sampling with stochastic rounding across two different counts regimes.

## E.2. CIFAR-10

### E.2.1. COSINE $p$-SCHEDULE STABILIZES TRAINING

We observe substantially reduced instability in the training curves of models trained with the cosine noise schedule versus the FI continuous noise schedule, though both seem to converge to the same optimum value for CIFAR-10. See Figure 10a.

### E.2.2. CIFAR-10 GUIDANCE SWEEP

*Table 5.* FID and IS of 5k images sampled with from CountsDiff with the Continuous FI (left) and Cosine (right) $p$-schedules at various guidance scales with $\eta_{\text{rescale}} = 0.01$

| Guidance Scale | FID ↓ | IS ↑ |
|---|---|---|
| 0.0 | 11.648 | $8.537 \pm 0.391$ |
| 0.1 | 11.606 | $8.599 \pm 0.269$ |
| 0.2 | 12.078 | $8.636 \pm 0.460$ |
| 0.5 | 12.145 | $8.943 \pm 0.439$ |
| 1.0 | 9.820 | $9.286 \pm 0.452$ |
| 2.0 | 13.296 | $9.467 \pm 0.342$ |
| 3.0 | 17.620 | $9.337 \pm 0.394$ |

| Guidance Scale | FID ↓ | IS ↑ |
|---|---|---|
| 0.0 | 11.233 | $8.952 \pm 0.271$ |
| 0.1 | 11.331 | $8.987 \pm 0.498$ |
| 0.2 | 11.933 | $8.741 \pm 0.396$ |
| 0.5 | 11.985 | $9.001 \pm 0.412$ |
| 1.0 | 9.507 | $9.561 \pm 0.368$ |
| 2.0 | 14.154 | $9.498 \pm 0.370$ |
| 3.0 | 18.542 | $9.641 \pm 0.293$ |
| 5.0 | 24.063 | $9.493 \pm 0.167$ |

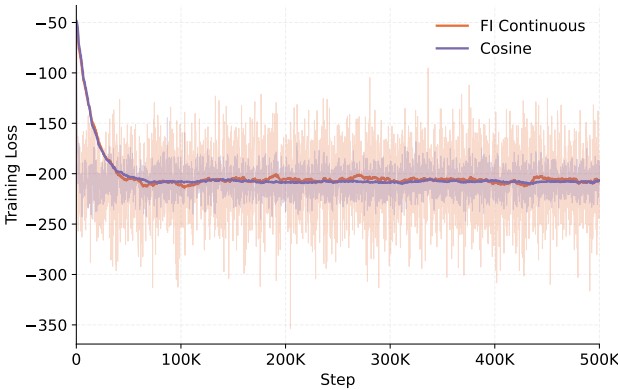

*(a)* Train loss of CountsDiff over 50k steps for FI continuous and cosine $p$-schedules.

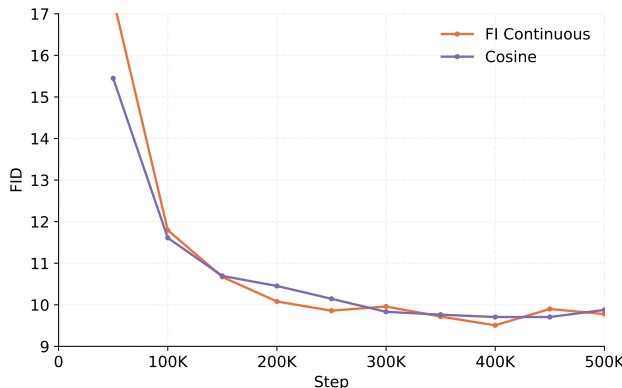

*(b)* Frechet Inception Distance (FID) of 5000 unconditionally generated samples with no attrition of CountsDiff over 50k steps for FI continuous and cosine $p$-schedules.

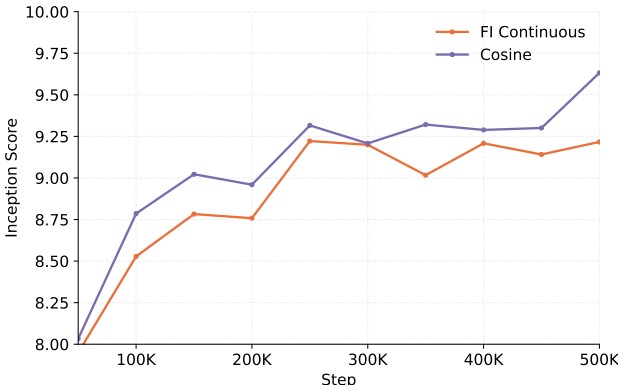

*(c)* Inception Score (IS) of 5000 unconditionally generated samples with no attrition of CountsDiff over 50k steps for FI continuous and cosine $p$-schedules.

*Figure 10.* Train losses and validation metrics throughout training

See Table 5. We observe that at larger guidance scales, increasing guidance improves the IS at the expense of FID, in line with the effect of guidance in other diffusion frameworks. We also find that the cosine $p$-schedule is more responsive to guidance: both metrics vary more in the model trained with the cosine schedule. The cosine schedule also seems to be more

stable at moderate guidance scales, while the FI schedule is more stable at extreme guidance scales.

We observe that both the FID and the IS suffer from small guidance scales, indicating poor unconditional generation compared to the unconditional models. We believe this may be caused by training for the same number of iterations as the unconditional models with ap_uncond = 0.2, so the model has effectively one-fourth as many train steps in the unconditional case as the conditional case.

### E.2.3. CIFAR-10 ATTRITION SWEEP

*Table 6.* FID and IS of 5k images sampled with from CountsDiff various attrition rate strategies

| Attrition Rate Strategy | FID $\downarrow$ | IS $\uparrow$ |
|---|---|---|
| None | 9.666 | $9.463 \pm 0.402$ |
| $\eta_{rescale} = 0.005$ | 9.562 | $9.504 \pm 0.382$ |
| $\eta_{rescale} = 0.01$ | 9.507 | $9.561 \pm 0.368$ |
| $\eta_{rescale} = 0.02$ | 10.400 | $9.653 \pm 0.269$ |
| $\eta_{rescale} = 0.05$ | 12.727 | $9.565 \pm 0.268$ |

Empirically, we find that small, nonzero $\eta_{rescale}$ improved evaluation metrics. We emphasize that these metrics are reported on only $5,000$ samples, so the FID is substantially poorer than it would be for a larger number of samples, as seen in Table 1.

Although the bounds on the attrition schedule for the CountsDiff sampling process resemble those in ReMDM, the strategies that work for remasking in their framework do not necessarily transfer to attrition schedulers in CountsDiff. We hope that future work will shed light on how best to design attrition rate schedules. A particularly exciting direction is value-dependent attrition rates: since the marginal is valid regardless of the attrition rate, one could conceivably set *different attrition rates* for each position in $d$ if, for example, the value in that particular position is deemed "unfit".

### E.3. CelebA

#### E.3.1. QUANTITATIVE METRICS ON CELEBA

*Table 7.* FID of 10k images sampled with from 30M-parameter CountsDiff model trained on CelebA for 500k steps

| $p$-schedule | $\gamma$ | attrition schedule | FID $\downarrow$ |
|---|---|---|---|
| FI Continuous | 1.0 | $\eta_{rescale} = 0.01$ | 9.844 |
| Cosine Continuous | 1.0 | $\eta_{rescale} = 0.01$ | **7.637** |

*Table 8.* FID of 50k images sampled with from 60M-parameter CountsDiff model trained on CelebA for 1M steps

| $p$-schedule | $\gamma$ | attrition schedule | FID $\downarrow$ |
|---|---|---|---|
| Cosine Continuous | 1.5 | $\eta_{rescale} = 0.005$ | 4.948 |

*Table 9.* FID of 5k images sampled with from 60M-parameter CountsDiff model trained on CelebA for 1M steps

| $p$-schedule | $\gamma$ | attrition schedule | FID $\downarrow$ |
|---|---|---|---|
| Cosine Continuous | 1.0 | $\eta_{rescale} = 0.005$ | 7.580 |
| Cosine Continuous | 1.0 | $\eta_{rescale} = 0.01$ | 7.541 |
| Cosine Continuous | 1.5 | $\eta_{rescale} = 0.005$ | 7.217 |
| Cosine Continuous | 1.5 | $\eta_{rescale} = 0.01$ | 7.483 |

### E.3.2. ATTRITION RATE SMOOTHING

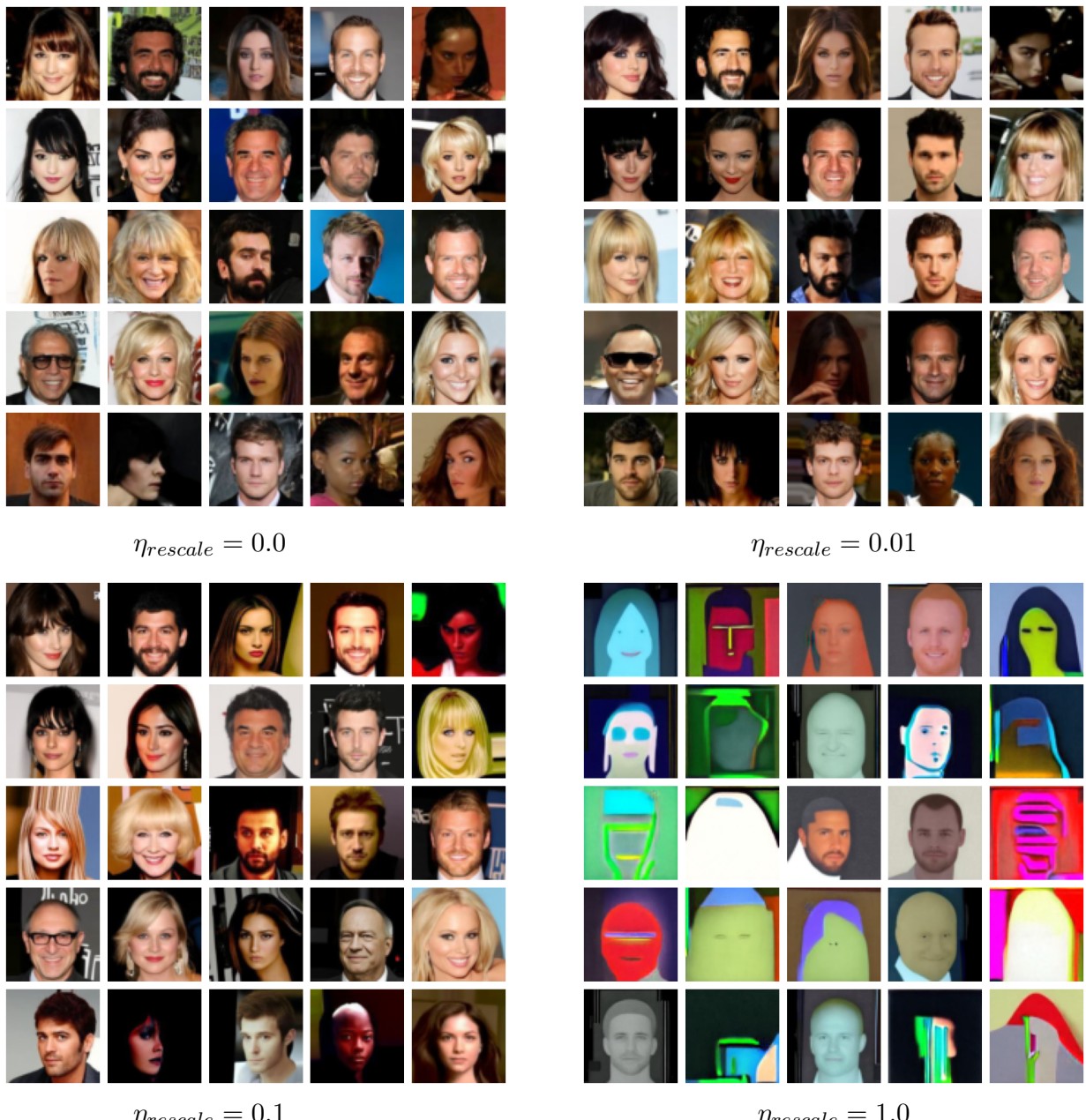

$\eta_{rescale} = 0.0$

$\eta_{rescale} = 0.01$

$\eta_{rescale} = 0.1$

$\eta_{rescale} = 1.0$

*Figure 11.* 25 images drawn from CountsDiff trained on CelebA with increasing $\eta_{rescale}$.

## E.4. Heart cell imputation

*Table 10.* Benchmarking on scRNA-seq imputation, heart with 50% MCAR. Mean (standard error). Methods are grouped into three categories: naive baseline (top), scRNAseq/imputation-specific (middle), and general generative (bottom). Best performance in each category for each metric is bolded, and second best is italicized.

| Method | Pointwise | | | Distributional | | | |
| --- | --- | --- | --- | --- | --- | --- | --- |
| | Spearman↑ | RMSE↓ | Bias | ED↓ | log(scFID)↓ | log(MMD)↓ | SWD↓ |
| Zero imputation | N/A | 9.27(0.13) | -3.49(0.01) | 1.73(0.00) | -0.32(0.00) | -4.50(0.00) | 1.43(0.03) |
| Mean imputation | *0.31(0.00)* | *8.32(0.17)* | *0.03(0.01)* | *0.87(0.00)* | *-2.89(0.00)* | *-5.19(0.01)* | *1.25(0.04)* |
| Conditional Mean | **0.49(0.00)** | **7.62(0.19)** | **-0.01(0.01)** | **0.39(0.00)** | **-4.29(0.01)** | **-6.65(0.01)** | **1.11(0.06)** |
| MAGIC | 0.39(0.00) | 9.24(0.18) | -3.45(0.01) | 1.68(0.00) | -0.38(0.00) | -4.50(0.00) | 1.47(0.05) |
| scIDPMs, 1-sample | 0.31(0.00) | 8.89(0.11) | 1.55(0.01) | 0.76(0.00) | -3.16(0.01) | -5.11(0.01) | 0.88(0.04) |
| scIDPMs, 5-sample | 0.36(0.00) | 7.34(0.14) | 1.55(0.00) | 0.87(0.00) | -3.00(0.00) | -5.29(0.01) | **0.77(0.07)** |
| GAIN | 0.14(0.00) | 7.48(0.22) | -2.02(0.00) | 1.20(0.00) | -0.73(0.00) | -4.88(0.00) | 0.93(0.08) |
| HI-VAE (Poisson) | *0.44(0.00)* | *6.70(0.08)* | *-0.30(0.00)* | 0.29(0.00) | **-5.26(0.01)** | **-6.54(0.01)** | 0.96(0.03) |
| HI-VAE (Gaussian) | 0.32(0.00) | 7.95(0.12) | **-0.21(0.01)** | 0.45(0.00) | -3.67(0.00) | -6.29(0.00) | 1.12(0.04) |
| scGPT (scratch) | **0.49(0.00)** | **6.27(0.14)** | -0.54(0.00) | **0.19(0.00)** | *-5.12(0.01)* | *-6.36(0.00)* | *0.80(0.05)* |
| scGPT (pretrained) | 0.42(0.00) | 6.72(0.14) | -0.56(0.00) | *0.21(0.00)* | -4.05(0.00) | -5.91(0.01) | *0.80(0.04)* |
| xTrimoGene | 0.35(0.00) | 8.14(0.25) | -1.22(0.01) | 0.46(0.00) | -3.82(0.01) | -5.35(0.00) | 1.31(0.08) |
| Forest-Diffusion | 0.05(0.00) | 190.69(0.21) | 88.05(0.06) | 6.65(0.00) | -0.42(0.00) | -0.65(0.00) | 58.76(0.18) |
| ReMDM, 1-sample | *0.33(0.00)* | 8.60(0.36) | **-0.02(0.00)** | **0.01(0.00)** | **-7.46(0.02)** | **-10.18(0.01)** | 1.01(0.13) |
| ReMDM, 5-sample | **0.43(0.00)** | **6.05(0.15)** | **-0.02(0.00)** | 0.16(0.00) | -6.84(0.02) | -8.45(0.01) | **0.60(0.06)** |
| Blackout | 0.32(0.00) | 9.30(0.22) | 0.66(0.02) | 0.12(0.00) | -5.94(0.01) | -9.34(0.03) | 1.29(0.08) |
| **CountsDiff (Ours)**, 1-sample | 0.31(0.00) | 7.34(0.25) | *-0.16(0.01)* | *0.02(0.00)* | *-7.07(0.02)* | *-9.81(0.04)* | *0.87(0.10)* |
| **CountsDiff (Ours)**, 5-sample | **0.43(0.00)** | *6.32(0.23)* | *-0.16(0.01)* | 0.18(0.00) | -6.01(0.01) | -7.56(0.01) | 0.89(0.07) |

## E.5. Fetus Imputation Ablations

We study the effect of varying levels of attrition and guidance on models trained with Cosine schedule, continuous Blackout schedule, and discrete Blackout schedule for 20k steps. Models are evaluated on imputation with 40% of elements missing completely at random (MCAR). We find that optimal guidance and attrition levels are similar, but not identical, to those in imaging data.

### E.5.1. ATTRITION SWEEP

Moderate levels of attrition improve sample quality for cosine and Blackout continuous schedules, with a larger effect on the cosine-schedule trained model. Blackout discrete generates subtantially poorer samples than the other two methods. See Table 11.

*Table 11.* Attrition sweep across schedules. Metrics are reported as mean (standard error)

*(a)* Cosine

| $\eta_{\text{rescale}}$ | scFID $\times 10^4 \downarrow$ | ED $\times 10^2 \downarrow$ |
| --- | --- | --- |
| 0.000 | 2.13(0.34) | 1.86(0.11) |
| 0.005 | 1.86(0.26) | 1.89(0.09) |
| 0.010 | 1.70(0.11) | 1.85(0.06) |
| 0.020 | 1.70(0.16) | **1.68(0.05)** |
| 0.050 | **1.54(0.06)** | 2.01(0.10) |

*(b)* Blackout continuous

| $\eta_{\text{rescale}}$ | scFID $\times 10^4 \downarrow$ | ED $\times 10^2 \downarrow$ |
| --- | --- | --- |
| 0.000 | 3.52(0.15) | 2.51(0.06) |
| 0.005 | 3.44(0.03) | **2.47(0.09)** |
| 0.010 | 3.93(0.58) | 2.49(0.08) |
| 0.020 | 4.13(0.29) | 2.54(0.09) |
| 0.050 | **3.39(0.30)** | 2.60(0.05) |

*(c)* Blackout discrete

| $\eta_{\text{rescale}}$ | scFID $\times 10^4 \downarrow$ | ED $\times 10^2 \downarrow$ |
| --- | --- | --- |
| 0.000 | 17.52(1.38) | 10.49(0.15) |
| 0.005 | 20.09(0.24) | 10.37(0.12) |
| 0.010 | 19.58(1.21) | 10.48(0.14) |
| 0.020 | 20.31(1.42) | 10.83(0.26) |
| 0.050 | 20.39(2.14) | 10.89(0.29) |

### E.5.2. GUIDANCE SWEEP

Guidance also improves sample quality on cosine and continuous Blackout noise schedules, again with a larger effect on the cosine model. Blackout discrete again generates substantially worse samples.

*Table 12.* Guidance sweep across schedules. Metrics are reported as mean (standard error)

*(a)* Cosine

| $\gamma$ | scFID $\times 10^4 \downarrow$ | ED $\times 10^2 \downarrow$ |
|---|---|---|
| 0.0 | 2.15(0.21) | 1.90(0.05) |
| 0.1 | 1.98(0.19) | 1.85(0.11) |
| 0.2 | 1.97(0.26) | 1.75(0.06) |
| 0.5 | 2.05(0.34) | 1.83(0.09) |
| 1.0 | **1.61(0.19)** | 1.73(0.12) |
| 2.0 | 6.04(0.33) | **1.21(0.02)** |
| 3.0 | 17.51(1.19) | 1.79(0.07) |

*(b)* Blackout continuous

| $\gamma$ | scFID $\times 10^4 \downarrow$ | ED $\times 10^2 \downarrow$ |
|---|---|---|
| 0.0 | 3.39(0.56) | 2.72(0.15) |
| 0.1 | 3.74(0.52) | **2.57(0.16)** |
| 0.2 | 3.83(0.17) | 2.64(0.08) |
| 0.5 | **3.17(0.43)** | 2.63(0.10) |
| 1.0 | 4.12(0.52) | 2.66(0.03) |
| 2.0 | 9.47(0.92) | 2.78(0.13) |
| 3.0 | 16.43(1.84) | 3.85(0.14) |

*(c)* Blackout discrete

| $\gamma$ | scFID $\times 10^4 \downarrow$ | ED $\times 10^2 \downarrow$ |
|---|---|---|
| 0.0 | 19.28(0.95) | 11.01(0.25) |
| 0.1 | 18.73(1.43) | 10.67(0.28) |
| 0.2 | 18.51(0.95) | 11.45(0.37) |
| 0.5 | 18.82(1.30) | 10.43(0.42) |
| 1.0 | 21.98(1.79) | 10.90(0.27) |
| 2.0 | 30.78(1.20) | 10.00(0.42) |
| 3.0 | 49.45(1.29) | 11.86(0.21) |

### E.5.3. SWEEP OVER NUMBER OF STEPS

*Table 13.* Effect of number of sampling steps on generation quality, measured by scFID and Energy distance (ED) between imputed and ground-truth test set samples. Metrics are reported as mean (standard error)

| CountsDiff Sampling Steps | scFID $\times 10^4 \downarrow$ | ED $\downarrow$ |
|---|---|---|
| 30 | 2.73(0.19) | 0.01(0.00) |
| 20 | 2.55(0.28) | 0.01(0.00) |
| 15 | 2.32(0.18) | 0.01(0.00) |
| 10 | 2.35(0.15) | 0.02(0.00) |
| 7 | 2.58(0.18) | 0.02(0.00) |
| 5 | 2.97(0.20) | 0.03(0.00) |
| 3 | 3.77(0.25) | 0.05(0.00) |
| 2 | 8.40(0.49) | 0.10(0.00) |
| 1 | 94.72(8.3) | 1.44(0.00) |

## E.6. Additional Metrics for scRNAseq imputation evaluation

Methods are grouped into three categories: naive baseline (top), scRNAseq/imputation-specific (middle), and general generative (bottom). Best performance in each category for each metric is bolded, and second best is italicized. Metrics are reported as Mean (standard error).

*Table 14.* Full metrics (fetus, 50% MCAR).

| Method | Sample-level | | | | | | Distributional | | | |
|---|---|---|---|---|---|---|---|---|---|---|
| | $R^2$ ↑ | RMSE↓ | MAE↓ | Bias | Spearman↑ | Pearson↑ | ED↓ | log(scFID)↓ | log(MMD)↓ | SWD↓ |
| Zero imputation | -7.84(0.01) | 1.91(0.01) | 1.31(0.00) | *-1.31(0.00)* | N/A | N/A | 1.44(0.00) | -2.35(0.00) | -4.08(0.00) | 0.17(0.00) |
| Mean imputation | *-0.86(0.00)* | *1.31(0.04)* | *0.48(0.00)* | **0.00(0.00)** | 0.17(0.00) | 0.25(0.00) | *0.17(0.00)* | -5.01(0.01) | -7.03(0.00) | 0.12(0.01) |
| Conditional Mean | **-0.75(0.01)** | **1.12(0.01)** | **0.45(0.00)** | 0.00(0.00) | 0.20(0.00) | 0.31(0.00) | **0.15(0.00)** | **-6.23(0.01)** | **-7.49(0.01)** | **0.08(0.00)** |
| MAGIC | -7.82(0.01) | 1.88(0.01) | 1.31(0.00) | -1.31(0.00) | **0.21(0.00)** | **0.52(0.01)** | 1.44(0.00) | -2.35(0.00) | -4.07(0.00) | 0.17(0.00) |
| scIDPMs, 1-sample | -22.26(0.16) | 2.25(0.02) | 1.18(0.00) | 0.86(0.00) | 0.08(0.00) | 0.13(0.00) | 0.68(0.00) | -3.04(0.01) | -3.39(0.00) | 0.20(0.00) |
| scIDPMs, 5-sample | -14.81(0.04) | 1.89(0.00) | 1.09(0.00) | 0.86(0.00) | 0.10(0.00) | 0.17(0.00) | 0.88(0.00) | -2.92(0.00) | -3.65(0.00) | 0.18(0.00) |
| GAIN | -7.17(0.01) | 1.87(0.02) | 1.27(0.00) | -1.27(0.00) | 0.04(0.00) | 0.07(0.00) | 1.44(0.00) | -2.34(0.00) | -4.07(0.00) | 0.17(0.00) |
| Hi-VAE (Poisson) | -0.40(0.01) | 1.27(0.02) | 0.47(0.00) | 0.02(0.00) | 0.15(0.00) | 0.25(0.00) | **0.11(0.00)** | *-6.26(0.01)* | -6.68(0.00) | 0.11(0.00) |
| Hi-VAE (Gaussian) | -0.69(0.01) | 1.27(0.03) | 0.45(0.00) | -0.01(0.00) | 0.14(0.00) | 0.24(0.00) | *0.12(0.00)* | -5.78(0.00) | **-6.93(0.01)** | 0.10(0.01) |
| scGPT (scratch) | *-0.13(0.00)* | **1.05(0.02)** | **0.35(0.00)** | -0.20(0.00) | 0.17(0.00) | 0.25(0.00) | 0.25(0.00) | -5.95(0.01) | -6.20(0.00) | *0.09(0.01)* |
| scGPT (pretrained) | -0.79(0.01) | 1.19(0.02) | 0.46(0.00) | -0.44(0.00) | 0.13(0.00) | 0.15(0.00) | 0.51(0.00) | -4.63(0.00) | -5.33(0.00) | 0.11(0.00) |
| xTrimoGene | **-0.03(0.00)** | *1.08(0.02)* | *0.38(0.00)* | -0.11(0.00) | 0.18(0.00) | 0.28(0.00) | 0.37(0.00) | **-6.71(0.02)** | *-6.75(0.01)* | **0.08(0.00)** |
| Forest-Diffusion | $< -10^3$ | 27.18(0.06) | 8.98(0.01) | 8.41(0.01) | 0.03(0.00) | 0.09(0.00) | 2.18(0.00) | -1.54(0.00) | -0.80(0.00) | 4.94(0.01) |
| ReMDM, 1-sample | -2.46(0.35) | 1.66(0.06) | 0.46(0.00) | 0.00(0.00) | *0.11(0.00)* | 0.15(0.00) | **0.01(0.00)** | *-8.98(0.05)* | *-11.69(0.03)* | 0.12(0.01) |
| ReMDM, 5-sample | *-0.56(0.02)* | *1.20(0.03)* | **0.42(0.00)** | 0.00(0.00) | **0.12(0.00)** | **0.22(0.00)** | 0.28(0.00) | -8.44(0.04) | -8.76(0.01) | *0.08(0.01)* |
| Blackout | -2.03(0.01) | 1.39(0.02) | 0.48(0.00) | *0.03(0.00)* | 0.11(0.00) | 0.16(0.00) | *0.02(0.00)* | -7.37(0.01) | -10.78(0.02) | **0.07(0.01)** |
| **CountsDiff (Ours)**, 1-sample | -1.78(0.01) | 1.42(0.03) | 0.48(0.00) | 0.00(0.00) | 0.09(0.00) | 0.13(0.00) | **0.01(0.00)** | **-9.18(0.05)** | **-11.94(0.02)** | *0.08(0.01)* |
| **CountsDiff (Ours)**, 5-sample | **-0.44(0.00)** | **1.17(0.03)** | *0.44(0.00)* | 0.00(0.00) | 0.12(0.00) | 0.20(0.00) | 0.30(0.00) | -7.60(0.03) | -8.64(0.01) | *0.08(0.01)* |

*Table 15.* Full metrics (fetus, 25% MNAR (low-biased))

| Method | Sample-level | | | | | | Distributional | | | |
|---|---|---|---|---|---|---|---|---|---|---|
| | $R^2$ ↑ | RMSE↓ | MAE↓ | Bias | Spearman↑ | Pearson↑ | ED↓ | log(scFID)↓ | log(MMD)↓ | SWD↓ |
| Zero imputation | -38.25(0.15) | 1.00(0.00) | 0.99(0.00) | -0.99(0.00) | N/A | N/A | *1.41(0.00)* | -5.29(0.01) | -6.64(0.00) | 0.03(0.00) |
| Mean imputation | *-11.12(0.15)* | *0.51(0.00)* | *0.32(0.00)* | 0.32(0.00) | 0.26(0.00) | 0.63(0.00) | **0.12(0.00)** | -4.81(0.01) | -7.99(0.00) | 0.02(0.00) |
| Conditional Mean | **-9.70(0.19)** | **0.47(0.00)** | **0.29(0.00)** | **0.28(0.00)** | **0.36(0.00)** | **0.79(0.00)** | **0.12(0.00)** | **-5.82(0.01)** | **-8.69(0.01)** | **0.01(0.00)** |
| MAGIC | -38.48(0.15) | 1.00(0.00) | 0.99(0.00) | -0.99(0.00) | 0.09(0.02) | 0.09(0.02) | 1.41(0.00) | -5.28(0.00) | -6.63(0.00) | 0.03(0.00) |
| scIDPMs, 1-sample | -211.47(4.50) | 2.22(0.01) | 1.23(0.00) | 1.20(0.00) | 0.03(0.00) | 0.24(0.00) | 0.98(0.00) | -3.61(0.00) | -4.20(0.00) | 0.14(0.00) |
| scIDPMs, 5-sample | -117.44(0.68) | 1.77(0.00) | 1.20(0.00) | 1.20(0.00) | 0.03(0.00) | 0.34(0.00) | 1.17(0.00) | -3.52(0.00) | -4.41(0.00) | 0.11(0.00) |
| GAIN | -31.73(0.14) | 0.91(0.00) | 0.91(0.00) | -0.91(0.00) | 0.02(0.00) | 0.44(0.00) | 1.40(0.00) | -5.42(0.01) | -6.64(0.00) | 0.03(0.00) |
| Hi-VAE (Poisson) | -16.91(0.60) | 0.59(0.00) | 0.39(0.00) | 0.39(0.00) | 0.03(0.00) | 0.60(0.00) | 0.22(0.00) | -5.24(0.00) | -8.23(0.00) | 0.02(0.00) |
| Hi-VAE (Gaussian) | -25.97(0.42) | 0.73(0.00) | 0.45(0.00) | 0.45(0.00) | 0.02(0.00) | 0.56(0.00) | 0.37(0.00) | -4.83(0.00) | -7.55(0.00) | 0.03(0.00) |
| scGPT (scratch) | *-4.30(0.13)* | *0.28(0.00)* | **0.15(0.00)** | 0.12(0.00) | **0.33(0.00)** | **0.74(0.00)** | *0.07(0.00)* | *-7.79(0.00)* | **-11.92(0.01)** | *0.01(0.00)* |
| scGPT (pretrained) | **-1.73(0.05)** | **0.24(0.00)** | *0.16(0.00)* | -0.09(0.00) | 0.31(0.00) | 0.66(0.00) | **0.04(0.00)** | **-8.71(0.02)** | *-11.19(0.01)* | **0.00(0.00)** |
| xTrimoGene | -8.88(0.15) | 0.38(0.00) | 0.25(0.00) | 0.25(0.00) | 0.29(0.00) | *0.73(0.00)* | 0.12(0.00) | -6.93(0.01) | -11.06(0.00) | *0.01(0.00)* |
| Forest-Diffusion | $< -10^3$ | 30.53(0.27) | 10.37(0.03) | 9.80(0.03) | 0.03(0.00) | 0.04(0.00) | 2.36(0.00) | **-7.26(0.02)** | -4.13(0.00) | 0.88(0.01) |
| ReMDM, 1-sample | -50.74(3.75) | 1.01(0.05) | 0.32(0.00) | *0.31(0.00)* | **0.47(0.00)** | 0.44(0.00) | 0.31(0.00) | *-6.72(0.01)* | -7.56(0.01) | 0.05(0.01) |
| ReMDM, 5-sample | *-25.34(2.00)* | 0.66(0.01) | 0.31(0.00) | 0.31(0.00) | 0.38(0.00) | **0.61(0.00)** | 0.28(0.00) | -6.61(0.00) | -8.64(0.00) | *0.02(0.00)* |
| Blackout | -42.15(1.93) | 0.88(0.01) | **0.30(0.00)** | 0.30(0.00) | *0.45(0.00)* | 0.44(0.00) | 0.31(0.00) | -6.32(0.01) | -7.73(0.00) | *0.03(0.00)* |
| **CountsDiff (Ours)**, 1-sample | -42.00(1.00) | 0.87(0.00) | **0.30(0.00)** | 0.30(0.00) | 0.42(0.00) | 0.40(0.00) | 0.30(0.00) | -6.41(0.01) | -7.62(0.00) | *0.03(0.00)* |
| **CountsDiff (Ours)**, 5-sample | **-19.44(0.40)** | **0.58(0.00)** | **0.30(0.00)** | 0.30(0.00) | 0.35(0.00) | 0.58(0.00) | **0.26(0.00)** | -6.24(0.01) | **-8.78(0.00)** | **0.02(0.00)** |

*Table 16.* Full metrics (heart, 50% MCAR). Mean (standard error)

| Method | Sample-level | | | | | | Distributional | | | |
|---|---|---|---|---|---|---|---|---|---|---|
| | $R^2$ ↑ | RMSE↓ | MAE↓ | Bias | Spearman↑ | Pearson↑ | ED↓ | log(scFID)↓ | log(MMD)↓ | SWD↓ |
| Zero imputation | **-2.02(0.01)** | 9.27(0.13) | 3.49(0.01) | -3.49(0.01) | N/A | N/A | 1.73(0.00) | -0.32(0.00) | -4.50(0.00) | 1.43(0.03) |
| Mean imputation | -6.71(0.03) | *8.32(0.17)* | *2.93(0.01)* | *0.03(0.01)* | 0.31(0.00) | 0.33(0.00) | *0.87(0.00)* | *-2.89(0.00)* | *-5.19(0.01)* | *1.25(0.04)* |
| Conditional Mean | *-2.94(0.05)* | **7.62(0.19)** | **2.36(0.01)** | **-0.01(0.01)** | **0.49(0.00)** | **0.57(0.00)** | **0.39(0.00)** | **-4.29(0.01)** | **-6.65(0.01)** | **1.11(0.06)** |
| MAGIC | -2.01(0.01) | 9.24(0.18) | 3.45(0.01) | -3.45(0.01) | 0.39(0.00) | 0.49(0.00) | 1.68(0.00) | -0.38(0.00) | -4.50(0.00) | 1.47(0.05) |
| scIDPMs, 1-sample | -6.37(0.06) | 8.89(0.11) | 3.20(0.01) | 1.55(0.00) | 0.31(0.00) | 0.33(0.00) | 0.76(0.00) | -3.16(0.00) | -5.11(0.01) | 0.88(0.04) |
| scIDPMs, 5-sample | -4.17(0.02) | 7.34(0.14) | 2.96(0.01) | 1.55(0.00) | 0.36(0.00) | 0.40(0.00) | 0.87(0.00) | -3.00(0.00) | -5.29(0.01) | **0.77(0.07)** |
| GAIN | -1.86(0.01) | 7.48(0.22) | 2.84(0.01) | -2.02(0.00) | 0.14(0.00) | 0.15(0.00) | 1.20(0.00) | -0.73(0.00) | -4.88(0.00) | 0.93(0.08) |
| Hi-VAE (Poisson) | 0.01(0.00) | 6.70(0.08) | *1.99(0.01)* | -0.30(0.00) | *0.44(0.00)* | *0.51(0.00)* | 0.29(0.00) | **-5.26(0.01)** | **-6.54(0.01)** | 0.96(0.03) |
| Hi-VAE (Gaussian) | -1.49(0.00) | 7.95(0.12) | 2.50(0.01) | -0.21(0.01) | 0.32(0.00) | 0.36(0.00) | 0.45(0.00) | -3.67(0.00) | -6.29(0.00) | 1.12(0.04) |
| scGPT (scratch) | **0.12(0.00)** | **6.27(0.14)** | **1.74(0.01)** | -0.54(0.00) | **0.49(0.00)** | **0.54(0.00)** | **0.19(0.00)** | *-5.12(0.01)* | -6.36(0.00) | 0.80(0.05) |
| scGPT (pretrained) | *0.04(0.00)* | 6.72(0.14) | 1.99(0.01) | -0.56(0.00) | 0.42(0.00) | 0.45(0.00) | *0.21(0.00)* | -4.05(0.00) | -5.91(0.00) | *0.80(0.04)* |
| xTrimoGene | 0.00(0.00) | 8.14(0.25) | 2.23(0.01) | -1.22(0.01) | 0.35(0.00) | 0.38(0.00) | 0.46(0.00) | -3.82(0.01) | -5.35(0.00) | 1.31(0.08) |
| Forest-Diffusion | $< -10^3$ | 190.69(0.21) | 90.49(0.06) | 88.05(0.06) | 0.05(0.00) | 0.12(0.00) | 6.65(0.00) | -0.42(0.00) | -0.65(0.00) | 58.76(0.18) |
| ReMDM, 1-sample | -1.39(0.03) | 8.60(0.36) | 2.19(0.01) | **-0.02(0.00)** | *0.33(0.00)* | 0.41(0.00) | **0.01(0.00)** | **-7.46(0.02)** | **-10.18(0.01)** | 1.01(0.13) |
| ReMDM, 5-sample | *-0.27(0.01)* | **6.05(0.15)** | **1.82(0.01)** | -0.02(0.00) | **0.43(0.00)** | **0.53(0.00)** | 0.16(0.00) | -6.84(0.02) | -8.45(0.01) | **0.60(0.06)** |
| Blackout | -1.12(0.02) | 9.30(0.22) | 2.63(0.02) | 0.02(0.00) | 0.32(0.00) | 0.39(0.00) | 0.12(0.00) | -5.94(0.01) | -9.34(0.03) | 1.29(0.08) |
| **CountsDiff (Ours)**, 1-sample | -0.93(0.01) | 7.34(0.25) | 2.24(0.01) | -0.16(0.01) | 0.31(0.00) | 0.39(0.00) | *0.02(0.00)* | *-7.07(0.02)* | *-9.81(0.04)* | *0.87(0.10)* |
| **CountsDiff (Ours)**, 5-sample | **-0.03(0.00)** | *6.32(0.23)* | *1.86(0.01)* | -0.16(0.01) | **0.43(0.00)** | *0.52(0.00)* | 0.18(0.00) | -6.01(0.01) | -7.56(0.01) | 0.89(0.07) |

### E.7. Cell Classifier Predictions

We trained an XGBoost Classifier with 100 estimators and a max depth of 6 on the training set for both the fetal and heart datasets to predict cell type. Imputed data points were then evaluated on classification accuracy and F1-score with the classifier. Using the same imputation missingness schemes, we report the results on competitive methods in Tables 17 and 18.

*Table 17.* Cell classifier results for scRNA-seq imputation on human fetal cell atlas with 50% MCAR. Metrics are reported as mean (standard error)

| Model Method | Accuracy ↑ | F1↑ |
|---|---|---|
| Zero imputation | 0.82(0.00) | 0.53(0.01) |
| Mean imputation | 0.81(0.00) | 0.49(0.01) |
| Conditional Mean | 0.82(0.00) | 0.55(0.01) |
| MAGIC | 0.62(0.00) | 0.27(0.01) |
| ReMDM, 1-sample | 0.82(0.00) | 0.53(0.01) |
| ReMDM, 5-samples | 0.82(0.00) | 0.53(0.01) |
| CountsDiff, 1-sample | 0.81(0.00) | 0.50(0.01) |
| CountsDiff, 5-samples | 0.81(0.00) | 0.51(0.01) |

*Table 18.* Cell classifier results for scRNA-seq imputation on human fetal cell atlas with 25% low-biased MNAR. Metrics are reported as mean (standard error).

| Model Method | Accuracy ↑ | F1↑ |
|---|---|---|
| Zero imputation | 0.82(0.00) | 0.54(0.01) |
| Mean imputation | 0.81(0.00) | 0.51(0.01) |
| Conditional Mean | 0.82(0.00) | 0.53(0.01) |
| MAGIC | 0.76(0.00) | 0.47(0.01) |
| ReMDM, 1-sample | 0.82(0.00) | 0.54(0.01) |
| ReMDM, 5-samples | 0.82(0.00) | 0.54(0.01) |
| CountsDiff, 1-sample | 0.81(0.00) | 0.52(0.01) |
| CountsDiff, 5-samples | 0.82(0.00) | 0.52(0.01) |

*Table 19.* Cell classifier results for scRNA-seq imputation on human heart cell atlas with 50% MCAR. Metrics are reported as mean (standard error)

| Model Method | Accuracy ↑ | F1↑ |
|---|---|---|
| Zero imputation | 0.99(0.00) | 0.94(0.01) |
| Mean imputation | 0.99(0.00) | 0.93(0.01) |
| Conditional Mean | 0.99(0.00) | 0.95(0.01) |
| MAGIC | 0.93(0.00) | 0.79(0.01) |
| ReMDM, 1-sample | 0.99(0.00) | 0.94(0.01) |
| ReMDM, 5-samples | 0.99(0.00) | 0.940.01) |
| CountsDiff, 1-sample | 0.98(0.00) | 0.94(0.01) |
| CountsDiff, 5-samples | 0.99(0.00) | 0.93(0.01) |

# F. Large Language Model Statement

Large Language Models and associated AI tools were used for the following:

1. Deep research queries to Gemini and ChatGPT were used for retrieval and discovery of related works, to ensure fair credit was given to works we may not have been previously aware of.

2. AI IDE assistants were used to aid in debugging, figure generation, and implementation of certain simple, canonical methods. They were also used to aid re-factoring of TF methods (GAIN, HI-VAE) into PyTorch.

3. LLM assistants were used intermittently to polish already written text to make it more comprehensible to readers.

