# OpenReview forum: "CountsDiff: A diffusion model on the natural numbers for generation and imputation of count-based data"
_ICML.cc/2026/Conference — ICML 2026 regular_

### Official Review · Reviewer_v3Ny · 2026-02-28

**Soundness:** 2
**Presentation:** 2
**Significance:** 2
**Originality:** 2
**Overall Recommendation:** 4
**Confidence:** 5

**Summary:**

This manuscript develops a diffusion modeling framework operating natively on non-negative integers through pure-death stochastic processes, introducing reparameterized noise schedules and attrition-based sampling mechanisms adapted from continuous and categorical diffusion paradigms. Empirical evaluation spans synthetic count distributions, quantized natural images, and single-cell transcriptomic imputation tasks, demonstrating competitive performance against existing discrete generative approaches. The work primarily offers conceptual unification and incremental methodological extensions rather than fundamental algorithmic breakthroughs, with limited theoretical novelty and experimental validation confined to artificially constructed or heavily preprocessed benchmarks.

**Compliance With Llm Reviewing Policy:**

Affirmed.

**Final Justification:**

I think I should be kind and be a good reviewer. So I plan to improve my score.

**Key Questions For Authors:**

Q1 Modeling limitations and expressive power of  pure death process

The pure death process used in the paper prohibits overall growth during the generation process compared to the general birth and death process. What are the basic limitations that this constraint theoretically brings? Has the author empirically verified that this setting does not artificially weaken the ability to express the distribution of excessively discrete count data?

Q2 Theoretical rationality of  scheduling and noise structure

The theoretical basis for directly transferring cosine scheduling derived from Gaussian diffusion to discrete counting space is mainly based on signal-to-noise ratio matching. However, there are essential differences between binomial distribution and Gaussian distribution in tail behavior and moment structure. Is there a theoretical guarantee that they are still reasonable in a counting space with strong coupling of mean and variance?

Q3 extinction - regeneration mechanism and distribution consistency

The extinction regeneration mechanism allows for non monotonic trajectories, but the reverse process theoretically needs to maintain consistent edge distributions. How does the author reconcile this requirement at the theoretical and implementation levels, and is there any empirical diagnosis indicating that finite step sampling does not introduce cumulative distribution drift?

Q4  Simplification of Algorithm Design and Risk of Information Loss

The design choices, including random rounding and scalar gated compression, sacrifice modal matching and feature resolution while maintaining expectations. For biological counting applications with zero expansion characteristics, will such simplification confuse true zero values with technical deficiencies, and will the cost be systematically evaluated?

Q5 model capacity, structure selection, and algorithm advantage differentiation

In single-cell experiments, the method achieves similar or better performance than a strong baseline with fewer parameters. Has the author conducted capacity matching experiments to distinguish whether performance differences come from architectural constraints or algorithmic advantages of counting native modeling?

Q6 The practicality and scope of application of  experimental setup

Image experiments construct artificial counting data by quantifying pixels, while biological experiments involve preprocessing steps such as gene filtering. To what extent have these designs changed the mechanism of raw data generation and loss, and are the conclusions transferable to the native domain of real counting?

Q7 The trade-off between cost and accuracy of  theory modification

Compared to existing methods that achieve precise NLL equivalence through specific configurations, this paper introduces additional flexibility through scheduling and guidance mechanisms. Has the author sacrificed any guarantee of accuracy, and if so, how can this trade-off be reasonably supported in theory or experiment?

Q8 computational efficiency, scalability, and method positioning

Continuous time modeling supports flexible discretization, but it is still unclear how sample quality deteriorates and how much computational benefit can be obtained when the number of steps is drastically reduced; Meanwhile, in the context of more general tables or hybrid diffusion models, how should the author argue for the necessity of using CountsDiff instead of more general methods in real-world multi type data?

**Limitations:**

This work attempts to solve a meaningful counting modeling problem, but overall there are still prominent limitations: in theory, key assumptions such as pure dead processes, infinite support, and noise design based on moment matching limit the model's ability to express complex distribution patterns, and lack more rigorous distribution level arguments; In terms of methodology, the introduction of attrition, random rounding, and guidance mechanisms increases the complexity of sampling and optimization, but does not provide sufficient convergence, stability, or variance control analysis. Some designs rely more on analogical intuition rather than principled deduction; At the empirical level, there is a deviation between the evaluation settings and the real application scenarios, and the baseline comparison and ablation are insufficient to clearly attribute the sources of performance improvement. The value of downstream tasks also lacks verification; In addition, the assumptions of dimension independence, data type, and scale in the model limit generalization, and explanatory and deployment level uncertainties have not been fully discussed. These factors collectively weaken the practical applicability and scientific influence of the method in its current form.

**Strengths And Weaknesses:**

# There is no conclusion, and the introduction summarizes contributions and randomly cites papers.

## **Summary**

This manuscript proposes a diffusion framework that operates directly on **natural numbers** for count‑based data generation and imputation. Although the work addresses a legitimate gap in current diffusion methodologies, the technical contributions remain largely incremental, and the experimental validation falls short of the rigor expected for a top‑tier venue.

## **Strengths**

**S1.** The paper correctly identifies that standard **continuous diffusion** applied to count data exhibits mode collapse, while **categorical diffusion** ignores ordinal structure. This provides a reasonable motivation for native count‑based modeling.

**S2.** The connection drawn between **survival probability schedules** in pure‑death processes and noise schedules in Gaussian diffusion offers some conceptual clarity for practitioners familiar with continuous diffusion frameworks.

## **Weaknesses**

W1 Insufficient novelty of  core method

The main technical contribution of the paper is closer to the recombination of Blackout Diffusion and the introduction of standard components from other diffusion model fields, rather than proposing fundamentally new algorithmic ideas; The so-called attrition mechanism is highly similar in form and function to the heavy masking strategy in discrete diffusion, and its theoretical analysis only stays at direct operation on the edges of the binomial distribution, without providing substantial new insights.

W2 The practical significance of the  experimental setup is questionable

By quantifying continuous pixels to construct counting data on CIFAR-10 and CelebA, a human task that does not exist in natural applications has been formed, making it difficult to verify the practical value of the method in real counting scenarios; At the same time, scRNA seq imputation experiments only achieved marginal improvements above weaker baselines, and often failed to surpass simple conditional mean imputation in Spearman correlation.

W3 compares fairness and capacity issues covered up

The comparison with ReMDM is misleading because CountsDiff uses significantly fewer parameters due to architectural constraints, which reflects model capacity limitations rather than algorithm efficiency advantages; However, the paper presents it as a method advantage without distinguishing the impact of structural limitations and algorithm improvements through capacity matching experiments.

W4 Several key designs of  lack principled support

The key designs, including classifier free guidance for random rounding and counting spaces, rely heavily on heuristic processing such as expected preservation or logarithmic interpolation, lacking strict theoretical support; The negation of Poisson approximation is more like a consideration of differentiability rather than a methodological necessity, and there is not sufficient discussion on the consistency guarantee of non negativity and integer constraints in the sampling process.

W5 Influence Representation and **Related Work Coverage** Imbalance

The paper's argument on its biological impact is clearly biased, but it ignores the long-standing controversy in the field of single-cell imputation about the possibility of generative models "fabricating biological signals"; In addition, the relevant work section did not fully discuss the continuous diffusion models of tabular data (such as TabDDPM, etc.), which have demonstrated strong competitiveness without the need for counting domain specialization assumptions, thereby weakening the necessity of CountsDiff positioning.

### **The author lacks relevant citations for the latest papers and developments on diffusion, and the cited papers are outdated and need to be updated.**

## **Additional Critical Observations**

Firstly, the theoretical framework does not match the complexity of expression.

The paper heavily relies on continuous time Markov processes and their corresponding Kolmogorov forward equations, introducing significant sign and derivation complexity, but does not bring proportional practical benefits; The relevant theoretical content could have been presented in a more intuitive and target oriented manner to reduce the cost of understanding and reproduction.

Secondly, experimental preprocessing and evaluation settings weaken the relevance of reality.

In scRNA seq experiments, aggressive gene filtering significantly altered the mechanism of missing raw data, making it difficult to extrapolate the results to complete transcriptome and heavy tailed count distribution scenarios; In the image experiment, although the FID and IS metrics are close to Blackout Diffusion, they still lag significantly behind mainstream continuous diffusion models, weakening the persuasiveness of the method for expanding into complex practical fields.

Thirdly, there is a lack of principled support for key hyperparameters and method effectiveness.

**The selection of guidance strength** clearly relies on empirical regulation and lacks theoretical or systematic analysis. Its trade-off trend between FID and IS also conforms to common patterns and does not reflect the unique advantages of counting diffusion; At the same time, the introduction of some terms and mechanisms in the article is not clear enough, which reduces the overall readability and reproducibility of the method.

Fourthly, the discussion on reproducibility and impact is superficial.

Although the paper promises open code, only an anonymous repository is provided during the review period, and key implementation details cannot be verified; In addition, the discussion of ethical issues in the impact statement is relatively vague and lacks in-depth analysis of the potential risks of amplifying systematic biases in biological data with technical deviations using generative counting models.

## **Overall Assessment**

While the manuscript explores an important direction in extending diffusion models to discrete structured spaces, it does not deliver commensurate **technical innovation** or **experimental rigor**. The contribution would benefit from clearer scoping, stronger ablations isolating individual design choices, and validation on **genuinely count‑native datasets** rather than artificially constructed benchmarks.

---

> ### Author Rebuttal · Authors · 2026-03-31
>
> We appreciate the detailed feedback and first clarify some possible misunderstandings on **theoretical constraints:** Monotonicity (pure-birth/death) (**Q1**) *would* be restrictive in the reverse process, which is why *we do not use it and* instead introduce *attrition*. Training depends only on marginals of the forward process [**L1148**], which are independent of birth/death dynamics, so there are *no structural constraints* nor loss of expressivity to validate empirically.
>
> Rounding (**W6, Q4**) concerns are valid for prior work (Poisson approximation) and *resolved by our randomized rounding* which allows sampling directly from binomials [**L1169**] without approximation.
>
> Non-negativity of guidance (**Q6**) is preserved since the product of non-neg values [**L218**] is non-neg. Log-interpolation is thoroughly motivated (**W7**) by Nisonoff (2024).
>
> On theoretical guarantees of transferred noise schedule (**Q2**), loss weighting (**Q8**), attrition schedule (**Q14**), guidance (**Critical Obs. 4**): all lack theoretical guarantees in original domains. Providing guarantees would require fundamental theoretical advances in diffusion modeling as a whole.
>
> Reverse process marginals are preserved (**Q3**), shown in **Prop 3.2 (W2)**; Exact NLL equivalence (**Q11**) is shown in **Prop** **B1**.
>
> Guidance, transfer, and reverse-process are validated empirically in **Sec 5** & responses to **Pnhx, nUDR**.
>
> Adapting RePaint(**Q9**): their eqs 8a,b,c only require Markovianity, satisfied by CountsDiff being a CTMC.
>
> The concerns about **ReMDM**, capacity constraints, infinite support (**W5, Q5, Theoretical constr.**) stem from a misunderstanding: existing methods ReMDM/categorical diffusion predict logit vectors over the support of each dimension, which must be truncated to fit the output layer. CountsDiff *resolves these key limitations* by predicting one rate per dimension **[eq 6]**.
>
> Significance of our contribution (**W1,W2**): CountsDiff *resolves key limitations* of prior work including fixed p(t), implicit weighting, monotonic sampling, and no guidance, and *elucidates the design space* of count diffusions, bridging the gap to Gaussian/Categorical Diffusion.
>
> Improvement over baselines **(W4):** see response to Pnhx W1.
>
> On missing ref/comparisons to tabular diffusion (**W10, Q15**)
> TabDDPM and successor Forest Diffusion (FD) are cited [**L38**]: we outperform FD [Tables 2,3,10]. Both are wrappers around diffusion and extensible to CountsDiff.
>
> We experiment on 5 datasets with different count scales/sparsity patterns (**scope & generalization**), without any adaptation beyond routine hyperparameter tuning.
>
> On “overstating biological impact” (**W9,Q10**), we merely "highlight biological count assays as a natural use case" [**L38**] where we "match or surpass the performance" [**L41**] of leading methods and "yield promising results for scRNA-seq imputation" [**L433**]. We acknowledge the concerns regarding imputation, which are beyond the scope of this paper, and include experiments on downstream cell-type classification (**Tables 11,12**), though the task is simple and not significantly aided by any imputation method. We will explore effects on harder tasks like gene program detection and regulatory network inference in future work.
>
> “**Aggressive gene filtering**": we follow routine practice for scRNA-seq data for differentially expressed genes with missingness imposed post-filtering.
>
> (**Interpretability & Deployment**): on concerns about stochasticity introduced by our contributions: diffusion models are inherently stochastic. Uncertainty quantification can be attained by sampling multiple times and aggregating (benefit over deterministic methods). Strong single-point predictions on distributional metrics indicates realistic samples, not shortcoming.
>
> Independent dimensions (**Scope & Generalization**): the forward process, like Gaussian/Categorical diff, factorizes across dimensions; cross-dimensional dependencies are captured by learned reverse model, shown empirically in Sec 5, fig 8c.
>
> On **systematic investigation of sample quality vs num steps**, a sweep of num steps found high-quality samples in \<20 steps, 50x less than Blackout default. Performing this sweep without training separate models for each step size is a key **practical advantage of continuous time form** (**Scope and Generalization).** Sweep results will be added to final version.
>
> **Sec 4** clarifies that **images** are used as visual validation in a well-studied space **(Q7, W3)**; Images are routinely quantized and modeled by discrete methods; insights transfer directly: the effect of schedule/attrition/guidance in images (E.2) and scRNAseq (response to nUDR W2) match.
>
> On ablations, please refer to response to nUDR’s W2.
>
> **The full implementation is shared anonymously** per ICML rules; the method is evaluated on **count-native** simulated and scRNAseq data (**Overall Assessment);** **Conclusions** in **Sec 7**.

---

> > ### Author Rebuttal · Reviewer_v3Ny · 2026-04-01
> >
> > renew: Yes, I think the basic solution to my previous problem is that **there are still some details that have not been fixed.**
> > # Please provide specific numerical values and table data for the issue I mentioned earlier, and I will reconsider the rating.
> > I acknowledge the **quality of the author's data**, but I still have **one issue regarding insufficient citation theory**. If the author can address and supplement the citations in the relevant papers below, then my problem will be resolved.
> > [1] White-Box Diffusion Transformer for single-cell RNA-seq generation
> > [2] DAV-GSWT: Diffusion-Active-View Sampling for Data-Efficient Gaussian Splatting Wang Tiles
> > [3] ESCFD: Probabilistic Flow Diffusion Model for Accelerated High-Quality Single-Cell RNA-seq Data Synthesis
> > [4] ScDiVa: Masked Discrete Diffusion for Joint Modeling of Single-Cell Identity and Expression
> > [5] GaiaFlow: Semantic-Guided Diffusion Tuning for Carbon-Frugal Search
> > [6] scSTD: A Swin Transformer-Based Diffusion Model for Recovering scRNA-Seq Data
> > [7] SwiftRepertoire: Few-Shot Immune-Signature Synthesis via Dynamic Kernel Codes​
> > ### Finally, I hope the author can address the issues I mentioned in the final version.

---

> > > ### Author Response · Authors · 2026-04-01
> > >
> > > **[Edit 4/7]**:
> > > We thank the reviewer for acknowledging that we addressed their original concerns and for recommending acceptance with the updated score. Below we discuss the papers mentioned, and we will update the manuscript accordingly.
> > > [1] Though also applied to scRNA-seq generation, this work aims to resolve a different problem with existing generative models, namely lack of interpretability. The contribution is architectural, in contrast with CountsDiff, which is a modeling framework. A White-Box Transformer could be trained with the CountsDiff framework to combine the benefits of the two.
> > >
> > > [3] This work proposes an alternative approach to modeling scRNAseq data, namely using latent diffusion (Gaussian diffusion in a latent space). It is a sensible strategy for generating full rnaseq profiles, but because generation occurs in latent space, it is nontrivial to generate missing counts conditioned on observed counts, which is useful for imputation. CountsDiff, in contrast, does not suffer from this limitation because it generates directly in counts-space.
> > >
> > > [6] This work takes a similar approach to [3], training a latent diffusion model. For “imputation”, they generate 30k synthetic embeddings in latent space, find the 10 nearest synthetic embeddings to the input cell’s embedding, decode each of them, and average them.  **This is more accurately described as denoising than imputation: the method has no mechanism to condition on a partial observation with missing values**, suffering from the same limitation as [3]. ScSTD can be used for true imputation by passing in the masked profiles and checking the replaced values, but it performs poorly (see table), since the input is out-of-distribution for the VAE.
> > >
> > > scSTD on fetus MCAR task
> > > 50k AE train steps, 100k diffusion train steps.
> > >
> > > | Spearman | Pearson | RMSE | MAE | Bias | Log SCFID | Energy Distance | SWD | MMD \\times 10^4 |
> > > | ----- | ----- | ----- | ----- | ----- | ----- | ----- | ----- | ----- |
> > > | \-0.02 (0.00) | \-0.01 (0.00) | 1.87 (0.02) | 1.31 (0.00) | \-1.30 (0.00) | \-2.37 (0.00) | 1.42 (0.00) | 0.16 (0.00) | 168.38 (0.20) |
> > >
> > > Since [2], [4], [5], and [7] are all not peer-reviewed and were posted on arxiv after our work was submitted; we are reading them carefully to determine the relevance to our work and will update the final manuscript accordingly.
> > >
> > > We thank the reviewer for bringing these intriguing works to our attention. We hope this response resolves the remaining concerns.
> > >
> > > ------------
> > >
> > > **[Edit 4/2]**: ***Before we proceed to addressing additional concerns, can the reviewer please clarify that our responses address all issues raised in the original review?***
> > >
> > > -----------
> > > We thank the reviewer for the prompt response. Here is our sweep of sample quality versus number of steps, evaluated by scFID and energy distance:
> > > Table 1:
> > > Sampling steps vs scFID, ED. Metrics are reported as [mean (standard error)] over 10 size-5000 samples of the validation dataset.
> > > | Sampling Steps | scFID$\times 10^4$ (lower is better) | ED (lower is better) |
> > > | :---- | :---- | :---- |
> > > | 30 | 2.73 (0.19) | 0.01 (0.00) |
> > > | 20 | 2.55 (0.28) | 0.01 (0.00) |
> > > | 15 | 2.32 (0.18) | 0.01 (0.00) |
> > > | 10 | 2.35 (0.15) | 0.02 (0.00) |
> > > | 7 | 2.58 (0.18) | 0.02 (0.00) |
> > > | 5 | 2.97 (0.20) | 0.03 (0.00) |
> > > | 3 | 3.77 (0.25) | 0.05 (0.00) |
> > > | 2 | 8.40 (0.49) | 0.10 (0.00) |
> > > | 1 | 94.72 (8.3) | 1.44 (0.00) |
> > >
> > > We observe steady sample quality down to ~10-15 steps, below which sample quality rapidly  degrades. 10-15 steps is a 60-100x speedup from Blackout diffusion’s default 1000 steps, and performing this sweep was made 9x faster by our continuous time contribution, which makes num_steps an inference-time parameter, allowing us to avoid training a separate model for each step size.
> > >
> > > If the “issue [v3Ny] mentioned earlier” instead references controlled ablation on scRNAseq data, please see tables 1-2 in the response to nUDR, copied below for self-containedness.
> > >
> > > **Tables 1-2**: Ablations on fetus data (20k steps)
> > >
> > > Attrition:
> > >
> > > **1a** Cos schedule
> > > |Eta rescale|scFID $\times 10^4$|ED$\times 10^2$|
> > >  |----:|----:|----:|
> > >  |0|2.13 (0.34)|1.86 (0.11)|
> > > |.02|**1.70 (0.16)**|**1.68 (0.05)**|
> > >
> > > **1b** Blackout schedule, continuous
> > > |Eta rescale|scFID$\times 10^4$|ED$\times 10^2$|
> > >  |----:|----:|----:|
> > >  |0|3.52 (0.15)|2.51 (0.06)|
> > >  |.005|**3.44 (0.03)**|**2.47 (0.09)**|
> > >
> > > Guidance sweep
> > >
> > > **2a** Cosine schedule
> > > |Guid. scale|scFID$\times 10^4$|ED$\times 10^2$|
> > >  |----:|----:|----:|
> > > |0|2.15 (0.21)|1.90 (0.05)|
> > > |1|**1.61 (0.19)**|1.73 (0.12)|
> > >  |2|6.04 (0.33)|**1.21 (0.02)**|
> > >
> > > **2b** Blackout schedule, continuous
> > > |Guid. scale|scFID$\times 10^4$|ED$\times 10^2$|
> > > |----:|----:|----:|
> > > |0|3.39 (0.56)|2.72 (0.15)|
> > > |.1|3.74 (0.52)|**2.57 (0.16)**|
> > > |.5|**3.17 (0.43)**|2.63 (0.10)|
> > >
> > > If you intended a different issue, we kindly ask that you edit your rebuttal acknowledgement to clarify, if possible.

---

### Official Review · Reviewer_nUDR · 2026-03-11

**Soundness:** 2
**Presentation:** 3
**Significance:** 2
**Originality:** 3
**Overall Recommendation:** 3
**Confidence:** 3

**Summary:**

This paper introduces a discrete diffusion model called CountsDiff by making the noise schedule and loss-weighting components of Blackout Diffusion adaptive and dynamic. In addition, this incorporates the benefits of continuous time diffusion model to bring in guidance. Empirical evaluations are conducted on three data domains: synthetic data illustrate the use-case for discrete diffusion models vs continuous and categorical diffusion models; images for scaling to high-dimensional data; and scRNA-seq for showcasing a natural application.

**Compliance With Llm Reviewing Policy:**

Affirmed.

**Final Justification:**

I detail my concerns and reasoning for the final assessment in my response to the author's rebuttals.

**Key Questions For Authors:**

**Questions:**

1. Can MAGIC, scIDPMs, GAIN, ForestDiffusion, etc. be applied on CIFAR-10 data? How do they compare with CountsDiff when it comes to scalability?
2. Are cell-type labels or niche-type labels available for the scRNA-seq datasets? If not available, these labels could be obtained from existing methods like scBERT or similar. How does mean imputation conditioned on cell-types perform? Conditioning on sample level covariates is highly aggregative.
3. In selecting genes by sorting by coefficient of variance (line 1220), does this mean selecting top 1000 highly variable or differentially expressed genes (line 235)? How large was the original gene panel in the dataset and how sensitive are the results to dimensionality?
4. A classic concern with scRNA-seq data is integrating samples across multiple batches that show systematic differences unrelated to the underlying biology. Does the model behaviour  account for this? Alternatively, perhaps a discussion can be included to provide insight on how it could be adapted to overcome batch effects.

**Limitations:**

In my opinion, the authors have not meaningfully engaged with the limitations of their approach in the context of relevant scRNA-seq literature which is the primary showcased usage for CountsDiff.

**Strengths And Weaknesses:**

**Strengths:**

- The paper tackles the challenge of overcoming the fixed components of Blackout Diffusion by making it more generalizable and adaptive.
- This method makes available some of the benefits of continuous models to discrete models.
- The manuscript presentation is systematic and inclusion of code for review is commendable.

**Weaknesses:**

A recurring theme within my evaluation of the paper is the mismatch between the model design motivation and empirical validation.

1. Synthetic Data Design:  Experiments on synthetic data are intended to illustrate the limitations of existing diffusion frameworks on natural numbers (line 66). CountsDiff and masked diffusion perform comparably on MMD and Wasserstein-1 metrics. On SWD, masked diffusion's poor performance is attributed to potential overfitting to outliers, but is not validated.
2. Comparisons with Blackout Diffusion: This is missing from the evaluation and is critical to be included because CountsDiff builds on the same theoretical framework. By systematically ablating each of the adaptive changes (e.g. noise schedule, loss weighting), the impact on CountsDiff relative to Blackout Diffusion would be highly impactful. This may be ideally demonstrated using synthetic data where generation is fully controllable.
3. Motivation behind image data: The authors note that CIFAR-10 is used for highlighting the  scalability of the approach to high-dimensions. This is unclear to me because the primary use-case relies on scRNA-seq data which already has high dimensionality, albeit with large sparsity (number of genes can be as high as 20000). Relatedly, benchmarking w.r.t. to time and memory as a function of dimensionality is missing.
4. Real Datasets: Several prominent scRNA-seq datasets are missing, for example, multiple Sclerosis (https://www.ebi.ac.uk/gxa/sc/experiments/E-HCAD-35). Please see the section on Data Availability [1] for references. Baselines such as xTrimoGene [2] and cell foundation models like scGPT [1] which can be finetuned for gene expression prediction are missing. Standard metrics such as Pearson correlation and Energy distance [3] from single-cell and multi-omics literature are missing. Also, MMD is typically computed across multiple bandwidths [4].

**References**:

[1] scGPT: toward building a foundation model for single-cell multi-omics using generative AI https://www.nature.com/articles/s41592-024-02201-0

[2] xTrimoGene: An Efficient and Scalable Representation Learner for Single-Cell RNA-Seq Data, https://openreview.net/forum?id=gdwcoBCMVi

[3] CellFlow enables generative single-cell phenotype modeling with flow matching https://www.biorxiv.org/content/10.1101/2025.04.11.648220v1

[4] Learning single-cell perturbation responses using neural optimal transport https://www.nature.com/articles/s41592-023-01969-x

---

> ### Author Rebuttal · Authors · 2026-03-31
>
> Thanks for the in-depth review; below we address W1-W4 to show that our empirical validation is thoughtfully designed to showcase the strengths and expose the limitations of our framework, which we emphasize is not exclusive to scRNQ-seq imputation.
>
> **W1:** Our synthetic data highlight both the strengths (MMD, marginal Wass) and weaknesses (SWD, variance-matching) of Masked Diff, which CountsDiff resolves. Fig 2 shows significantly higher variance despite similar-shaped distributions, which can be explained only by too-large and/or too-frequent outliers.
>
> **W2:** We fully agree on the criticality of ablation but respectfully disagree they are missing: we ablate different noise schedules [App E.2.2], guidance [E.2.2], attrition [E.2.3], and all changes by **comparing to blackout** (BOD) in Table 1. We don’t explicitly ablate loss weighting because we match to the weighting implicit in BOD [App B.6, L956]. CountsDiff could be further improved by optimizing the weighting (not an option in BOD), but leave this compute-intensive task to future work.
>
> Though ablation on synthetic data is valuable, our low-dim neg-binomial data is not complex enough to differentiate hyperparameters. Images are more complex, well-studied, interpretable, and thus suitable for ablation.
>
> See table 1 in response to Pnhx for **comparison with BOD** in scRNAseq imputation.
>
> For thoroughness, we add ablations for models trained on fetus data for 20k steps to the final manuscript; key rows below.
>
> Tables 1-2: Ablations on fetus data
> Attrition:
> 1a Cosine sched.
> |Eta rescale|scFIDx10^4|EDx10^2|
> |----:|----:|----:|
> |0|2.13 (0.34)|1.86 (0.11)|
> |.02|**1.70 (0.16)**|**1.68 (0.05)**|
>
> 1b BOD continuous
> |Eta rescale|scFIDx10^4|EDx10^2|
> |----:|----:|----:|
> |0|3.52 (0.15)|2.51 (0.06)|
> |.005|**3.44 (0.03)**|**2.47 (0.09)**|
>
> Guidance:
> 2a Cosine sched
> |Guid. scale|scFIDx10^4|EDx10^2|
> |----:|----:|----:|
> |0|2.15 (0.21)|1.90 (0.05)|
> |1|**1.61 (0.19)**|1.73 (0.12)|
> |2|6.04 (0.33)|**1.21 (0.02)**|
>
> 2b BOD continuous
> |Guid. scale|scFIDx10^4|EDx10^2|
> |----:|----:|----:|
> |0|3.39 (0.56)|2.72 (0.15)|
> |.1|3.74 (0.52)|**2.57 (0.16)**|
> |.5|**3.17 (0.43)**|2.63 (0.10)|
>
>  Guidance/attrition both improve predictions. Cosine schedule has better performance and improvements from guidance/attrition are more significant than blackout. Trends match imaging data (E2.2).
>
> **W3:** Images provide a well-studied, visually interpretable setting for ablation [L231]. On benchmarking wrt time/memory, CountsDiff shares scaling behavior with Gaussian Diffusion, largely determined by modeling architecture, not framework, our contribution.
>
> **W4:** While these datasets are commonly benchmarked for other tasks (cell type annotation, pert. prediction, etc.), including in [1], they are not established imputation benchmarks. Our model and most baselines require retraining per-dataset which would be computationally costly. Furthermore, E-HCAD-35 and many other datasets in [1] lack raw count data or have too few samples for diffusion benchmarking. For a general diffusion framework not specific to scRNA-seq we feel our evaluation across two large atlases, two imaging datasets, and synthetic counts is comprehensive and informative.
>
> We emphasize CountsDiff is a general-purpose count diffusion framework, not an scRNA-seq foundation/repr-learning model, like xTrimoGene[2]/scGPT[1]. When used to impute, as regressors rather than generative models, they score well on sample-level metrics but are dominated on distributional metrics (Table 1 in response to Pnhx, which also includes Pearson, ED [3] and updated MMD [4]).
>
> On Key Questions (KQ), MAGIC, scIDPMs, GAIN (**KQ1**) are scRNAseq imputation-specific models (unlike CountsDiff); ForestDiffusion is a tabular-data method not applicable directly to CIFAR-10.
>
> On scalability, MAGIC and GAIN are 1-step predictors and faster than CountsDiff by a constant factor num\_steps. scIDPMs is much slower to train than CountsDiff due to the imputation-specific training. ForestDiffusion took weeks to train/evaluate as it scales both with num\_steps and data dim.
>
> Cell type labels are available(**KQ2**). The conditional mean imputation is conditioned on cell type in addition to sample-level covariates, which is why it is so strong; see response to Pnhx
>
> We select the top genes by coeff of variance (**KQ3**); apologies for the imprecise language which we will correct. Both atlases measure against \~33K genes. We didn’t test extensively on different dimensionality, but preliminary evidence suggests stable model rankings across a mild range in the number of genes. The chosen genes explain 99% and 40% of the experimental variance for the heart and fetus data accordingly.
>
> (**KQ4**): In training, we address batch effects by conditioning on batches, similar to scVI. CountsDiff may additionally help by imputing experimental dropouts, one source of batch effects; further correction can be applied to samples post-imputation.
>
> On limitations, see response to LHg9.

---

> > ### Author Rebuttal · Reviewer_nUDR · 2026-04-02
> >
> > I thank the authors for their detailed response. My concerns on W1 and W2 are adequately addressed. In particular, I acknowledge my oversight in accounting for BOD scores.
> >
> > On W3 and W4:
> > The motivation for Countsdiff as introduced in the abstract, introduction, and related work sections is to model distributions on natural numbers for native applications to scRNA-seq imputation. I view visual interpretability of generated images, which is not count data, as tangential to this. Lines 33-36 explicitly state that the use of image datasets is for scalability to complex, high-dimensional data. The 33K atlas for the heart and fetus is ideally suited to exploring scalability for imputing raw counts. But in the experiments, these have been subset to 500 and 1000 (respectively) out of 33K. Some terminology is unclear with regards to pre-processing (filter genes 'rarely expressed' across cells, Line 1219) and subsetting (coefficient of variance). Besides, lowly expressed genes may have high variance. For final reporting, perhaps a more consistent policy would be increasing the number of top highly variable genes until the explained variance meets a pre-determined threshold rather than 99% for heart and 40% for fetus.
> >
> > To be used as a general-purpose count diffusion framework, Countsdiff would require demonstrating competitive performance across multiple domains involving count data. The proposed use-case for Countsdiff is scRNA-seq imputation. Benchmarking against foundation/ML models designed for imputing scRNA-seq data is fair.
> >
> > scFID measures similarity between the empirical distribution of the model and the underlying data (line 1269). While MMD and scFID would capture different properties, I don't understand how MMD is 0 for all models. MMD = 0 indicates that the two underlying distributions are identical across bandwidths.
> >
> > I respectfully disagree with the authors that the empirical validation and motivation are well-aligned. Results on sample-level metrics are poor compared to other methods for the imputation task and performance on standard a distribution-level metric MMD is unclear.
> >
> > I revise my score to 3 to highlight that the paper has clear merits, but also some weaknesses which overall outweigh the merits in my opinion.

---

> > > ### Author Response · Authors · 2026-04-07
> > >
> > > We thank the reviewer for the continued feedback and hope the following can resolve the remaining concerns.
> > >
> > > > The motivation for Countsdiff as introduced in the abstract, introduction, and related work sections is to model distributions on natural numbers for native applications to scRNA-seq imputation.
> > >
> > > **We feel that the key concerns are centered around the premise that CountsDiff’s main motivation is scRNAseq imputation; this is not the case**. Our motivation is to fill a gap in diffusion modeling literature with a diffusion framework that models natural numbers. scRNA-seq imputation is a proposed application, not the intrinsic motivation of the work (note that we submitted the work to Deep Learning: Generative Models/Autoencoders category, not applications).
> > >
> > > We describe this in the abstract: “CountsDiff, a diffusion framework designed to natively model distributions on the natural numbers … We then highlight biological count assays as **a** natural use case, evaluating CountsDiff on single-cell scRNA-seq imputation”. Thanks to your feedback, we see the introduction mentions only biological assays, and we will revise to include other sources of count-based data. We also see how the last two sentences of section 6.2 can be interpreted as implying that CountsDiff is designed for imputation. We will clarify that the difference between CountsDiff and the referenced diffusion models is the space in which they noise/denoise (observed versus latent space) and consequent adaptability to imputation, not that they are/aren’t designed for imputation.
> > >
> > > > I view visual interpretability of generated images, which is not count data, as tangential to this
> > >
> > > Since our goal is the introduction of a diffusion framework, we feel that the image experiments are valuable and useful in evaluating the framework. They have well-defined and commonly accepted evaluation metrics, enable simple validation of guidance, and yield valuable insight on the smoothing behavior of attrition that we may not have discovered otherwise. Though images come from continuous intensities, pixel values are stored as natural numbers and are commonly used to evaluate discrete diffusion as well [1,2].
> > >
> > > > Lines 33-36 explicitly state that the use of image datasets is for scalability to complex, high-dimensional data.
> > >
> > > Thank you for pointing out this incomplete description; we will update to “exploring the effects of varying the introduced design parameters in a complex, well-studied, and interpretable data domain.”
> > >
> > > > The 33K atlas for the heart and fetus is ideally suited to exploring scalability for imputing raw counts … these have been subset to 500 and 1000 (respectively) out of 33K.
> > >
> > > We agree that heart and fetus data could be used to explore scalability; however, several of the baselines do not scale, so we elected to subset the genes to allow for broader comparison. We feel this decision is justified given the imaging data already validates the ability to scale to high-dimensional data.
> > >
> > > >Some terminology is unclear with regards to pre-processing … lowly expressed genes may have high variance
> > >
> > > We apologize again for the lack of clarity here: these are two separate pre-processing steps: we first filter out genes that were expressed in too few cells and then subset by coefficient of variance. We will update the final version to clarify this.
> > >
> > > >To be used as a general-purpose count diffusion framework, Countsdiff would require demonstrating competitive performance across multiple domains involving count data.
> > >
> > > We feel that our five datasets across three modalities (simulated data, 2x imaging, 2x scRNAseq) are sufficient to evaluate the CountsDiff framework and comparable to other published diffusion modeling papers. For example [1] and [2], two seminal works in discrete diffusion, also evaluate on simulated data, imaging data, and one other data modality.
> > >
> > > **On MMD**
> > > We discovered a bug that biased MMD negatively, which caused 0-clipping. We apologize and report the updated numbers below:
> > >
> > > | Model | MMD $\times 10^4$ (avg across 5 bandwidths) |
> > > | :---- | ----- |
> > > | Blackout | 0.21 (0.00) |
> > > | CountsDiff 1 | **0.07 (0.00)** |
> > > | CountsDiff 5 | 1.77 (0.01) |
> > > | ReMDM 1 | 0.08 (0.00) |
> > > | ReMDM 5 | 1.57(0.02) |
> > > | scGPT pretrained | 48.60 (0.08) |
> > > | scGPT | 20.35 (0.04) |
> > > | xTrimoGene | 11.76 (0.06) |
> > >
> > > Like in other distributional metrics, CountsDiff has the strongest performance. Please refer to response to Pnhx W1 for discussion on sample vs distributional metrics.
> > >
> > > We hope that these clarifications are helpful, and we thank the reviewer again for their time.
> > >
> > > [1] Austin, Jacob, et al. "Structured denoising diffusion models in discrete state-spaces." Advances in neural information processing systems 34 (2021): 17981-17993.
> > > [2] Campbell, Andrew, et al. "A continuous time framework for discrete denoising models." Advances in Neural Information Processing Systems 35 (2022): 28266-28279.

---

### Official Review · Reviewer_LHg9 · 2026-03-11

**Soundness:** 4
**Presentation:** 4
**Significance:** 4
**Originality:** 3
**Overall Recommendation:** 5
**Confidence:** 3

**Summary:**

The paper presents a novel diffusion model operating on the natural numbers that builds on a birth-death process. The method is described well in a well-written paper and appears to work well. The description is convincing, the theory appears to hold, and the experiments are described well and appears to give good results.

**Compliance With Llm Reviewing Policy:**

Affirmed.

**Final Justification:**

The authors clarified all my concerns, and I have nothing more to add. This is a good paper with a solid contribution that should be published.

**Key Questions For Authors:**

Please answer specifically to the questions under *Soundness* from *Strengths And Weaknesses* above.

Line 154: The connection through SNR between CountsDiff and Gaussian diffusion. It would be great if you could prove this, or provide references.

**Limitations:**

Limitations of the present work have not been discussed specifically. I would be interested in the time it takes to generate the images, for instance, compared to Gaussian diffusion models, for example.

**Strengths And Weaknesses:**

*Soundness:*

The theory is explained well, is easy to follow, and appears to hold up.

Line 673: The sum should be divided by |D|, right?

Line 688: The P_noise * q = N(0, 1) is note clear. Please define these objects, or explain, to make it more clear.

Line 799: p(t = 1) = 0, so \mu(t = 1) is not defined.

Add standard errors to all statistics, so that the results can be compared. For instance, tables 1 and 7-9 don't have them.

You say you used LMMs to find works you may have otherwise missed. Did you read these works to see if they were actually relevant? Also, what implementations of simple and canonical methods were predicted by LLMs?

Figure 10: It seems convergence is reached at iteration 2000 or so. Did you observe any improvements after that?


*Presentation:*

It is clear overall what the authors are trying to do, and the technical presentation is clear.

I think it would be helpful if the authors explained the concepts of attrition and the p-schedule.

Line 1538: It says you report results for the following methods, but no methods follow. The section ends there.

Minor things:
 - Line 139, right: Space before comma.
 - Lines 188 and 193: Do mention that the algorithms are in the appendix.
 - Line 210: Remove the "the" before the citation.
 - It sometimes say "$p$-schedule" and sometimes "$p$--schedule" or $p-$scredule. The correct should be "$p$-schedule". For example, Line 223 (right), Figure 5, and Line 1023.
 - Footnote 3: Capital initial letter.
 - Tables 2 and 3: Add references to the compared methods. Add space before the parentheses.
 - Tables: The numbers of significant digits vary. Be consistent, and probably better chose 3 than 4 digits.
 - Line 427: Use \citet here instead of \cite or \citep.
 - Line 783: Missing full stop.
 - Line 801: "form" -> "from"
 - Line 830: Define Logit.
 - Line 859: Should probably be \bar{\alpha}(t), to be consistent.
 - Define random_round in Algorithm 2.
 - Line 1348: Full stop missing.
 - Line 1396: Table reference wrong.
 - Tables 10, 11, 12, and 13: Add spaces before the parentheses.
 - Line 1587: Missing full stop.


*Significance:*

This is an interesting work. It is a generalisation of existing work (Blackout diffusion), but there seem to be plenty of extensions, so there is sufficient novelty. The work could definitely be of use in many other fields, both to generate count data and to impute count data.


*Originality:*

The proposed method appears to be new, even though it is a generalisation and build on top of other works. It is a nice contribution to the field of discrete diffusion models.

---

> ### Author Rebuttal · Authors · 2026-03-31
>
> We thank the reviewer for their positive assessment of our work and are very grateful for their careful scrutiny of typos. We address the specific questions and concerns below:
>
> L673: Yes, this is correct, and will be fixed in the final version, thank you. Please note that this does not change the minimizer.
>
> L688: The line in question is missing an equals sign and should read $P_{noise}(x_T | x_0 ) = q(x_T|x_0) = N(0, 1)$. We hope this is clear now.
>
> L799: Thank you for catching this. We correct this by defining $\mu(t)=-\frac{p'(t)}{p(t)}$ on the open interval (0,1), where it is well defined. The value of $\mu(t)$ in the endpoints $t \in{0,1}$ can be chosen arbitrarily as $\mu(t)$ only interacts with p(t) through the integral $\exp(-\int_0^t \mu(u)du)$. Any choice of $\mu(1)$ will yield  $\exp(-\int_0^1 \mu(u)du)=p(1)=0$ as the points {0,1} have measure zero.
>
> On standard errors: It is standard practice to report a single FID number for image generation experiments (See DDIM[1], DDPM[2], MeanFlows[3]); FID calculated on 50k samples is generally quite stable. We agree in principle that having a standard error for FID would be nice, but generating 50k samples is costly, and computing FID on fewer samples biases it.
>
> On LLM usage: All cited works are relevant; we’ve discovered an instance of ambiguous reference of papers by Ho et al. 2022, which we disambiguate in the final version. Coding assistance was mostly restricted to implementing infrastructure, including training loops, logging, and parallelization of evaluation. Some docstrings were also generated by LLMs, and they were periodically used to help debug dependency issues with baselines. We implemented all novel, CountsDiff-specific methods.
>
> On convergence: In line with other diffusion models, CountsDiff sample quality continues to improve long after the training loss converges. Will add FID vs train time curve to appendix of final version to illustrate this point.
>
> Thanks for the suggestion! We will add a more self-contained explanation to make intuition clearer in the final manuscript. The p-schedule is CountsDiff’s analogue to noising schedules in Gaussian Diffusion (Sec 3.1) and is used to control the rate at which information is destroyed. This allows you to (heuristically) adjust how much of the model’s capacity is dedicated to different noise levels.
> Attrition rate is CountsDiff’s analogue to remasking (3.3). The key intuition motivating attrition is that with monotonic generation if a model overshoots, the error cannot be corrected and can accumulate. Attrition allows for counts to decrease, enabling correction. In images, this manifests as a smoothing effect (fig 4). This intuition also motivates exploration of confidence-based attrition schedules, similar to recent work in Remasking.
>
> >Line 1538: It says you report results for the following methods, but no methods follow. The section ends there.
>
> This is referencing the methods in the tables below it. We recognize that this is unclear and are updating it in the final version.
>
> The minor feedback is fantastic, thank you for pointing all of them out; the final manuscript will be updated to address them.
>
> On limitations: we thank the reviewer for pointing out that we do not explicitly discuss limitations of the work. The final manuscript will contain a dedicated limitations/future work section, discussing the following:
>
> Generative Modeling:
> CountsDiff requires a moderate number of sampling steps, matching DDPM/DDIM-style Gaussian Diffusion in sampling speed but significantly slower than recent few- and one-step samplers (i.e., consistency models, mean flows).Extending such ideas to CountsDiff is an important avenue to explore.
> We only consider one transferred candidate per introduced design choice (noise schedule, loss weighting, attrition). Further optimization, particularly attrition, the least studied of the three, is likely to yield meaningful modeling gains.
>
> Limitations for scRNAseq imputation applications:
> For scRNA-seq imputation specifically, our evaluation is limited to a subset of highly variable genes, in line with many deep-learning methods in scRNA-seq. Scaling to the full transcriptome is untested. We also do not explore pre-training across multiple scRNA-seq datasets or task/domain-specific adaptations to the training objective, analogous to those developed for Gaussian diffusion-based models in scIDPMs[4] and Squidiff[5].
>
> References:
> 1. Ho, Jonathan et al.. "Denoising diffusion probabilistic models." (2020):
> 2. Song, Jiaming et al. "Denoising diffusion implicit models." (2020)
> 3.  Geng, Zhengyang, et al. "Mean flows for one-step generative modeling." (2025).
> 4. Zhang, Z et al. scIDPMs: Single-Cell RNA-Seq Imputation Using Diffusion Probabilistic Models (2025)
> 5. He, Siyu, et al. "Squidiff: predicting cellular development and responses to perturbations using a diffusion model." (2026)

---

> > ### Author Rebuttal · Reviewer_LHg9 · 2026-04-02
> >
> > Thank you for the clarifications. Please also make sure those go in the final version of the paper. Great job!

---

### Official Review · Reviewer_pnhx · 2026-03-14

**Soundness:** 3
**Presentation:** 3
**Significance:** 4
**Originality:** 3
**Overall Recommendation:** 5
**Confidence:** 3

**Summary:**

The paper extends and clarifies the design space of Blackout diffusion by introducing the CountsDiff framework which uses birth-death processes to model the natural numbers. This framework allows for continuous time training, classifier free guidance, a noise schedule more analogous to those in Gaussian diffusion, attrition as an alternative to remasking, and explicit loss weighting. The paper also introduces a randomized rounding scheme at inference time to prevent mode collapse in the small $y$ regime.

**Compliance With Llm Reviewing Policy:**

Affirmed.

**Final Justification:**

The rebuttal reinforced my prior assessment, my initial and final scores are both 5. The additional results and comments in the rebuttal will be valuable additional to the final paper, especially those on the role of generative models vs. conditional means for single cell imputation tasks. Overall I recommend acceptance.

**Key Questions For Authors:**

1. There seems to be a discrepancy between the FID numbers reported for Blackout diffusion in Table 1 of this paper and the FID numbers reported in its original paper [1] (Table 1). What would explain this?
2. Which further design and hyperparameter optimizations would you consider to be most fruitful in realizing the potential of this framework for scRNA-seq data imputation?

[1] Blackout Diffusion: Generative Diffusion Models in Discrete-State Spaces - https://arxiv.org/abs/2305.11089

**Limitations:**

Yes

**Strengths And Weaknesses:**

Strengths:
- The paper is well written and clearly explains the previous related works and its contribution.
- The framework introduced by the paper is a principled extension of Blackout diffusion that extends the design space and allows for many features from modern continuous diffusion models in the count domain.


Weaknesses:
- For the scRNA-seq imputation results in Tables 2 and 3, Conditional Mean is a very strong and simple baseline which achieves the strongest performance in at least 1 metric in each of the tables. This raises the question of whether in practice it would even be worth using discrete diffusion models for scRNA-seq data imputation. That being said, the authors do note further design and hyperparameter optimization could improve performance.
- Standard Blackout diffusion is a missing baseline in Tables 2 and 3.

Minor note: The table at the bottom of Figure 2 would be easier to read if it was showing the diff between the true and modelled variances, rather than the raw variances.

---

> ### Author Rebuttal · Authors · 2026-03-31
>
> We thank the reviewer for their positive assessment of our work, and address their concerns regarding its applicability to scRNA-seq data imputation and missing baseline.
>
> **W1:** Indeed the conditional mean is a strong baseline on some benchmarks, but it is not directly comparable to CountsDiff and other generative models that output samples from the learned conditional distribution. Nonetheless, it highlights limitations of non-generative imputation methods (MAGIC, GAIN, Hi-VAE), which approximate expectations, only learning the first moment of the conditional. Conditional means and non-generative methods perform poorly in distributional metrics [scFID in Table 2, 3 of paper, attached table 1] compared to CountsDiff. Furthermore, despite strong average performance, conditional mean imputation is ill-suited for tasks that require preservation of distributional structure across cells, such as learning gene programs within cell types or gene regulation
>
> Regarding the missing baseline in Tables 2 and 3, we originally omitted Blackout Diffusion as it is a special case of CountsDiff. However, we recognize its value in showcasing the benefits of the additional design choices; see table 1 for results, along with competitive deep learning models and additional metrics for comparison.
>
> Table 1 Blackout (baseline), CountsDiff, and competitive deep learning models on Fetus data with 50% MCAR dropout. CountsDiff uniformly outperforms on distributional metrics. scFID, ED, SWD, MMD are distributional metrics
>
>
> | Model | Spearman | Pearson | RMSE | MAE | Bias | log scFID | ED | SWD | MMD (avg across 5 bandwidths) |
> | :---- | ----- | ----- | ----- | ----- | ----- | ----- | ----- | ----- | ----- |
> | Blackout | 0.11 (0.00) | 0.16 (0.00) | 1.43 (0.04) | 0.48 (0.00) | 0.04 (0.00) | \-7.32 (0.02) | 0.02 (0.00) | 0.08 (0.01) | **0.0 (0.00)** |
> | CountsDiff 1 | 0.09 (0.00) | 0.13 (0.00) | 1.34 (0.01) | 0.48 (0.00) | **0.0 (0.00)** | **\-9.06 (0.03)** | **0.01 (0.00)** | **0.06 (0.00)** | **0.0 (0.00)** |
> | CountsDiff 5 | 0.12 (0.00) | 0.2 (0.00) | 1.2 (0.03) | 0.44 (0.00) | **0.0 (0.00)** | \-7.55 (0.02) | 0.3 (0.00) | 0.09 (0.01) | **0.0 (0.00)** |
> | ReMDM 1 | 0.11 (0.00) | 0.15 (0.00) | 1.72 (0.05) | 0.46 (0.00) | **\-0.0 (0.00)** | \-8.92 (0.02) | **0.01 (0.00**) | 0.14 (0.01) | **0.0 (0.00)** |
> | ReMDM 5 | 0.12 (0.00) | 0.22 (0.00) | 1.22 (0.02) | 0.42 (0.00) | **\-0.0 (0.00)** | \-8.56 (0.03) | 0.28 (0.00) | 0.08 (0.00) | **0.0 (0.00)** |
> | scGPT pretrained | 0.13 (0.00) | 0.15 (0.00) | 1.19 (0.03) | 0.46 (0.00) | \-0.44 (0.00) | \-4.64 (0.01) | 0.51 (0.00) | 0.11 (0.00) | **0.0 (0.00)** |
> | scGPT | **0.17 (0.00)** | 0.25 (0.00) | **1.03 (0.02)** | **0.35 (0.00)** | \-0.2 (0.00) | \-5.97 (0.01) | 0.25 (0.00) | 0.09 (0.00) | **0.0 (0.00)** |
> | xTrimoGene | **0.17 (0.00)** | **0.27 (0.00)** | 1.06 (0.02) | 0.38 (0.00) | \-0.11 (0.00) | \-6.72 (0.01) | 0.37 (0.00) | 0.08 (0.00) | **0.0 (0.00)** |
>
>
>
> **On the minor note:** While it may be easier to read the rankings of the four models, we found it more complete to keep the absolute variances as to not remove information about the ground-truth variance.
>
> Key Questions: **KQ1**: Blackout Diffusion [1] did not share pretrained weights, and only some architecture-specific hyperparameters. FID hyperparams were also not shared. We base our image models on the Diffusers library and matched reported hyperparams when possible, and made judgement calls otherwise. Though absolute values of FID changes, we find that the ranking of models is consistent, even across drastic hyperparameter changes such as model depth and training steps. See table 2 for an example.
>
> Table 2: comparison of attrition rate sweep for small and large model: trends and rankings independent of model size/train time
>
> | Attrition Rate (eta rescale) | FID (4 res layers per block, 1M steps) | FID (2 res layers per block, 300k steps) |
> | :---- | :---- | :---- |
> | 0.0 | 9.666 | 11.189 |
> | 0.005 | 9.562 | 10.844 |
> | 0.01 | **9.507** | **10.696** |
> | 0.02 | 10.300 | 11.428 |
> | 0.05 | 12.727 | 14.528 |
>
> **KQ2**: One promising scRNA-seq imputation specific direction is to combine CountsDiff with an imputation-specific training objective, such as that described in scIDPMs. For broader applications, exploring the attrition schedule is promising and exciting: remasking literature proposes linking remasking rate to prediction confidence (Kim et al. 2025), one could similarly link attrition rate to confidence in CountsDiff. Loss weighting has also been shown to be beneficial to Gaussian diffusion modeling (Kingma & Gao, 2023\) these strategies can be readily transferred to CountsDiff as well.
>
> References:
> Kim, Jaeyeon, et al. "Fine-tuning masked diffusion for provable self-correction." arXiv preprint arXiv:2510.01384 (2025).
> Kingma, Diederik, and Ruiqi Gao. "Understanding diffusion objectives as the elbo with simple data augmentation." Advances in Neural Information Processing Systems 36 (2023): 65484-65516.

---

> > ### Author Rebuttal · Reviewer_pnhx · 2026-04-03
> >
> > Thank you for the response and additional results. I maintain my score and confidence level as I still recommend acceptance. Please make sure to emphasize the points brought up about the role of generative models vs. conditional means for single cell imputation tasks and the include the additional results in the final version of the paper.

---

### Decision · Program_Chairs · 2026-04-30

**Decision:**

Accept (regular)

**Comment:**

This paper proposes CountsDiff as an extension of Blackout diffusion framework, which aims to work more compatible on count-valued data such as on the natural numbers.  The idea is built on a survival-probability parameterization and extends count diffusion with successful modern design elements from continuous diffusion, such as continuous-time training, classifier-free guidance. Empirically, the paper shows that CountsDiff scales beyond toy count data to image generation and on scRNA-seq imputation, suggesting that it is a promising framework for structured ordinal/count domains.

The strength is the problem formulation. Modeling count data natively is well motivated, and the paper clearly explains why both real-valued relaxation and categorical discretization are imperfect for natural-number data. In addition, the survival-probability view makes the method easier to relate to standard diffusion design choices, and the additions of weighting, continuous-time training, guidance, and attrition make the framework feel like a genuine modernization of Blackout Diffusion.


The reviewers have made comments on the presentations and the clarity regarding the motivation and relavant prior works to make the contribution in a clear way. On the experiment side, the reviewers have concerns that the validation only partially supports that motivation. In particular:

- While the image-generation experiments on CIFAR and CelebA are useful for demonstrating that the method can scale to high-dimensional settings and produce visually plausible guided samples, the FID is only compared with BlackOut diffusion, and the FID is weak compared to discrete diffusions, which also target to model the data distribution with categorical value [1,2,3]. Thus the motivation that count diffusion could work better on count-value data is not fully supported in this experiment.

- The scRNA-seq imputation evaluation had missing baselines and the metrics are not comprehensive. Also the benchmark is not strong enough and the dimensionality-efficiency-accuracy tradeoffs are not analyzed.

After rebuttal the presentation and scRNA-seq imputation evaluation are improved, and most of the reviewers are supportive to the paper, while the concerns regarding image experiment still exists. The authors should carefully revise the paper and are encouraged to consider more experiments, such as work on language generation tasks to further support the effectiveness of the method.

------

[1] Subham Sekhar Sahoo, Marianne Arriola, Aaron Gokaslan, Edgar Mariano Marroquin, Alexander M Rush, Yair Schiff, Justin T Chiu, and Volodymyr Kuleshov. Simple and effective masked diffusion language models. In Advances in Neural Information Processing Systems (NeurIPS), 2024.

[2] Chen-Hao Chao, Wei-Fang Sun, Hanwen Liang and Chun-Yi Lee, and Rahul G. Krishnan. Beyond Masked and Unmasked: Discrete Diffusion Models via Partial Masking. In Proceedings of the Conference on Neural Information Processing Systems (NeurIPS), 2025.

[3] Huangjie Zheng, Shansan Gong, Ruixiang Zhang, Tianrong Chen, Jiatao Gu, Mingyuan Zhou, Navdeep Jaitly, and Yizhe Zhang. Continuously Augmented Discrete Diffusion model for Categorical Generative Modeling. In The Fourteenth International Conference on Learning Representations (ICLR), 2026.